# Carbon Nanomaterials: Synthesis, Functionalization and Sensing Applications

**DOI:** 10.3390/nano11040967

**Published:** 2021-04-09

**Authors:** Giorgio Speranza

**Affiliations:** 1CMM—FBK, v. Sommarive 18, 38123 Trento, Italy; speranza@fbk.eu; 2IFN—CNR, CSMFO Lab., via alla Cascata 56/C Povo, 38123 Trento, Italy; 3Department of Industrial Engineering, University of Trento, v. Sommarive 9, 38123 Trento, Italy

**Keywords:** carbon nanostructures, fullerenes, nano onions, quantum dots, nanodiamonds, carbon nanotubes, graphene

## Abstract

Recent advances in nanomaterial design and synthesis has resulted in robust sensing systems that display superior analytical performance. The use of nanomaterials within sensors has accelerated new routes and opportunities for the detection of analytes or target molecules. Among others, carbon-based sensors have reported biocompatibility, better sensitivity, better selectivity and lower limits of detection to reveal a wide range of organic and inorganic molecules. Carbon nanomaterials are among the most extensively studied materials because of their unique properties spanning from the high specific surface area, high carrier mobility, high electrical conductivity, flexibility, and optical transparency fostering their use in sensing applications. In this paper, a comprehensive review has been made to cover recent developments in the field of carbon-based nanomaterials for sensing applications. The review describes nanomaterials like fullerenes, carbon onions, carbon quantum dots, nanodiamonds, carbon nanotubes, and graphene. Synthesis of these nanostructures has been discussed along with their functionalization methods. The recent application of all these nanomaterials in sensing applications has been highlighted for the principal applicative field and the future prospects and possibilities have been outlined.

## Contents

IntroductionProperties of Carbon NanostructuresFullerenes and Carbon Onions3.1.Fullerene Synthesis3.2.Fullerene Functionalization3.3.Carbon Onion Synthesis3.4.Carbon Nano-Onion Functionalization3.5.Fullerenes Sensing3.6.Carbon Nano-Onion SensingNanodiamonds4.1.Nanodiamond Synthesis4.2.Nanodiamond Functionalization4.3.Nanodiamond SensingCarbon Quantum Dots5.1.Carbon Quantum Dot Synthesis5.2.Carbon Quantum Dot Functionalization5.3.Carbon Dots SensingCarbon Nanotubes6.1.Carbon Nanotube Synthesis6.2.Carbon Nanotube Functionalization6.3.Carbon Nanotube SensingGraphene7.1.Graphene Synthesis and Functionalization7.1.1.Bottom-Up Synthesis7.1.2.Top-Down Synthesis7.2.Graphene Functionalization7.2.1.Covalent Functionalization7.2.2.Non-Covalent Functionalization7.3.Graphene SensingConclusions

## 1. Introduction

Novel technological solutions are needed to face emerging global challenges such as environmental pollution, massive energy production, need of additional agricultural and food production. To increase the productivity while reducing the generation of pollutants, new advanced systems are needed to improve and automate operations which autonomously can monitor infrastructures, the environment, and the processes efficiencies [1,2,3,4,5,6,7,8]. In this respect, the potentialities enabled by sensing technologies are considerable. 

The global gas sensor market size was estimated at USD 2.33 billion in 2020 and is expected to reach USD 2.50 billion in 2021. There is growing request of sensing power to monitor the environment, indoor areas, the health conditions, the industrial processes, the components in operando conditions (see applications in automotive), and sparse controls in smart cities. In this respect, the electrochemical segment dominated the market in 2020 and accounted for a share of more than 21.0% of the global revenue [9].

Nowadays there is broad class of sensors for checking gas molecules, heavy metals, humidity, recognize biomolecules, toxic substances, pH, pressure, and many others [10]. However, most of them display a non-ideal limit of detection (LOD), scarce sensitivity and/or selectivity, have slow responses, need pretreatment and are expensive. In this panorama nanomaterials can play a crucial role because they offer a solution to the limitations of conventional systems providing important advances in the material properties [10].

A revolution in material science occurred with the recognition that reduction in the nanorange of at least one of the system dimensions introduces novel unusual properties. In this respect, nanomaterials open the perspective to improve parameters such as the sensibility and the reliability, make the response and recovery times shorter, open the possibility to perform in situ analysis, and their cost is low. All these properties are required for producing sensor devices.

Nanomaterials are attractive for a number of different applications, including energy production, biomedical applications, environmental protection, information technology, food, agriculture and many others. Then, significant efforts have been recently devoted for both the mass production of structurally homogeneous nanomaterials with well controlled surfaces and interfaces and their assembly into device architectures. In this scenario, the carbon-based nanomaterials have become one of the dominating materials in several sensor applications. A literature survey reveals that more than 2200 publications are related to nanomaterial and sensing and roughly about 50% of them regard carbon nanomaterials [11].

Among the various chemical elements, carbon plays a very special role. It provides the basis for the life in nature, it displays different orbital hybridization leading to the ability to generate different chemical bonds with different orientations. For this reason, carbon possesses different allotropic forms (graphite and diamond) and has the capability to generate a list of nanostructures namely graphene single sheets, mono and multiwalled carbon nanotubes, carbon fibers, fullerenes, onions, and nanodiamonds. In addition, carbon is able to bind to nearly all chemical elements generating an unlimited variety of molecules and compounds [12] possessing a number of different chemical and physical properties. Typically the size of carbon nanomaterials ranges from 1 to 100 nm and leads to significant changes with respect to the bulk counterparts namely the increased surface to volume ratio, the nanostructure shapes, the different chemical reactivity, the different optical properties… [13].

As a consequence, carbon nanomaterials are widely utilized in many sectors. They are used in environmental applications for water treatment [14,15], and other separation processes [16,17], for environment remediation [18,19,20]; in electronics where they have shown remarkable utility for the excellent electrical [21,22] and optical properties [23,24,25]. This, combined with the molecular sized diameter and microscopic structure enable the development of novel electronic devices [26]. The high mechanical strength, electrical and thermal conductivity [27,28,29] make them ideal as reinforcing elements [30,31,32], as protective materials [33,34,35] and to make conductive polymers [36,37]. Carbon nanomaterials find application also in the biomedical field for sensing applications and in controlled pharmaceuticals and drugs delivery [38,39,40,41,42].

As mentioned before, carbon (C) atoms are able to organize themselves in different structures as depicted in Figure 1. When C atoms are arranged in a honeycomb lattice, they form the graphite crystal, a stack of two-dimensional single sheets. The single graphite layer constitutes the graphene [43] atomic crystal. Graphene nanostructures (GNs) were firstly isolated by the Nobel-prizes Geim and Novoselov in 2010. Another carbon-based structure is carbon nanotube (CNT). Discovered by Iijima in 1991, it may be regarded as a single graphene layer rolled along an axis aligned along the graphene crystalline directions [44]. As CNTs also carbon fibers (CFs) are unidimensional system. However, CFs are disordered, tangled structures possessing a two-dimensional long-range order of C atoms organized in planar hexagonal networks while in the direction orthogonal to these planes, CFs display only a short range order due to parallel plane stacking [45].

If in unidimensional carbon structures we reduce their length to the nanometer size we will obtain nanocages. Fullerenes discovered by Kroto and Smalley in 1985 is a perfectly spherical nanocage formed by a number of pentagonal and hexagonal rings [46]. Carbon nano-onions (CNOs) are cages with spherical or polyhedral shape formed by several fullerene-like overlapped carbon shells which are defective and disordered to a certain degree [47]. They were discovered in 1992 by Ugarte during electron beam irradiation of an amorphous carbon sample using a TEM microscope [48]. Another unidimensional carbon nanostructure (CNS) is represented by the carbon dots (CDs). CDs are nanoparticulate where graphitic and amorphous carbon phases coexist. Typically, the average dimension of carbon dots is about 5 nm. Quantum confinement effects induce excellent optical properties as highly tunable photoluminescence (PL), high photostability easy functionalization of their surface and biocompatibility [49] make them good competitor of quantum dots based on toxic chemical elements such as cadmium. Another CNS with excellent degree of biocompatibility is the nanodiamond. Nanodiamonds (NDs) were discovered in the sixties [50]. Depending on the synthesis process, NDs dimensions are in the 5–100 nm range. In NDs, C atoms are sp^3^ hybridized orbitals [51] leading to the formation of the hexagonal or cubic diamond lattices. NDs possess distinctive electronic and optical properties deriving from dopants (N, Si, Ge…) present in the structure as defects [52].

This review complements previous works on this topic giving a complete description of carbon-based nanostructure’s properties, of their synthesis and functionalization processes. For each of these nanostructures, an extensive survey of the sensing applications is also provided together with an accurate summary of the detection modalities, the kind of surface chemistry and the analyte sensed to facilitate the consultation and retrieve the information sources.

## 2. Properties of Carbon Nanostructures

Carbon is a unique element of the periodic table possessing the extraordinary capability to organize its four valence electrons in different hybridization states, namely sp, sp2, sp3 leading to both strong covalent and weak π-π bonds. The different hybridizations enable C-atoms to assume different allotropic forms with diamond and graphite as main prototypes and to form a wide range of structures, from small molecules to long chains [53]. The different organization of C-atoms in the crystalline lattice of graphite or diamond is also accompanied by rather different physical properties. Graphite is a stack of weakly bonded single layers where carbon atoms are organized in a honeycomb structure. Because the weak van der Waals interaction between layers, the π electrons are quasi-free thus leading to the semimetallic character of graphite [54]. The valence and conduction bands are overlapped in a point thus leading to a zero optical gap. As a consequence, the graphite has a dark aspect with high absorption coefficient. Different is the case of diamond where the sp^3^ hybridization generates four strong covalent bonds oriented along the axis of a tetrahedron and to a face centered Bravais lattice. In this case mobility of the electrons is absent and diamond is a highly insulating material characterized by an optical gap as high as 5.5 eV. The high optical gap makes pure diamond one of the most broadly transmitting of all materials [55]. It is transparent over a wide optical range extended well out from the visible regions (observe that absorption lines are present due to impurities mainly N, B, H, Ni…). Its transmission spectrum shows a flat featureless window for wavelengths longer than ~225 nm and moderate absorption in the range 2.6 to 6.2 μ m due to multiphonon processes [55].

Apart from these two representative forms, carbon can also organize in amorphous structures. However, some short-range order can be observed in amorphous carbon (aC) phases. In aC both graphitic sp^2^ and tetrahedral sp^3^ hybrids coexist. The prevalence of one or the other of these hybrids imparts properties mirroring those of graphite or diamond. In the first case the amorphous carbon has poor mechanical properties, high extinction coefficient. Differently, in diamond-like-carbons and in highly tetrahedral amorphous carbon are very hard and the optical gap can be increased till to 4.5 eV with correspondent high transparency. Presence of hydrogen can also modulate the aC properties leading to both hard and polymer-like structures which can be interesting as biomaterials. Amorphous phases can be present in CNS or amorphous carbon nanoparticles can be produced [56,57,58,59] for their optical [60], electrical [61], mechanical properties [62].

As in the macroscopic allotropes, also the properties of a given CNS depend on the type of hybridization assumed by carbon atoms in the nanostructure. However, due to the quantum confinement imposed by the nanometric dimension or on structural arrangements as in fullerenes some differences appear. Considering the applicative point of view, the transduction of a given physical entity (generally an electrical signal) is made by the integration of a CNS into a microsystem which is used in the development of electronic, photonic, and optoelectronic devices. The selection of a given nanostructure depends on the signal to transduce and on the physicochemical properties of the CNS. In the following we will consider the properties of the single carbon nanoallotropes and the relative applications which are summarized in Figure 2.

## 3. Fullerenes and Carbon Onions

As seen, fullerenes are closed hollow cages in which C-atoms are sp^2^- hybridized carbon atoms. However, to generate the spherical cage, the structure cannot be generated using only hexagonal rings. Following Euler’s theorem, it is possible to show that a spherical surface must contain exactly 12 pentagons. Fullerenes may be regarded as a class of closed-cage carbon molecules, C_n_, where n indicates the total number of carbon atoms forming 12 pentagons and a variable number of hexagons. Depending on the number of hexagons, fullerenes of different sizes are obtained. The number *n* may assume values *n* = 20 and *n =* 20 + 2k (k = 1, 2, 3…). A fullerene containing 22 C-atoms does not exist. The different dimensions and shapes of fullerenes have significant effects on their optical and chemical properties. For example the absorption spectra of the fullerene depend on the number of atoms *n* as pictorially showed in Figure 3 where the color changes with *n*. Fullerenes display a hydrophobic trait and the solubility decreases with increasing their size [63]. For their aromatic character fullerenes are generally soluble in organic hydrocarbons and halogenated solvent [64]. The dimension of fullerene structure affects also its reactivity. In addition, one of the important driving forces is the relief of strain affecting the cage structure, enabling the return to sp^3^ hybridization. Then the chemical reactivity of fullerenes increases as their size increases resulting in a reduced curvature and strain towards a more graphitic-like planar surface [65].

### 3.1. Fullerene Synthesis

C_60_ is the more common among the fullerenes and the most widely studied. However, despite numerous attempts of explanation, still the mechanism of fullerene formation is unknown. The common methods to produce C_60_ vaporization of graphite electrodes using arc or plasma discharges [66,67,68] or using laser ablation [46,66,69], pyrolysis of hydrocarbons [70,71,72]. This process was optimized for mass production by Murayama [73] leading to a soluble mixture of 60% C_60_, 25% C_70_ and a remaining 15% composed by bigger fullerenes up to C_96_. Initially the fullerenes are separated from soot using solvents as benzene or toluene. The remaining product is processed using column or liquid chromatography to separate fullerenes of different sizes [74].

Concerning carbon-onions (COs), these structures were obtained irradiating CNTs using an electron beam [75]. A modification of this method introduces gold nanoparticles promoting the formation of the nanoparticles with less extreme irradiation conditions [76] Bigger amounts of COs are produced by heating carbon soot at 2100–2250 °C under vacuum [77]. The nanostructures obtained with this method generally have a non-spherical shape and are formed by 2–8 shells with diameters of 3–10 nm. Using arc discharge in water it is possible to obtain bigger spherical COs with a 25–30 nm diameter [78,79]. More recent synthesis process are based on solution ozonolysis [80] or by electrochemical processes [81].

C_60_ has an external diameter of 0.71 nm, and its chemical properties are very similar to those of an organic molecule. However, the C_60_ molecule is considered to be not superaromatic due to the presence of pentagons where C-atoms have the tendency to avoid formation of double bonds. In particular it is shown in literature that fullerene the C_60_ molecule will undergo a facile reduction, and because of the surface curvature of the surface, fullerene hybridization falls between sp^2^ and sp^3^. This together with topology account for the extraordinary ability of C_60_ to accept electrons [82]. This characteristic has a peculiar influence on the fullerene chemical reactivity and electrochemical properties and the large number of fullerene functionalization reactions that can be made on fullerenes.

### 3.2. Fullerene Functionalization

The list of possible chemical reactions utilized to functionalize fullerenes is very long. Here we will present a selection of the most important reactions which are reviewed in [65,83,84,85]. Surface functionalization is a convenient route making fullerene soluble in both water and organic solvents [86,87]. Chemical modification of the fullerene surface may be performed following two different methods: (i) complexation with solubilizing agent to partially hide the fullerene hydrophobic surface [88]; (ii) covalent functionalization of the fullerene surface [89]. Oxygen based functional groups, mainly hydroxyl groups, may be grafted on the fullerene surface using strong acids at high temperature [90].

C_60_ and C_70_ fullerenes were mixed with an aqueous solution of nitric and sulphuric acids at a temperature of 85–115 °C. The acid induced oxidation of the fullerenes surfaces and to a grafting of about 15 hydroxyl groups/fullerene in average. Another possible reaction to form OH functionalized fullerenes (fullerenols) can be performed also adding strong basic NaOH to C_60_ benzene solutions in presence of tetrabutylammonium hydroxide acting as a catalyzer [91]. The reaction is more efficient than that based on strong acids resulting in a higher number of grafted hydroxyl groups. In another work, a drop of liquid Na/K alloy was added to a suspension of C_60_ in tetrahydrofuran and a subsequent reaction of this intermediate with O_2_ and water [92]. The reaction is very efficient and produced an extended hydroxylation of the fullerenes, inducing hydrophilicity among the highest ever reported in literature. These reactions are broadly used to make fullerenes hydrophilic despite the lack of control on the density of addends. An alternative method to produce highly polyhydroxylated fullerenes is based on the use of a suspension of partially hydroxylated fullerenes and hydrogen peroxide at a temperature of 60 °C [93]. The reaction is rather slow (4 days) but results in fullerenols with 36–40 hydroxyl groups in average, with 8–9 secondary bound water molecules. The product exhibited a high water solubility up to 58.9 mg/mL at pH = 7. Description of other methods of fullerenol synthesis may be found in [94,95,96,97,98].

Grafting of amine groups is another common functionalization process performed mixing C_60_ with different aliphatic primary amines as n-propylamine, t-butylamine, and dodecylamine [99]. In other reactions, smaller primary, secondary amine chains as methylamine, diethylamine or ethylenediamine (as shown in Figure 4A) are reacted with C_60_, C_70_ fullerenes [100]. Another popular amination reaction is the 1,3-dipolar cycloaddition of an azomethine ylide to a C_60_ molecule (Prato reaction, see Figure 4B) produces a stable compound grafting a pyrollidine ring to C_60_ [101]. The beauty of this reaction is the possibility to introduce different functional groups to the fullerene moiety. Another possible reaction for the generation of well-defined amine addition products utilizes a modest excess of secondary diamines like N,N′-dimethylethylenediamine or piperazine resulting in both mono- and bisadducts with an overall 50–85% yield [102]. Addition of amine to C_60_ can take place at room or high temperature involving electron transfer process [103]. In [104] diamine were added to C_60_ at low temperature. The reaction was performed to prepare fullerene diamine adduct which was the starting material for the synthesis of C_60_—polyamides. Amination was performed adding diamine, l,4-Butanediamine, 1,6-Hexanediamine, to a benzene solution of C_60_ and diamine in benzene in appropriate proportions. The reaction led to the formation of the desired adducts in 80–89% yields. Formation of Penta- and Hexaamino- [60] fullerenes was obtained in presence of oxygen and secondary amines such as piperidine and N-methylbenzylamine and C_60_ fullerenes [105]. A solvent-free chemical modification of C_60_ fullerene surface was carried out using amino terminated polyethylene glycol [106]. Authors studied the effect of the stirring rate and temperature on the C_60_ modification yield.

As a result, the 65% of C_60_ was modified at 60° after 24 h stirring. Amination was also utilized to produce transparent fullerene containing sol-gel glasses. Fullerene amination was performed at 100 °C under nitrogen by reacting C_60_ with 6-amino-1-hexanol, cyclohexylamine, 2-(2-aminoethoxy)ethanol, and 3-aminopropyltriethoxysilane [107]. After filtration and purification, the authors obtained a yield from 32% to 82%. A review on the reaction of between aliphatic amines and [60] fullerene may be found in [108].

The Bingel reaction is widely utilized for fullerene exohedral functionalization (see Figure 4C). This reaction involves the addition of an α-halo ester/ketone to the cage under strongly basic conditions resulting in a methanofullerene [109]. There are several examples of the application of the Bingel reaction to modify the chemistry of the fullerene surface. For example, the Bingel reaction was used to synthesize C_60_ Hexakis-adducts [110]. Authors found that the original Bingel-Hirsch method cannot be utilized while a modification to the original Bingel reaction without the use of any templating agent was needed for a high-yield synthesis of the compound. Differently the Bingel-Hirsch reaction was used with C_60_ and 2-methacryloyloxyethyl-Me-malonate or 2-methacryloyloxyethyl-dichloroacetate to produce fullerene-containing methacrylates [111].

Same reaction was used to produce a bis-malonate C_60_ derivative with terminal alkyne groups [112]. The reaction was carried out using copper-catalyzed azide-alkyne cycloaddition to produce a series of new fullerene glycoconjugate derivatives. C_60_ and an enantiopure bismalonate tether equipped with two acetonide moieties were Bingel-reacted to produce an enantiomerically pure [60] Fullerene bisadducts [113]. The produce combines the inherent chirality of the fullerene core with the functional glycol groups located on the tether. A variation of the Bingel reaction based on cyclopropanation of [60] fullerene with bromo-substituted active methylene compounds was used to produce methanofullerene in the presence of amino acid and DMSO without the use of catalysts [114]. More information about the fullerene functionalization chemistry including the use of the Bingel reaction may be found in [115,116]

Among the different chemical processes available for functionalizing fullerenes, the Diels-Alder cycloaddition reactions are very versatile resulting in a large variety of cycloadducts (see Figure 5A). The fullerene being electron-deficient reacts with electron-rich 1,3-dienes to form adducts in which a 6-6 bond is fused to a cyclohexene ring. An example is the reaction of C_60_ with anthracene or other 1,3-dienes [116]. Diels-Alder cycloaddition was also utilized for reacting 2-[(trimethylsilyl)oxy]buta-1,3-diene with C_60_ after hydrolysis and reduction, obtaining additional functional groups such as α-amino acid derivatives could be attached through esterification. There is a large variety if cycloaddition reactions utilized to modify the fullerene surface chemistry upon desired properties [116,117,118,119,120] as shown in Figure 5B,C.

A variety of reactions can be performed to attach complex molecules to fullerenes. Some of the functionalization reactions are sketched in Figure 6.

### 3.3. Carbon Onion Synthesis

Concerning carbon nano-onions, besides electron irradiation there are several methods for the synthesis developed at the end of the 20th century. CNOs can be produced by decomposition of are acetylene, boron trichloride and ammonia precursors using hydrogen as a carrier via chemical vapor deposition (CVD) [121]. The resulting CNOs are containing quasi-stoichiometric boron-nitride domains. In another work CNOs were produced by decomposition of methane via CVD with the help of a Ni/Al catalyst. CVD was also utilized by other authors to synthesize metal-containing CNOs [122]. Firstly, a sol-gel containing Ni and Sn was spin-coated on a Si substrate and dried and annealed to 600 °C to obtain the catalysts. Then decomposition of a mixture of methane and hydrogen was performed by CVD leading to the formation of the CNOs containing Ni_3_Sn_2_ core. Ni was utilized as a catalyzer also in a counterflow diffusion flames to synthesize CNTs and CNOs [123]. A mixture of 5% ethylene and 15% to 45% methane and the remaining part oxygen was supplied in an upper burner while a mixture of oxygen and nitrogen was used in a lower burner. In low flow rate conditions, a quasi-stable one dimensional flame with working temperatures in the range 400–1200 °C is obtained. Changing the ethylene, methane and oxygen concentrations, the decomposition of methane led to the formation of high yield of CNTs or CNOs. A combustion flame was also utilized burning naphthalene to produce gram-scale of CNOs [124]. Other precursors as phenolic resins [125] and methods to improve the efficiency of the CNOs production are make use of resonant excitation of precursor molecules [126], acoustic modulation of the laminar flame [127]. CNOs may also be obtained transforming nanodiamonds via thermal treatment in vacuum at ~1700 °C [128]. The process starts at the surface with the formation of a graphitic outer shell on the diamond core which then transforms into a carbon onion surrounding an inner sp^3^ phase. Increasing the temperature, the diamond phase transforms in a facetted graphitic structure. In another study, the evolution of the diamond to onion transformation as a function of the temperature was analyzed [129].

Depending on the process, detonation nanodiamonds obtained at high temperatures (up to 2000 °C) and high pressure (up to 200 GPa), were purified from soot in H_2_SO_4_ and HClO_4_ acids and then heated in vacuum at temperature ranging from 900 °C to 1400 °C. The analysis of the evolution of the diamond to onion transformation, shows that the graphitization process starts at 900 °C and proceeds with increasing the temperature till a complete transformation of the diamond phase into onions at 1400 °C. The type of precursor and the synthesis conditions have a strong impact on the structure, as seen from transmission electron micrographs of different types of carbon onions given in Figure 7, but all carbon onions share the multi-shell fullerene-like architecture.

### 3.4. Carbon Nano-Onion Functionalization

Similarly to fullerenes also pristine CNOs show different solubility in polar, non-polar, protic and aprotic solvents [131]. The different behavior respect to solvents can be explained through their capability to be good proton acceptors as in the case of N-N-dimethylformamide. This leads to negative ζ-potentials values and good CNOs solubility. Differently, as in the case of chloroform, tetrachloroethane, CNOs are unable to donate protons to the medium. This leads to a positive ζ-potential thus resulting in a poor solubility. Increasing the solubility may be obtained functionalizing the CNO surface. The presence of curvature renders the CNO functionalization rather easy via covalent and non-covalent reactions. One of the common reactions is the oxidation of CNOs obtained using oxidative or reductive reagents such as HNO_3_ or H_2_SO_4_ or KOH. However, the use of strong acids or bases can damage the CNOs. These strong acids introduce ether-like, hydroxyl, carboxyl functional groups. An alternative route is the use of nitric acid or ozone to attack the CNOs surface grafting oxygen functional groups [132]. Carboxyl groups and azomethine ylide addition (1,3-dipolar cycloaddition) reactions were also utilized to make CNOs soluble [133]. Further reactions with carboxylated CNOs may involve direct acid–base interaction, amidation via in situ generated acid chloride, and carbodiimide-activated coupling. For example, amidation of carboxylated CNOs is performed by direct reaction with diamine terminated polyethyleneglycole. Another possible amidation reaction is obtained reacting carboxylated CNOs with 1-Octadecylamine described as a very good agent for the amidation for CNTs [134]. Introduction of NH groups may be obtained also by microwave irradiation of CNOs suspended in dimethylformamide [133]. Finally, 1,3-dipolar cycloaddition of azomethine ylide is another route to functionalize the CNOs surface [133,135]. The azomethine ylides can be produced by condensation of an α-amino acid and an aldehyde, a reaction which was extensively applied to the organic modification of fullerene C_60_ [101]. In another example, a [2+1] Bingel-Hirsch cyclopropanation was used to functionalize the CNOs in presence of dodecyl malonate ester, carbon tetrabromide and 1,8-diazabicyclo[5.4.0]undec-7-ene [136].

Well-dispersible CNOs can be obtained through nucleophilic substitution [137]. The functionalization proceeds in two steps: first the CNOs are reduces in presence of Na-K alloy in 1,2-DME under vacuum, follows the covalent functionalization by an electrophile precursor (1-bromohexadecane). The alkylated CNOs exhibited high dispersion in polar solvents. A covalent CNOs amidation can be performed using 4-aminopyridine, 1-octadecylamine (ODA), polyethylene glycol (PEG1500N), disulfide derivatives, [133,138,139]. Complexation of CNOs was also produced using the procedure indicated by Prato resulting in N-Boc-protected amino-CNOpyrrolidine [140]. Another interesting chemical modification of the CNO surface is the formation of CNO derivatives using folic acids polyethylene glycol and fluorescein species using a copper-catalysed azide-alkyne Huisgen cycloaddition [141]. Functionalized CNOs are used as both imaging and targeting cancer cells. The modified CNOs display high brightness and photostability in aqueous solutions, rapid uptake in two different tumor cell lines and negligible cytotoxicity. CNOs may be functionalized also attaching polymeric molecules. As an example using polypyrrole CNO the specific capacitance of supercapacitors are improved [142]. CNOs are coated with polypyrrole using chemical polymerization or electrostatic potentiostatic deposition to form bilayers. The composites obtained with the two techniques show higher mechanical and electrochemical stabilities, with high specific capacitances with specific capacitances of 800 Fg^−1^ and 1300 Fg^−1^ respectively. Other non-covalent functionalization of the CNO surface include surfactants as and poly(4-vinylpyridine-co-styrene) (PVPS) or poly(ethylene glycol)/polysorbate 20 (PEG-P20). Successive attachment with 3-mercaptopropionic or 2-mercapto-4-methyl-5-thiazoleacetic acids enabling binding phenolic compounds as quercitine [143]. Another example is the modification of CNOs with PEG to attach phenolic compounds as 1,2,3,4,6-penta-β-O-galloyl-d-glucopyranose (β-PGG) and gallic acid molecules known for their protecting properties on mammalian cells. The CNO/PEG systems may be used as signal amplifier in the analytical biosensors or for drug targeting applications. Complete description of the surface functionalization of CNOs may be found in [133,144].

### 3.5. Fullerenes Sensing

Technology and automation are even more expanding in our everyday life and this requires an even more distributed use of sensors. Some of the potential applications of sensors are in biomedical industry, electronics and automotive industries, environmental monitoring, agricultural and food industries, defense and homeland security. In all these applications the integration of sensors into intelligent devices and systems require improved sensitivity, selectivity and stability to measure, analyze physical entities. Carbon nanostructures including fullerenes, are materials of choice due to their superior properties and we will report here some applicative examples.

Fullerene films and fullerene compounds with iodine are used to fabricate temperature and pressure sensors [145]. Fullerene films of 2–3 μm thickness were deposited by evaporation at a temperature of 600–680 °C. Authors demonstrated that the film resistivity is sensitive to the temperature the humidity and the pressure [145]. Sensitivity of the sensor increases if the surface of the fullerenes films is oxidized. Another moisture sensor was fabricated immobilizing the C_60_ nanoparticles on an alumina substrate [146]. Fullerenes led to a corrugated surface with increased specific surface, resulting in an enhancement of the device sensitivity enabling moisture detection at the ppm level. Thanks to the small pore size of the active surface, this device has the potential to be used successfully in the gas, oil, and food industries.

In electrochemical sensors, electrodes are used as transducers of redox chemical reactions. Among the materials carbon derivative electrodes can be used in combination with the immobilized recognition agent for obtaining high analyte selectivity. Electrochemical sensors may be classified in amperometric if an electroactive species is formed at the electrode, in potentiometric it the electrode detects presence of ions or impedimetric if the coupling with an external analyte leads to a change of the electrode impedance [147,148]. It is well known the electron accepting ability of C_60_ molecules. This property has been exploited to modify many substrates to lower their potential of electroreduction and increasing the reaction rates, improving sensitivity and selectivity in electrochemical sensors [149]. For these properties, fullerenes have been used both in mediators electrochemical catalysis and as redox catalysts [149,150]. Electrodes modified with fullerenes can be used biomolecular sensing and environmental monitoring [151,152]. As an example, nanoporous fullerene C_60_ crystals were obtained via self-assembling process. The C_60_ crystal showed an excellent sensing property in particular for aromatic vapors, due to the easy diffusion through the porous crystal architecture and strong π–π interaction between the aromatic rings [153]. Fullerenes were also utilized to detect toxic substances as bisphenol. The sensor showed linear behavior in a concentration range 74 nM—0.23 μM [154]. In another work, authors utilized fullerenes derivatives and artificial macrocyclic polyethers to coat a quartz crystal [155]. The oscillating frequency of the quartz crystal changed as a result of the adsorption of organic or inorganic molecules on coating material. The sensor exhibited a short response time (<2.0 min.) and a high sensitivity, good selectivity, and good reproducibility for polar organic gases.

C_60_ was also used to drop-coating a film onto an electrode surface which was next coated with a protective Nafion film [156]. Authors showed a quasi-reversible one-electron electroreductions at this fullerene modified electrode CME. Oxygen dissolved in a 2% (C_2_H_5_)_4_NOH water-DMF (3:2, *v*:*v*) solution was electrocatalytically reduced at his electrode and the more negative was the redox potential E° value, the more reversible was the electrochemical process. Therefore, this electrode may find practical application as a sensor. Alternative highly sensitive oxygen sensors rely on optical properties of C_70_ [157,158,159]. In the last work, authors utilized isotopically enriched carbon-13 fullerene C_70_ dissolved in polymer matrixes characterized by different oxygen permeability, as polystyrene (PS), ethyl cellulose (EC) and an organically modified silica gel (OS). Rapid lifetime determination method was applied to determine oxygen concentration. At room temperature, the C_70_ based sensor had a very low LOD dependent on the polymer used: ~250 ppbv for EC, 320 ppbv for OS and ~530 ppbv for PS. In another work, fullerene nanorods were synthesized by liquid–liquid interface and immobilized on the surface of a glassy carbon electrode [160]. To produce a highly conductive electrochemically reduced fullerene nanorod the electrode was electrochemically reduced in 1.0 M potassium hydroxide (KOH). The electrode was utilized for sensing ethylparaben (EP) in a concentration range from 0.01–0.52 μM and showed a LOD 3.8 nM. A stable non-enzymic electrochemical sensors was fabricated coupling zinc porphyrin and fullerene (ZnPp-C_60_) [161]. Then ZnPp-C_60_ was entrapped in tetraoctylammonium bromide film on glassy carbon electrode. The electrode showed excellent reproducibility with extremely fast response in sensing H_2_O_2_ in the range 0.035 to 3.40 mM, with LOD of ~0.81 μM.

Thanks to the biocompatibility of carbon nanomaterials, a wide area of fullerene application is the biosensing where they show sensitive interaction with analytes, efficient transduction of the biorecognition events, and fast response times. The biosensor is based on a transducer element able to convert an interaction with biological molecules into an electrical signal. Among the interactions, that with DNA is very important in life science being in direct contact with the transcription processes, mutation of genes, origins of diseases, and molecular recognition studies [162]. Biorecognition of DNA was accomplished by using [Co(phen)_3_]^3+/2+^ as an appropriate electroactive element [163]. In particular, the interaction with DNA led to a decreased peak current and the electrode recovered significantly in the presence of H_10_C_60_(NHCH_2_CH_2_OH)_10_. An ultra-sensitive fullerene-modified biosensor was made to detect miRNA-141 [164] as depicted in Figure 8.

In the first step A, two sequences containing G-quadruplex were connected by click chemistry-mediated nucleic acid strands as shown in Figure 8. To separate the miRNA, the obtained complete G-quadruplex was used to form DNA-RNA hybrid duplexes. Subsequently, the DNA parts of the duplexes were cleaved and the miRNA-141 was released. The second step B consists in a signal amplification by enzyme-assisted target recycling. In this respect multi-labeled functionalized fullerene nanoparticles (FC60) thiol-attached to an Au electrode, provides a large surface area with active sites to obtain multiply-enhanced amplified signal. The device sensitivity allowed detection of miRNA-141in the range 0.1 pM and 100 nM, and the lowest LOD of 7.78 fM. Fullerenes were also utilized in the ink of screen printed electrodes to detect rDNA of *Escherichia Coli* [165]. The electrode was treated in a plasma to graft a DNA probe allowing a direct detection of the 16S rDNA extracted from *Escherichia Coli* with reduction of the redox peak of [Co(phen)_3_]^3+/2+^ occurring only when perfect hybridization between 16S rDNA sequence and probe strands occurred. An impedimetric Fetuin-A biosensor was fabricated by modifying a gold electrode activated EDC/NHS poly-hydroxylated fullerene and then coated with PAMAM (G5) [166]. Then anti-Fetuin-A antibodies were attached to the electrode surface enabling detection of Fetuin concentrations in the range between 5 and 400 ng/mL and the lowest LOD of 1.44 ng/mL. Compared to ELISA test the sensor provides linear behavior and fast response time. Another ultrasensitive electrochemical sensor based on fullerenes was fabricated to detect *Mycobacterium tuberculosis* (MTB) [167]. The sensor provided a rapid and efficient detection method for MTB with excellent specificity and sensitivity for MTB antigen in serum samples obtained from patients infected by tuberculosis. Authors produced fullerene C_60_-polyaniline (C_60_-PAn) nanohybrids possessing large surface area, high concentration of active groups and excellent electric performance. The nanoprobes were attached to gold nanoparticles and labeled with signal antigen MPT64 to form the tracer label. The electrochemical response signal is generated by the C_60_-Pan-MPT64 and is further amplified by the electrocatalysis of ascorbic acid through C_60_-Pan. The device showed a linear detection in the range from 0.02 to 1000 pg/mL with a LOD 20 fg/mL for MPT64. More importantly, it also exhibited excellent specificity and sensitivity for MPT64 detection in serum samples of tuberculosis patients, which provided a rapid and efficient detection method for MTB infection.

Fullerenes are also used in glucose sensors. In [168] glucose oxidase (GOD) electrochemistry observed using a glassy carbon electrode modified with GOD-hydroxyl fullerenes. In another work a glucose sensor was produced using a mixed-valence cluster of cobalt(II) hexacyanoferrate and fullerene C_60_-GOD enzyme [169]. The C_60_-GOD enzyme-based glucose sensor showed linear response up to 8 mM glucose with a sensitivity of 5.60 × 10^2^ nA/mM and a 5 s fast response time. Finally, the sensor also showed a detection limit of 1.6 × 10^−6^ M and a high reproducibility. Glucose was also detected using a piezoelectric quartz crystal [170]. Authors used GOD functionalized fullerenes to catalyze the oxidation of glucose. The production of gluconic acid was then detected by a piezoelectric quartz crystal sensor obtaining a good lower limit of detection of 3.9 × 10^−5^ M for glucose in aqueous solutions. Fullerenes were also used to fabricate urea sensors [171]. The urease enzyme was immobilized on C_60_ and subsequently deposited on a screen-printed electrode containing a non-plasticized poly(n-butyl acrylate) membrane entrapped with a hydrogen ionophore. In optimal pH condition, the biosensor led to a linear response in the range from 2.31 × 10^−3^ M to 8.28 × 10^−5^ M. Presence of cations such as Na^+,^ K^+^, Ca^2+^, Mg^2+^ and NH_4_^+^ did not influence the response of the urea biosensor. Sensing modalities analytes and sensor performances are summarized in Table 1.

### 3.6. Carbon Nano-Onion Sensing

As for CNOs, it has been demonstrated that polarization of the electronic states of fullerenes are preserved in CNOs. However, with respect to free fullerenes, the absorption energies of CNO states are significantly red-shifted allowing their use as photosensitizers in nanotechnology applications [178]. Luminescent properties of CNOs are also proposed in [179] as fluorophores. CNOs modified using a coupling the multicomponent Ugi reaction with a complex Pd-mediated cascade to attach a deep blue emitting furo[2,3-c]isoquinolines to CNO surface which renders them soluble.

Since CNO’s physical and chemical properties resemble those of fullerenes, also the sensing applications are similar. CNOs were used to produce a hydrogen sensor thanks to their hydrophobic nature and their non-porous texture [180]. The CNO/C_2_H_6_O sensor displayed a fast decrease in resistance with H_2_ concentration under 10 ppm making CNOs a good active element for the detection this gas. The decreased resistance can be attributed to the change of the CNOs electrical properties in a n-type semiconductor with an increase of the number of electrons in the conduction band. The electrical behavior of CNOs is similar to that found by other authors [181]. They tested the conductivity of CNOs in presence of different gaseous species as nitrogen, oxygen, hydrogen and methane. The gasses acted as electron donors for CNO semi-metallic electronic structure of CNOs leading to a decrease of their conductivity. In another work, palladium nanoparticles were deposited on onion-like mesoporous carbon vesicle to sense hydrazine [182]. Pd nanoparticles act as a catalyzer facilitating the hydrazine oxidation at a more negative potential and delivers higher oxidation current with respect the simple onion-like structure. The sensor provided a linear behavior from 2.0 × 10^−8^ to 7.1 × 10^−5^ M and a low LOD of 14.9 nM for hydrazine. Same authors developed also an amperometric sensor for hydrogen peroxide [182]. A glassy carbon electrode modified with a mesoporous onion-like coating decorated with Pd nanoparticles was developed. The electrode showed a linear enhanced amperometric responses towards hydrogen peroxide in the range from 1.0 × 10^−7^ to 6.1 × 10^−3^ M. In addition, the electrode showed a fast response of 1 s achieving the 95% of the steady-current in presence of hydrogen peroxide. Phosphorus doped CNOs are utilized to fabricate highly sensitive devices for NH_3_ [183]. The sensing mechanism is based on chemisorbed oxygen on the P-doped CNOs resulting in a charge transfer from C atoms to oxygen atoms. These last are adsorption site for NH_3_ molecules which reacting with oxygen atoms form H_2_O molecules and let the CNO surface less conductive. Then a measure of the device resistance can be correlated with the presence of ammonia gas in a range 10 ppb–1 ppm with a LOD of 10 ppb.

Another interesting application of CNOs is the electrochemical sensing of the pH [184]. A glassy carbon electrode was modified with a CNO deposited by electropolymerization. This modification induced a high chemical and electrochemical stability over a of 2–10 pH range with negligible interference of monovalent cations. The reproducibility and the low-cost easy fabrication open interesting perspectives in for the fabrication of miniaturized pH sensor devices. Some of the sensing modalities based on CNOs are summarized in Figure 9. The low toxicity of CNOs and their stability and high conductivity permit their use as sensors of biomolecules as hormones or enzymes. CNOs show high absorption in the UV region while its emission properties are wavelength dependent [185,186]. CNOs can be utilized directly as bio-imaging agent in living microorganisms [186].

There are different possible mechanisms explaining the CNO PL: the electron hole radiative recombination; the quantum confinement effects; the presence of emissive surface traps; dipole emitted centers; the electron–phonons coupling occurring in defects of CNO surfaces [185,187]. However, the need to detect specific molecules requires selectivity which is obtained through CNO surface functionalization [185] which renders the interaction between the analyte and the CNOs univocal. Functionalization was utilized to detect glucose using a “turn-off/turn-on” mechanism [185]. Methylene blue (MB) was adsorbed on the CNO surface via charge transfer and hydrophilic interactions. The charge transfer from CNOs to MB molecules caused the fluorescence to be turned off. CNO emission was recovered by the addition of glucose (0.1 mL, 1.8 × 10^−2^ M). A possible mechanism is the H-bonding interaction between the nitrogen atom of the central ring of the MB molecules and the primary alcoholic group of glucose. This likely weakens the interaction between MB and CNOs leading to a “turn-on” of the fluorescence. The detection limit of this mechanism is 1.3 × 10^−2^ M. Glucose was detected also using an amperometric sensing approach coupling CNOs to CNTs [188] or to Pt nanoparticles [189].

The nanodiamond-derived CNOs sensing performances towards dopamine, epinephrine, and norepinephrine were studied in [191] revealing the high sensitivity, selectivity, and stability of the responses of the CNO-modified electrode. In particular authors found that in a concentration range of 0.1–6μM the oxidation peak current was proportional to the analyte concentration. For all the neurotransmitters the detection limit was found to be 100 nM. In other works CNOs were utilized for electrochemical sensing of dopamine in combination with carbon nanofibers (CNF) and polyacrylonitrile (PAN) [192]. The characterization of the materials indicates that the electron transfer properties decreases from CNOs > CNOs-PAN > CNOs-CNF > PAN. The modified GCE were used as sensors for the dopamine using cyclic voltammetry (CV), square wave voltammetry (SWV), and electrochemical impedance spectroscopy. CNOs and CNOs-CNF gave comparable electrocatalytic activities in terms of sensitivity and LOD (CNOs 1.23 μM and sensitivity of 0.74 μA/μM, and CNOs-CNF 1.42 μM, 0.31 μA/μM). Surface of oxidized CNOs was modified using covalent interaction to link biotin-avidin molecules to fabricate an optical sensor [193]. The sensor was fabricated by layer-by-layer assembly utilized to modify a gold surface used to enable plasmon resonance. In addition, it was shown that also CNOs contribute to the amplification of the analytical signals of the biosensor. CNO based sensors find application also for the recognition of DNA strands. A DNA Sensors for Human Papillomavirus Oncogene Detection with Enhanced Sensitivity was fabricated modifying a glassy carbon electrode with CNOs [190]. These last are then functionalized with diazonium salts to bind streptavidin and then a recognition DNA sequence. This last is utilized to attach a target DNA sequence used to detect the papillomavirus oncogene via direct hybridization. This changes the electrochemical properties of the electrode thus allowing the hybridization to be detected through amperometric measurements. The sensor showed a higher sensitivity (0.91 μA/nM) and a lower limit of detection (0.54 nM) with respect to the unmodified GCE (sensitivity = 0.21 μA/nM and a LOD = 3.9 nM). Table 2 summarizes characteristics of GCO sensors.

## 4. Nanodiamonds

Diamond crystals are pure sp^3^ hybrids leading to a tetrahedral symmetry in which carbon atoms interact through strong covalent bonds. This perfect symmetric arrangement of the four orbitals of the valence electrons results in a structure with a density (3.514 g cm^3^) higher than that of graphite. This highly dense crystalline structure explains why diamond possesses an extraordinary strength, an unpaired resistance to compression, and hardness the highest of all other materials on both the Vickers and Mohs scales. The C-C bond properties also induce high chemical stability of diamond chemically also when in contact with strong acids. Diamond can react with the oxygen at a temperature of ~700 °C, leading to decomposition in CO, CO_2_ [194]. Besides high resistivity, the diamond crystal is characterized by a very prominent phonon mobility corresponding to the highest heat conductivity of 3320 W/m^−1^K^−1^ at room temperature (~five times that of copper) [195].

Concerning electronic properties, the carbon-carbon strong covalent bonds causes diamond to be a wide-bandgap material of ~5.5 eV [196] characterized by a remarkable resistivity from 10^11^ to 10^18^ Ω/m. Correspondingly, diamond possesses a high refracting index varying from 2.465 in the violet to 2.409 in the red [55]. This generates the prismatic colors of gemstones. The absorption of diamond is essentially due to the dopant exoatoms contained in the crystal. Diamond exhibits many different color centers. Nitrogen is the more common, boron, phosphorous, hydrogen, nickel, cobalt, silicon, germanium and sulphur are also frequently found in diamond crystals. Nitrogen defects are present in diamond crystals in a variety of forms called A-, B-, C- N2, N3 centers. In the visible, their characteristic absorption transitions fall at 575, 527, 478, 465, 452, 435, and 423 nm [197,198]. Diamonds are classified using the concentration of nitrogen defects. In *Ia* diamond the nitrogen impurities are ~0.3% (3000 ppm) and includes about 95% of all natural diamonds. In type *Ib* diamonds the nitrogen impurities are up to 0.05% (500 ppm) and are about 0.1% of all natural diamonds. Type *IIa* diamonds are almost or entirely impurity free, colorless and constitute ~1–2% of all natural diamonds. These diamonds possess the highest thermal conductivity. Finally, type *IIb* diamonds have the lowest level of nitrogen impurities but contain significant boron impurities diamonds and amount to ~0.1% of all natural diamonds.

### 4.1. Nanodiamond Synthesis

The diamond classification holds also for nanodiamonds which are obtained with top-down methods as the fragmentation of massive bulk diamonds. This means that the properties of fragmented NDs mirror those of their bulk counterpart in terms of biocompatibility, mechanical, optical, thermal, and electrical properties. Recently, the process of diamond fragmentation was studied to refine the production of high quality diamond powders [199]. Microcrystalline diamonds were compressed at different pressures (0.2–0.8 GPa) at ambient temperature to study the fragmentation process. The results show that with increasing pressure the fragmentation of the crystals proceeds through three stages: (i) fracturing of edges and corners, (ii) cracking of the crystal plane, and (iii) refinement of particle disorder. Experiments show also that increasing the pressure the particles reach a relatively stable size. Reduction of the size causes an increase of the contact area among the diamond particles with relaxation of the stress among the particles. As a consequence, the particle refinement vanishes. The production of macroscopic diamond crystals relies on two kinds of technologies, the HPHT and CVD which enable the production of single or polycrystalline diamonds at reasonable costs, fostering their use in diamond based technologies with outcomes in a variety of ordinary commercial products. CVD processes are utilized also to produce NDs [200,201]. In the first work NDs are obtained via dissociation of ethanol vapor in a microplasma at atmospheric pressure and neutral gas temperatures of 100 °C. Addition of H in the plasma atmosphere, allows etching of the non-diamond phases and passivation of the ND surfaces. The size of the produced NDs is in the range 2–5 nm exhibiting cubic and lonsdaleite crystal structures. In the second work a CVD plasma process in a vertical configuration was utilized to synthesize high quality diamond crystals from nanometric to macroscopic dimensions. A laser induced plasma in liquid ethanol is also utilized to decompose the ethanol molecules and synthesize ND particulate [202]. The authors used a femtosecond laser with pulse repetition rate of 1 KHz and a wavelength of 1030 nm. The laser energy was tuned in the range 360–550 μJ resulting in the formation of atomic C, ionized C and C_2_ clusters which acted as precursors for the formation of the nanodiamond crystals. Another used method for the production of NDs is the detonation process [203]. Three possible detonation processes are commonly utilized: (i) conversion of graphite or a carbon precursor into NDs in presence of catalysts at high pressure (~7 GPa) and temperature (around 2000 °C) [204]; (ii) detonation of carbon precursors in a closed chamber using explosives. The process is based on shock waves leading to pressures in the range 20–100 GPa and temperatures >1700 °C. Pressures and temperatures are high enough to induce a graphite to diamond conversion [205,206]; finally the third process relies on a mixture of 60 wt% TNT (C_6_H_2_(NO_2_)_3_CH_3_) and 40 wt% hexogen (C_3_H_6_N_6_O_6_)) which are detonated in a closed metallic vessel in presence of N_2_, CO_2_ and H_2_O [207]. During the explosion the pressure rises to ~28 GPa and the temperature rises around 3800 K (Jouguet point) corresponding to liquid carbon clusters region. The decrease of the temperature leads to carbon cluster solidification in diamond nanocrystals. These processes lead to the formation of very small nanodiamonds with dimensions in the 1–5 nm and likely surrounded by a graphitic/amorphous carbon shell which is removed using strong acids [208]. Figure 10 shows the nanodiamonds produced by different methods.

The utilization requires NDs be dispersible in a stable colloidal suspension. This involves the ND surface chemistry and the related ζ potential which must fall in the range −30 mV or higher than 30 mV to ensure the stability of the ND dispersion. Surface is a commonly used method to produce NDs with ζ~−30 mV at pH = 7 [211]. However, the surface chemistry of NDs strongly depends on the synthesis process used. Generally application of high temperatures induce graphitization with loss of surface functionalization [212]. HPHT processes induce a surface chemistry mainly formed by mainly hydroxyl groups and few carboxylic groups [213]. Differently, detonation nanodiamonds possess the richer surface chemistry with respect to other carbon nanostructures composed by hydroxyl and carboxylic groups as well as epoxides and lactones [213]. If CVD plasmas are utilized to produce Nds, generally the surface is H terminated. The different ND chemistries require a surface homogenization to obtain a uniform ND surface chemistry. Homogenization is performed either under oxidative or reductive conditions depending on the desired surface chemistry [214].

### 4.2. Nanodiamond Functionalization

Concerning the ND functionalization processes, the modification processes of the diamond surface chemistry are rather consolidated. The graphitic shell may also be removed using a mixture of FeSO_4_ and H_2_O_2_ (Fenton) which readily attack the non-diamond phases terminating the surface with OH groups [215]. Diamond surface hydroxylation may be obtained also via mechanical methods such as milling and ultrasounds in water causing radical reactions resulting in the –(OH) termination [216]. Oxidation of the ND surface can be obtained during the process of ND purification involving highly oxidizing agents such as strong acids, singlet oxygen in NaOH, strong ozone, air treated in the presence of a catalyst which are used to remove the amorphous/graphitic coating of the diamond core [217,218]. Surface oxidation is also performed using supercritical water at 350–450 °C [219] or using ozone [220]. Once the ND surface is oxidized, different routes are applied to obtain the desired functional groups. One route consist in the non-covalent functionalization which can be performed utilizing or electrostatic or hydrophobic interactions [221]. In the first case the electrostatic interaction is utilized to attach to the ND surface polar molecules as DNA sequences, polymeric molecules etc. The second route relies on apolar interactions utilized and based to self-assemble the desired molecules on the ND surface. This chemical functionalization has been widely used in preparation of composites and drug delivery [214,222].

Non-covalent functionalization is an easy process which however suffers from poor reproducibility due to the lack of control on the adsorption process. The covalent functionalization employs carboxylic and hydroxyl groups to graft additional molecules. As an example, after activation with thionyl chloride, carboxylic groups can react with amines [208]. This reaction was utilized to obtain PEGylated NDs [223]. We observe that PEG-chains can be terminated with amino, thiol, and azido groups frequently used to selectively bind a variety of different biomolecules through covalent bonds. Another widely utilized functionalization is the derivatization of the hydroxyl groups via esterification obtained via high temperature –OH carboxylation succinic anhydride at high temperatures [224]. An alternative route for the derivatization of NDs is the use of siloxanes [213]. However, this kind of functionalization offers a scarce control of the siloxane coating thickness on the ND core which also depend on the siloxane precursor used [213]. Siloxane functionalization was successfully utilized for drug delivery, for covalently anchor dyes and receptors [225]. Other kinds of functionalization include the reaction of alkyl chlorides of hydroxylated NDs, the functionalization with alkenes of H-terminated NDs, or the use of diazonium salts [226]. Finally, carboxyl groups on the ND surface can be transformed in azido groups ready for successive click chemistry reactions [227]. Some of the more common functional groups used to change the ND surface chemistry are summarized in Figure 11.

### 4.3. Nanodiamond Sensing

The possibility of synthesizing diamonds at reasonable low costs, have promoted the use of diamond in a variety of industrial applications. An example is the use of diamond to fabricate lenses for high-power high-energy radiations or for optics in harsh environments [229,230]. The high isolating power coupled to the outstanding thermal conductivity make diamond the material of choice for high power electrical devices [231]. Finally the superior diamond hardness and the capability to dissipate heat employed in diamond coatings for processing hard materials [232]. Diamond nanostructures are utilized in a variety of applications such as energy storage, catalysis, electroanalysis, tribology and lubrication, chromatography, and mass spectrometry [207]. In addition, one of the sectors where NDs are widely utilized is biology and medicine due to the extremely high biocompatibility. In fact, it has been shown that NDs do not influence the cell differentiation, growth and proliferation and their metabolic activity [233]. ND sensing exploits mainly its electrochemical and optical properties. With respect to sensing applications great effort was spent over the past years to develop fluorescent tags to localize individual small molecules as drugs, proteins, nucleic acids, as well as study complex biological processes [234,235,236]. Ideal fluorescent tags should possess the following properties: (i) high sensitivity, possibly to detect single molecules, (ii) spatial resolution at the nanoscale, (iii) high absorption coefficient and high emission quantum yield, (iv) absence of blinking and photo bleaching, (v) biocompatibility and (vi) possibility modify the original chemistry for functionalization. In the case of bio-sensing, important is also the excitation and detection fall in the “biological transparency window” thus avoiding simultaneous excitation of endogenous fluorescent molecules of blood constituents, of cofactors, and of water. Most of the fluorescent markers do not possess all of these features. NDs, owing to their unique optical and chemical properties, are proposed to be the best alternative to conventional organic dyes, quantum dots or nanoparticles if optical properties, photostability, sensitivity and biocompatibility are concerned. In particular NDs may couple fluorescence properties with quantum sensing [237] which make then an unique probe for the detection of several physical and biological entities.

As pointed out previously, diamond emission properties are due to defects acting as color centers and emitting at different wavelengths (see Figure 12). Among them, the nitrogen–vacancy (NV) centers constituted to a substitutional nitrogen atom next to a vacancy, is the more diffuse defect. Vacancies can be created by high-energy irradiation with electrons, protons, helium ions. Then annealing at 600–800 °C induce vacancy mobility and trapping by nitrogen atoms always present in diamond [207]. This process results in the generation of two kinds of NV centers: the neutral (NV°) and negatively charged (NV^−^) characterized by emission at 575 and 637 nm respectively.

As it appears from Figure 12, the NV^−^ center is of particular interest because (i) it can be optically pumped and emits in the visible; (ii) its ground state spin value is S = 1 and can be spin-polarized and manipulated using electron paramagnetic resonance; (iii) the NV^−^ is characterized by a long spin coherence time. These properties have been exploited for high-resolution magnetic sensing allowing biomolecule detection [239], fluorescence resonance energy transfer [240] and biomedical imaging [241] quantum information [242], base studies in quantum optics [243].

As for sensing, the ultra high sensitivity of nitrogen vacancy was utilized to detect single protein molecules [244] and reconstruct its structure. At this aim, a sort of “quantum lock-in” was utilized to reject noise when the NV^−^ is coupled to the protein molecule as shown in Figure 13. A microwave pulses delivered to the system with a timing in resonance with the NV^−^ spin precession. Coupling with the external molecule causes a resonance leakage thus enabling the detection of the molecule. The diamond NV^−^ was also used to detect substances in a solution with unprecedented sensitivity. In [245] a spectrometer detects the oscillating magnetic generated by the precession of the analyte nuclear spins weakly coupled to the diamond NV^−^ via magnetic dipole interactions. The NV^−^ centers are optically interrogated using a narrowband dynamical decoupling pulse sequence, leading to an NV^−^ spin-state.

An interesting review on the properties and applications of the diamond NV are reported in [238]. Besides magnetic fields, the NV^−^ is sensitive to the local electric field. Most of the techniques currently utilized to detect individual electric charges are limited to low-temperature methods such as single-electron transistors, single-electron electrostatic force microscopy and scanning tunnelling microscopy. In [246] authors demonstrate the possibility to perform high precision three-dimensional electric-field measurements using a single diamond NV^−^. They reached a sensitivity of 202 ± 6 V cm^−1^Hz^−1/2^ corresponding to the electric field generated by a single elementary charge placed at a ~150 nm from the NV^−^. More recent studies promise to enhance the sensitivity in detecting the electric field produced by a local charge distribution [247] reaching a sensitivity of 150 mV cm^−1^Hz^−1/2^. NV^−^ is also utilized to detect local stresses. Since at high pressure the material properties may greatly change, samples are placed in anvil cells where high pressure can be applied. However, in these experimental conditions the measure of the material’s properties is very complex. In [248] authors were able to monitor the behavior of a diamond sample as a function of the temperature and the pressure applied reaching a sensitivity of {0.023; 0.030; 0.027} GPa/Hz^1/2^ in agreement with the theoretical derived values. In another work authors were able to completely reconstruct the stress tensor elements from a two-dimensional field of view of NV optically detected magnetic resonance spectra [249] with a sensitivity of ~0.1 MPa at 10 mK. Another useful color center of diamond is the silicon vacancy (SiV). Differently form the NV^−^ whose emission yield is ~4%, the SiV coupling with phonons is very wea leading to an emission efficiency of~70% at room temperature. As for NV, also SiV exist in a charged SiV^−^ and neutral state SiV^0^ with correspondent zero phonon lines (ZPL) at 738 nm and 946 nm respectively. NV^−^ are electron paramagnetic resonant (EPR) defects possessing a rather long coherence time allowing a spin coupling with external species [250]. Differently the SiV are not EPR active and their ZPL relaxation time is of the order of 1 ns at room temperature [251]. To the long coherent times of NV spins, the correspondent ZPL is rather broad due to emission in the phonon sidebands. Differently the SiV is narrow (at room temperature < 1 nm) making it a very useful tool for sensing. The limited line-width of the SiV defects results in an almost indistinguishable photons from separate SiV^−^ centers which do not need any additional tuning techniques for the creation of high quality excellent single photon source building block for scalable quantum networks [251].

SiV vacancies can be utilized to perform high precision temperature measurements also in living cells [252]. In this work, authors used the temperature dependent position of the SiV^−^ ZPL to estimate the local temperature. The precision obtained corresponds to Δλ/DT = 0.0124 nm K^−1^ ([6.8 GHz K^−1^) using a bulk diamond sample. Same authors used the ZPL red shift induced of nanodiamonds to measure the temperature. Because of the lower signal-to-noise ratio, the heating was modulated to increase the measurement precision leading to an uncertainty of 521 mK/Hz^1/2^. This precision is important to study the molecular mechanism regarding the expression of temperature-related physiological functions. An interesting review regarding the different temperature probes (including diamond) and the techniques utilized for temperature sensing in living microorganisms may be found in ref. [253]. The SiV vacancy fluorescence lies in the NIR region which with respect to NV, better matches the optical transparency window of tissues thus giving the opportunity to better analyze biological samples. As an example, in [254] SiV^−^ were used as to label neuronal precursor cells obtaining a very high contrast versus the cell-autofluorescence. SiV are bright and highly stable defects which do not suffer from photobleaching as the organic dyes. This opened the possibility to use fluorescent NDs in high resolution microscopy where photostability is highly required. For example SiV defects were utilized to perform high resolution stimulated emission depletion microscopy with a resolution better than 150 nm [255]. In another work fluorescent nanodiamonds (FND) were used in correlative microscopy as dual-contrast probes [256]. Images were collected using both stimulated emission depletion and transmission electron microscopy to visualize cell organelle such as mitochondria. FNDs are currently utilized to follow fast and slow events in cells thanks to their photostability and resistance to photobleaching and photoblinking. In [257] FNDs were used for tracking the protein fate within an organism as for example the low-density lipoproteins (LDLs) and Yolk lipoprotein complexes (YLC) in *Caenorhabditis elegans* worm during 12 h. Confocal microscopy and FNDs were used for long-term labeling and tracking of stem cell division, proliferation, and differentiation [258]. Cell development, fate, and contribution to regenerating tissues were observed for up to 8 days without any photobleaching.

Detonation NDs are also used to follow the fate of drug molecules after delivery. The negative charge of functional groups of the DND surface can interact with the positively charged amino group of doxorubicin, a drug used in cancer therapy. Using the ND fluorescence, the release and the effect of the doxorubicin drug can be studied in vivo in animal models [259]. An excellent review on the use of nanodiamonds for biomedical applications is given in [260]. Finally, NDs can be used as electrochemical sensors. Electrochemical measurements involve the detection of different electrostatic potential of two different electrodes immersed in a solution. The potentials are induced by the charge transfer across the electrode/solution interface which depends on the material constituting the electrode. There are a few reasons why diamond shows better performances with respect to other materials: (i) diamond electrodes display a very low background signal thus allowing improved sensitivity; (ii) diamond electrodes do not show ionizable or redox active groups generally present on the surface of the electrodes and leading to distinct background peaks or a pseudo-capacitive background to the voltammetry measurements; (iii) most of the electrodes tend to show extremely large background currents at high potentials caused by the decomposition of the electrolyte. Diamond electrodes possess a much wider electrochemical window in which redox processes may be analyzed. NDs were utilized a nano-electrochemical sensors for the detection of pyrazinamide one of the most consumed antibiotics for the treatment of tuberculosis [261]. NDs were deposited on a glassy carbon electrode and led linear response in the range from 7.9 × 10^−7^ to 4.9 × 10^−5^ mol L^−1^ with a LOD of 2.2 × 10 ^−7^ mol L^−1^. NDs were also used to detect monophenols and bisphenols pollutants avoiding problems deriving from sensor fouling. In [262] authors were able to overcome this problem using an electrode modified with boron doped nanodiamonds to detect bisphenol-A reaching a low detection limit at 5 nM. A glassy carbon modified with biochar, nanodiamonds and chitosan, was used to detect toxic metals such as cadmium and lead [263]. The sensor showed an electrochemical sensitivities of 0.42 and 5.3 μA/(μmol cm^2^) was found for Cd and Pb respectively. In addition, the sensor showed good stability and reproducibility over a period of 30 days. A good review of the ND applications including electrochemical sensing may be found in [264]. Table 3 summarizes characteristics of ND sensing performances.

## 5. Carbon Quantum Dots

Among the carbon nanostructures, carbon quantum dots (CQDs) raised to the community attention for their versatile synthesis and intriguing properties. CQDs possess a carbon-based skeleton and a large amount of oxygen-containing groups on the surface making them easily dispersible in water [265]. However, the surface chemistry must be carefully adapted to enable CQD fluorescence. Besides surface chemistry also the quantum confinement rising when the CQD size is reduced to nanometric dimensions affects the electronic structure and then the CQD optical properties. As traditional semiconductor quantum dots also CQD display excellent emission properties photobleaching resistance and chemical stability. In addition, CQDs possess also good biocompatibility with low cytotoxicity, and are environmental, and biohazard friendly not always present in other quantum dots.

### 5.1. Carbon Quantum Dot Synthesis

CQDs were discovered by a top-down route consisting in cutting large carbon nanostructures such as graphite, long CNTs or micron-sized graphene patches etc. in smaller dimensions. This exfoliation/decomposition process is performed by acidic oxidation which produces carbon particulate of small dimension and, at the same time, introduces oxygen based functional groups as hydroxyl, carboxyl groups, thus making the particulate easily dispersible in polar solvents. In [266] Chinese ink precursor was oxidized using a mixture of HNO_3_, H_2_SO_4_, and NaClO_3_ acids. The carbon particulate was then hydrothermally treated with dimethylformamide to introduce nitrogen functional groups, sodium hydrosulfide for sulphur functional groups and sodium selenide to graft selenium. Three different kinds of carbon dots were obtained showing tunable PL improved quantum yield and lifetime with respect to conventional CQDs. Low-cost manufacturing of CQDs is performed by electrochemical exfoliation of pure graphite electrodes in different pH electrolytes as H_2_SO_4_, NaCl, NaOH. In [267] high purity graphite rods were exfoliated in 7 mL 0.1 M NaOH aqueous solution. The resulting solution was then treated adding 1 mL 80% hydrazine hydrate and then centrifuged and dialyzed in deionized water to wash the CQDs and adjust the solution pH to 7. The resulting particulate had a dimension in the 5–10 nm range and was characterized by a bright yellow emission with 14% quantum yield. A top-down method is the arc discharge [268] used to vaporize a bulk carbon electrode. During the arc discharge the temperatures reached are as high as 4000 K leading to the formation of a high-energy plasma. Carbon vapors condense on the opposite electrode forming carbon dots. In another experiment, CQDs were serendipitously produced in an arc discharge utilized for CNT production [269]. Fragmentation of CNTs led to the production of three kinds of CQDs with different sizes thus possessing different emission properties in the blue-green, yellow, and orange regions. Laser ablation is another technique utilized to generate carbon gas plasma using a high energy laser pulse in organic solvents [270]. In this work natural graphite flakes suspended in a mixture of ethanol and diethylenetriamine were irradiated with a Q-Switch ND:YAG laser system. The condensation of the laser-induced plasma plume led to the formation of N doped graphene-like quantum dots with size of about 6 nm. The three different chemical forms of nitrogen are generated by the pulsed laser deposition: pyridinic, pyrrolic, and graphitic GQDs. Presence of N doping results in an enhance emission with a quantum yield of ~9.1%. Pulse laser deposition is also used to vaporize carbon glassy particles suspended in polyethylene glycol 200 resulting in the formation of fluorescent CQDs [271]. The average dimension of the CQDs is 3 nm which were utilized for fluorescent imaging of healthy and cancer epithelial human cells. The CQDs showed good emission properties and photostability. Among the bottom-up techniques, the combustion [272] and microwave pyrolysis [273] are other methods to produce CQDs. These techniques ensure a due to the facile synthesis, safe for the environment, and are easy to scale-up and transfer to industry for mass production. The obtained CGDs are generally easily dispersible in water, possess a tunable PL. Interestingly in [274] a Maillard reaction was used to react glucose with a list of different amino acids [274]. The reaction was performed at 125 °C. The polymeric compound was the treated at 275 °C to obtain different kinds of CQDs displaying different optical properties upon the parent amino acid utilized. Reason for this is the different size, shape and composition of the synthesized CQDs leading to different optical properties. Microwaves were also utilized to assist a hydrothermal process. Glucose, sucrose or fructose were used as precursors to produce CQDs with an average diameter as small as 1.65 nm and water-soluble. The precursor was dissolved in water and heated using different microwave power and process duration. The CQDs exhibit deep ultraviolet emission of 4.1 eV. In a typical hydrothermal process, an organic precursor is dissolved in water or organic solvent. The mixture is then put in a Teflon-lined stainless steel autoclave and heated at relatively high temperature to form carbon seeding cores which then grow to form CQDs with dimensions less than 10 nm. The hydrothermal method is very popular because it is simple, it produces an almost monosized distribution of CQDs with different surface functionalizations such as sulphur [275], nitrogen [276], oxygen containing groups [277]. Figure 14 shows SEM, TEM images of CQDs produced by different synthesis methods.

Sulphur, chlorine, phosphor doping and their effect on the CQDs is reported in [278]. Different emission wavelengths are also obtained coordinating the CQDs with metal ions [279]. CQDs were produced polysaccharides to study the effect of Cu^2+^, Sn^2+^, Cd^2+^, Zn^2+^ ions on the CQDs optical properties. Authors found that CQD-Cu^2+^ ions cause an emission quenching while CQD-Sn^2+^, CQD-Cd^2+^, and CQD-Zn^2+^ led to an enhancement of photoluminescence intensity. Quenching is hypothesized to derive from a photoinduced electron transfer involving the Cu d-orbitals and the CQD electronic configuration while in the other case the enhancement should derive from internal charge transfer between the Sn^2+^, Cd^2+^, Zn^2+^ ions and CQDs. Tuning the CQDs optical properties was performed by an appropriate modification of the surface chemistry. In [280], are indicated the possible approaches to redshift the emission from CQDs: (i) increasing the oxidation state of the surface; (ii) introduce nitrogen doping in the in the CQDs core; (iii) solvatochromism; (iv) reacting CQDs with phenylenediamines, naphthalenediamines, or trihydroxybenzene, in harsh conditions. Arginine and ethylenediamine in different proportion can be utilized to produce blue and orange light emitting CQDs as well as white light emission [280]. Finally, same authors were able to obtain important tunable electrochemical properties with the introduction of quinones in the arginine and ethylenediamine reaction mixture.

### 5.2. Carbon Quantum Dot Functionalization

The surface chemistry of the CQDs is determined by the precursor and by the synthesis process selected as previously observed. Good reviews of the CGDs synthesis and properties can be found in [49,280,281,282,283,284].

### 5.3. Carbon Dots Sensing

The most useful and prominent property of the CQDs is their PL. Two main elements affect the optical properties of CQDs: their chemistry and the quantum confinement. CQDs display a typical absorption spectrum with two main bands at around 230 and 350 nm with a tail extending into the visible range. The feature at ~230 nm is ascribed to the π–π* transition C–C bonds in aromatic rings, whereas the shoulder at ~300 nm is assigned to the n–π* deriving from transitions in C=O bonds or other N, O containing functional groups [285]. These last are also responsible of modification of the absorption spectra caused by the different hybridization derivatives. In addition, absorption spectra may also depend on some molecules as those obtained in citric-acid-based synthesis, displaying similar absorption spectra in the 200–400 nm range [286] as reported in Figure 15A. Concerning PL, the emission spectra are formed by a unique broad feature characterized by a large Stokes shift with respect to organic dyes. The position of this feature depends on the excitation wavelength which is ascribed to the presence of different dot sizes, the different surface chemistries, different emissive traps [287]. Figure 15B,C show the dependence of the absorption spectrum and of the PL spectra on the excitation wavelength and on the surface chemistry. Besides chemistry, also the presence of quantum confinement influences the optical absorption and PL properties of the CQDs [286]. PL from carbon dots may derive from the presence of stabilized surface energy traps as a result of the surface passivation which become emissive [288]. In addition, electronic states of the carbon core are involved in the PL through radiative recombination of excitons resulting from the π–π* transition assisted by the quantum confinement [286]. Generally quantum confinement leads to emission wavelength shortening. As the size of the CQDs decreases, the HOMO–LUMO gap increases as experimentally verified in [289] in agreement with theoretical models [290]. These properties are important to tune the emission at the desired wavelength thus enhancing the efficiency of CQDs sensing.

As a matter of fact, the fluorescent emission changes in intensity, wavelength, anisotropy, or lifetime upon interaction with different analytes and upon the analyte concentration [294]. An example is the capability to detect metal ions as Hg, Fe, Cu. In this work a turn-off—turn-on of the fluorescence resonance energy transfer and ratiometric response are utilize to detect the metal ions [295] with detection limits of 10 μM—0.2 nM for Hg, 1 μM—0.58 pM for Cu^2+^, 17.5 μM—2 nM for Fe^3+^. In presence of the ions, an electron, charge or energy transfer occurs resulting in a selective interaction between CQDs and metal ions. The reason of the fluorescence quenching is ascribed to the functional groups on the surface of the CQDs, such as carboxyl groups, hydroxyl groups, aminogroups, etc which can selectively interact with the specific metal ions. As a consequence, the formation of a complex of metal-ion/CQDs the CQD electronic structure changes perturbing the exciton distribution and in particular increasing the non-radiative recombination processes thus leading to quenching. Another mechanism of ion metal detection is related to the change of the absorption properties of the metal-ion/CQD complex. CQD fluorescence quenching through metallic ions is the most utilized method to detect ferric ions. Iron assumes important roles in human physiological processes: active site of the hemoglobin and myoglobin of muscle cells. Its deficiency causes anemia reducing resistance to fatigue, cognitive problems, kidney malfunctioning and a general discomfort. Sensing iron ions is then relevant for the general wellness. The CQD fluorescence quenching induced by metal ions was exploited in [296] to detect Fe^3+^ ions with a LOD of 1 ppm. In this work, Fe^3+^ ions interact with the phenolic hydroxy groups of the CQDs, through electron-transfer between an electron in the excited state of CQD to the d orbital of Fe^3+^ leading to a nonradiative decay. CQDs are also utilized to detect Hg^2+^ which is one of the most poisonous and ubiquitous pollutants for the environment and health. CQDs were labeled with oligodeoxyribonucleotide. Presence of Hg^2+^ ions causes fluorescence quenching and a linear relation between degree of quenching and mercury concentration was found in the range 5–200 nM [297]. Fluorescence quenching was also utilized to detect Cu^2+^ ions. CQDs possessing surface carboxyl, hydroxyl and amine functional groups can coordinate Cu^2+^ ions which cause fluorescence quenching and the LOD of 23 nM [298]. The same mechanism was also utilized to detect Cr^6+^[299], Pb^2+^ [300], Au^3+^ [301] and K^+^ [302]. The Cr^6+^ was detected in a wide concentration range of 0–140 μM with a LOD of 40 nM. In the case of Pb the LOD was as low as 4.5 ppb while for Au^3+^ was 64 nM. Finally, for K^+^ the fluorescent detection had a linear behavior in a wide range of 1–100μM while the LOD was 0.0570 μM. In the case of Ag^+^ interaction with the CQDs led to a fluorescence enhancement [303]. This effect was attributed to the reduction of Ag^+^ ions to Ag^0^ with consequent enhancing of the radiative processes. Fluorescence enhancement occurs also when the absorption bands overlap the emission bands then enabling the metal ion detection [295]. Fluorescence enhancement had a linear behavior in a concentration range of 0–90 μM with a LOD of 320 nM.

Quenching is also used to detect anions. In a typical experiment, first the CQDs are coordinated with a metal ion to quench the fluorescence which is reactivated by adding an anion. Different couples of CQDs-metal ion/anion were tested such as CQDs-Cu^2+^/S^2^ [304]. The turn-on sensor showed good selectivity towards other metal ions and a LOD of 1.72 μM. For CQDs-Cu^2+^/H_2_S [305] sensing was linear with the H_2_S concentration in a range of 5 μM to 100 μM and a LOD of 0.7 μM. Polyethylenimine-capped CQDs were used to detect Cu^2+^/CN^−^ anions [306]. The sensor had a linear range from 2 to 200 μM and a LOD of 0.65 μM. A sensor based on carboxylated CQDs detected the Eu^3+^/PO_4_^3−^ couple with a LOD of 5.1 × 10^−8^ mol/L and a linear range from 4.0 × 10^−7^–1.5 × 10^−5^ mol/L [307]. Finally, a colorimetric fluorescence “turn-on” sensor was obtained using amino functionalized CQDs to detect thiocyanate ions [308]. The change of color was observed by eye at a concentration of 1 μM and a LOD of 0.36 μM using fluorescence spectroscopy.

Another mechanism exploited for sensing is the photo-induced electron transfer. In this process, exploiting the donor-acceptor interactions an electro-deficient group binds with an electron-rich fluorophores. In the photo-induced electron transfer, the excited-state of fluorophores is likely to donate an electron to the ground-state of the analyte explosive compounds. This coupling leads to a non-radiative de-excitation of the fluorophore which lowers fluorescence intensity. This mechanism was utilized in combination with CQDs to detect explosives [309]. Moreover, the measure of the fluorescence lifetime is an indication of the coupling of CQDs to the external analyte. In [310] water soluble CQDs with optical band gap of 3.4 eV were synthesized for bio-imaging. CQDs were used as fluorescent probes which were easily internalized in cells. Cell imaging was obtained by a two-photon excited fluorescence lifetime imaging microscopy. Using a 750 nm femtosecond laser excitation was utilized to get long fluorescence lifetime, high-contrast resolution images which coupled to the low CQD cytotoxicity make them interesting for biological applications. As for this kind of applications, CQDs are utilized as electrochemical sensors to detect important neurotransmitters as tyrosine, epinephrine, norepinephrine, acetylcholine, serotonin and in particular dopamine. Here we will give only some examples while a more extended description may be find in [282]. A glassy carbon electrode was modified with CQDs to sense dopamine [311]. The sensor provided an oxidation peak current linear with the dopamine concentration in the range from 0.1 μM to 30.0 μM with the LOD as 11.2 nM. Another dopamine sensor was fabricated using reduced graphene oxide-CQDs (rGO-CQDs) [312]. rGO-CQDs showed better electrochemical responses when detecting dopamine if compared to simple glassy carbon electrode or glassy carbon electrodes modified with CQDs or graphene oxide-CQDs in terms of linearity of the peak current and sensitivity which ranges between 0 and 10^−3^ M and a low LOD of 1 nM. In another study, CQDs were used to detect acetylcholine [313]. A layered flower-like formed by NiAl and its hydroxide displaying positive surface charge was decorated with negatively charged CQDs. The structure exhibited enhanced electroconductivity and electrocatalytic performance for acetylcholine oxidation. The sensor showed linear response in the concentration range from 5–6885 μM with a low LOD of 1.7 μM. An electrochemical sensor based on N doped graphitic CQDs was synthesized to detect epinephrine [314]. The tests were carried out using cyclic voltammetry in presence of in the presence of 100.0 mM pyrrole and of 25.0 mM epinephrine. The electrode was highly performant with a broad linear range from 1.0 pM to 1.0 nM and a LOD of 3 × 10^−13^ M. CQDs are extensively used to detect small molecules of rather different nature including biomolecules as glucose involved in diabetes pathology, through fluorescence quenching of boronic acid coordinated CQDS [315]. The sensor showed a linear behavior in the range from 0.1 mM to10 mM with a LOD of 5.0 μM. Sensing glucose is important because its unbalance in metabolism causes diabetes. Thiols as glutathione and cysteine are often involved in cellular metabolism and detoxification, are detected through Cu, Au turn off fluorescence, while S via a turn-on process induced by complexation of ion metals with thiols [316] or Ag for detecting cysteine following the same principle [317]. CQDs are also utilized to detect drug molecules [318] through fluorescence enhancement, vitamin B_12_ (linear behavior from 1 to 12 μg/mL and LOD = 0.1 μg/mL) [319] or proteins as hemoglobin (linear behavior from 0.05 to 250 nM and LOD = 30 pM) [320] or DNA. In this last case, a turn-off—turn-on process based on methylene blue molecules is used to sense DNA sequences [321]. The surface of CQDs was functionalized with methylene blue which quenches the CQDs fluoresce via an electron-transfer process. DNA can bind methylene blue molecules which are removed from the CQDs thus restoring the fluorescence. The sensor had a linear response in the range from 3.0 μM to 80 μM/L with a detection limit of 1.0 μM/L. CQDs are also used to discover presence of contaminants [322]. CQDs were recently used to sense trichlorophenol in the red wine and water. The sensing is based on the interaction of trichlorophenol with the aromatic rings of the CQD core leading to an enhancing of the emission from CQDs [323]. The sensor had a linear detection in the concentration range 0.1–20 μg/mL with a LOD of 0.07 μg/mL. More information can also be found in [324]. Finally, because of the biocompatibility, one of the more prominent uses of the CQDs is the fluorescent tag for bioimaging. Besides biocompatibility, CQDs possess outstanding properties as the possibility to tailor their surface chemistry/functionality, their superior photostability and high brightness, and spontaneous penetration capabilities which explain the excellent potential of CQDs as probes for the study of biological systems, and for imaging-guided biomedical applications. Over the past decade, applications of CDs in bioimaging were achieved in two broad categories: in vitro imaging of cells and cell organelles, and in vivo applications mainly devoted to drug delivery and visualization of CQDs biodistribution [265,324]. In in vitro cell imaging CQDs are used to enlighten specific structural elements or biological parameters. As an example, CQDs functionalized with both N and Cl based functional groups were utilized to detect the intracellular pH in HeLa cells [325]. As a matter of fact, such CQDs exhibit a fluorescence which decreases with increasing the pH of the medium. Authors found that the fluorescence lifetime in the pH range of 2.6–8.6. In another work CQDs were used to sense both pH and the cytochrome C released from mitochondria and leading to cell death. CQDs rich of hydroxyl and amine groups exhibited an average particle size of ca. 3.88 nm and a strong pH-sensitive fluorescent emission which is quenched at increasing pH values. In addition, dexamethasone is known to induce the release of cytochrome C into the interior of HeLa cells. It was found that the fluorescence intensity of doped CQDs was strictly correlated to the cytochrome C concentration [326]. The fluorescence turn-off allowed detection of cytochrome C within in a range of 10–500 mg/L with a detection limit of 3.6 mg/L. In another study [327], an ultrasensitive aptamer–CQDs was synthesized for sensing CA125 which is used in the clinical diagnosis as a marker of several cancers and, presently, is the best serum-based tumor marker for ovarian cancer. Gold nanoparticles are modified adding a PAMAM-Dendrimers on the surface. This dendrimer contains NH_2_ active sites to attach the CA125 antibody. These nanoparticles are able to capture the CA125 cancer marker and at the same time the Au is able to quench the aptamer-CQDs fluorescence. By measuring of fluorescence resonance energy transfer (FRET) signals between CDs and AuNPs as nanoquenchers, the decreasing fluorescence can be correlated to CA125 concentration. The immunosensor exhibited an extremely low calculated LOD of 0.5 fg/mL with a wide linear range from 1.0 fg/mL to 1.0 ng/mL of CA 125. CQDs were utilized for stem cell imaging which is important in regenerative medicine to study the progression of the cell growth and modification in specific tissues such as bone, skin, and neural. The small size and biocompatibility of the CQDs ensures them to be easily internalized in stem cells via endocytosis in a concentration- and time-dependent way. Thanks to the excellent photostability, the CQDs were used to label three different kinds of stem cells without evidence of any adverse or toxic effect providing the option for a long-term imaging [267]. Similarly CQDs were developed as a fluorescent neural tracer based on cholera toxin B (CTB-CQDs), which could be internalized and transported by neurons in the peripheral nervous system of rats [328].

Results suggested that CTB–CQDs could bind with high affinity to monoganglioside GM1 and within four days were retrogradely transported from axonal terminals to neuronal soma. CQDs can serve to stain specific part of the cells. Thanks to their small sizes, surface positive charges and dopamine-mimicking properties nitrogen doped CQDs were successfully used to image the cell nucleus of four different types of cancer model cells, including rat PC12, A549, HepG2 and MD-MBA-231 [329]. CQDs can also be utilized to stain parts of cells and organelle as mitochondria [330], lysosomes [331]. Figure 16 schematize the surface functionalization and the optical properties CTB–CQDs as staining agent in in vitro and in vivo experiments.

Finally, in vivo studies have benefited from using CQDs to study if they induce chronic toxicity, their stability, their biodistribution and possible accumulation in organs/tissues. Near infrared fluorescent CQDs were developed to study the biodistribution in mice. Results put forward that the O-functionalized CQDs were firstly accumulated in the reticuloendothelial system and kidneys but subsequently they were gradually cleared via renal and fecal pathways with low toxic effects [332]. In another study CQDs were utilized to image glioma tissues taking advantage of the enhanced permeation retention [333]. The hydrophilic character of the CQDs is supposed to be one of the reasons for the major accumulation in the glioma. Moreover, the average size of the CQDs in the range of 5–10 nm permits a good infiltration of the glioma tissue being characterized by a pore size in the blood tumor barrier of ~12 nm. Experiments show also an accumulation of the CQDs in liver and kidney indicating that CQDs are mostly eliminated by these two organs. Concerning accumulation and clearance, other authors studied the CQDs biodistribution and uptake in rats and tumor mice [334]. They studied the different biodistribution of silicon- and carbon-based dots through radioelement labelling and dynamic experiments of positron emission tomography. Results indicate a rapid renal clearance from the in vivo systems for both variants of the nanoparticles. However, marked differences in the biodistribution and pharmacokinetic properties of Si- and C-dots were observed. Depending on the surface charge, positive zeta potentials and hydrophilicity induced by the surface functionalities, different accumulation in liver and intestine are obtained. An example of in vitro and in vivo use of the CQDs is represented in Figure 17A,B where CQDs could distribute only in the cytoplasm of PC12 cells, without accumulating on the cell membrane or in the cell nucleus. In Figure 17B CQDs are colocalized with lysosomes while in Figure 17C in an in vivo experiment, a strong fluorescence signal is obtained after injection of the CQDs on the back of a nude mouse.

CQDs may also be utilized to image the drug delivery. A nanocarrier based on cisplatin(IV) prodrug-loaded and charge-convertible CQDs (CQDs-Pt(IV)) was designed for imaging-guided drug delivery [335]. An anionic polymer with dimethylmaleic acid used to coat the CQDs-Pt(IV), displayed a charge conversion in mildly acidic tumor (pH = 6.8) extracellular microenvironment. In addition, the positive charges of the nanocarried facilitated coupling with the negative cell membranes and the internalization. In vivo experiments showed an high tumor-inhibition efficacy and low side effects of the CQDs-Pt(IV) nanocarriers. In another study [336] a turn-on fluorescent CQDs was used as a theranostic nanoprobe. The dots were formed by assembling a polyethylenimine modified carbon dot with hyaluronic acid conjugated with doxorubicin (P-CQDs/HA-Dox). The functionalized P-CQDs/HA-Dox served as multipurpose probes for hyaluronidase detection, self-targeted imaging and drug delivery. Results showed that the nanoprobes could specifically target CD44 receptors overexpressed on many cancer cells thus enhancing the internalization. Once penetrated in the cells, hyaluronidase is activated leading to fragmentation of the HA-Dox, causing the release of Doxorubicin and the recovery of the P-CQDs fluorescence. The efficient release of Doxorubicin induced apoptosis of HeLa cells, as confirmed by MTT assay with effective treatment of cancer. Solid tumor tissues are characterized by hypoxia which remarkably reduces the efficiency of the photodynamic therapy. To solve this problem, carbon nitride was utilized to dope CQDs to induce water splitting under irradiation red light because carbon nitride increase the absorption in the red spectral region [337]. Then protoporphyrin photosensitizer, a tumor-targeting Arg-Gly-Asp motif and polyethylene glycol as a linker molecule were used to assemble PC_CN_CQDs. Results of in vitro studies show that PC_CN_CQDs succeed in increasing the O_2_ concentration in the cell compartments and, under light irradiation, the production of reactive oxygen species in both hypoxic and normoxic environments. The use of PC_CN_CQDs in cell viability assays showed their ability in reversing the hypoxia-triggered PDT resistance thus leading to a good growth inhibition of cancer cells.

In vitro study showed that PCCN, thus obtained, could increase the intracellular O_2_ concentration and improve the reactive oxygen species generation in both hypoxic and normoxic environments upon light irradiation. Cell viability assay demonstrated that PCCN fully reversed the hypoxia-triggered PDT resistance, presenting a satisfactory growth inhibition of cancer cells in an O_2_ concentration of 1%. In vivo experiments also indicated that PCCN had superior ability to overcome tumor hypoxia. The use of water splitting materials exhibited great potential to improve the intratumoral oxygen level and ultimately reverse the hypoxia-triggered PDT resistance and tumor metastasis. CQDs were also utilized to cross the blood brain barrier (BBB) and accumulate in the brain. This property was utilized to enhance the efficiency of drug delivery without harming the BBB integrity using CQDs [338]. Amphiphilic yellow-emissive, 3 nm sized CQDs were synthesized using citric acid and o-phenylenediamine. This resulted in the presence of both primary amine and carboxyl groups enabling the bioconjugation with small drug molecules. Interestingly, the amphiphilicity and the BBB penetration ability were preserved after coating CQDs with different hydrophilic molecules. Experiments showed the ability of CQDs to enter cells and inhibit the overexpression of human amyloid precursor protein and β-amyloid which is a major factor responsible for the Alzheimer disease. In another study, neuroactive properties of fluorescent C-dots obtained from β-alanine were synthesized to study their effects on the key characteristics of GABA- and glutamatergic neurotransmission in isolated rat brain nerve terminals [339]. Authors found that CQDs act in dose-dependent manner: (i) decreasing the exocytotic release of [3H] GABA and L-[14C] glutamate; (ii) reducing the acidification of synaptic vesicles; (iii) reducing the initial velocity of Na^+^-dependent transporter mediated uptake of [3H] GABA and L-[14C] glutamate, and (iv) increasing the ambient level of nerve neurotransmitters without significant changes of the potential of the plasma membrane of nerve terminals. Fluorescent CQDs with their neuromodulatory properties pave their potential usage for labeling/imaging key processes in nerve terminals with interesting theranostic perspectives. Table 4 summarizes characteristics of CQDs sensors.

## 6. Carbon Nanotubes

Carbon Nanotubes (CNTs) is a one-dimensional form of carbon possessing has a perfect hollow cylindrical shape. CNTs can be considered as a graphene sheet (an hexagonal network of carbon atoms) rolled up along certain directions corresponding to the CNT chirality. Rolling up a single graphene layer leads to a single walled SWCNT while rolling up multiple graphene sheets results in a multiwalled CTN (MWCNT). Similarity of CNT to graphene regards not only their geometrical structure but also their electronic properties. As in graphene, also in CNTs the carbon atoms are in sp^2^ hybrids forming three bonds along directions separated by 120° with three neighboring carbon atoms. This structural configuration leads to the generation of hexagonal rings and to a honeycomb lattice. CNTs and graphene share many interesting properties. The strong carbon-carbon bonds are responsible for the high mechanical strength of the SWCNTs have strong possessing a Young’s modulus value range from 320 to 1470 GPa and breaking forces ranging from 13 to 52 GPa [340]. Concerning the electrical properties, CNTs display semiconducting or metallic character. These differences are generated by the direction selected for rolling up the graphene sheet namely the chirality [341]. This direction is defined by two indices (m, n) as schematized in Figure 18. If the direction corresponds to the CNT main axis (m = i, n = 0) i = 1, 2, 3…then a zig-zag metallic CNT is obtained, for (m = i, n = i) an armchair CNT is obtained while (m = 1, n = j) i ≠ j, i,j = 1, 2, 3… a chiral CNT is formed [342].

It was found that when (m-n = 3i i = 1, 2, 3…) the CNT is metallic while for (m-n = 3i ± 1 i = 1, 2, 3…) the CNT behaves as a semiconductor. The electrical conductivity along the main axis of metallic CNTs can be as high as 2 × 10^7^ S m^−1^ [343], while for MWCNTs the electrical conductivity is about 2 × 10^5^ S m^−1^ [344]. Besides conductivity, CNTs possess also a good thermal conductivity which for individual SWCNT was reported to be 3500 W m^−1^ K^−1^ [345], higher than that of bulk graphite (about 2000 W m^−1^ K^−1^), while for individual MWCNT the thermal conductivity is ~3000 W m^−1^ K^−1^ [346]. SWCNTs possess a rather high theoretical surface area which in average is ~1315 m^2^ g^−1^ [347] (about one half that of graphene) which however depends on the quality of SWCNTs with an highest value of 1587 m^2^ g^−1^ [348]. The one-dimensional tubular morphology of CNTs also bring them unique properties. CNTs have super high length-to-diameter aspect ratio: the diameter measures typically 0.4–2.5 nm while the length can be as high as 20–1000 nm for SWCNTs, while for MWCNTs the diameter ranges between 1.4–100 nm and the length between 1–500 μm [349]. Due to the tubular shape, CNTs possess a certain internal volume which can be used to house specific functionalities. Furthermore, the presence of curvature makes CNTs more reactive than pristine graphene and chemical functionalization of their surface may be done more easily [350].

### 6.1. Carbon Nanotube Synthesis

The broad interest in CNTs is a direct consequence of the research around fullerenes. Sumio Iijima studying the production of fullerenes using an arc evaporation system, serendipitously discovered a new for of carbon consisting in multi walls filaments of carbon with a diameter from 3 to 30 nm invariably closed at both ends [351]. From that moment, a large number of alternative methods have been developed for the CNTs synthesis and it became clear that two kinds of CNT could be produced: single or multi walled CNTs. As original CNT synthesis technique, the arc discharge method is well established and broadly utilized. The arc discharge apparatus consists in two highly pure graphite electrodes which are faced at a distance of 2–3 mm. One of the electrodes contains also a certain amount of metal catalysts as Fe, Co, Ni, or Mo [352]. In the reaction chamber a DC arc discharge is then produced in a He atmosphere a voltage between the electrodes. As the discharge is triggered, the graphite and metal catalyst evaporate and condense onto the cathode or on the walls of the reactor forming carbon soot containing both SWCNTs and MWCNTs. Interestingly, the selection of the metallic catalyst and of the inert gas in the reactor chamber determinates whether the resultant CNTs are SWCNTs or MWCNTs [353]. The reactor atmosphere, pressure and arc current are important parameter for the process to control the yield and quality of CNTs [354]. It was shown that the final morphology of CNTs is strongly dependent on the different atmospheres used. High arc discharge currents in a He + CH_4_ atmosphere at high pressure leads to thick nanotubes decorated with carbon nanoparticles [355]. 

In recent arc discharge processes the production of MWCNTs is performed without catalysts. On method is the arc-discharge under He, ethanol, acetone and hexane atmosphere at various pressures (from 150 to 500 Torr) [44,356]. Another possibility to synthesize MWCNTs is the arc discharge in an NH_3_ atmosphere at 0.02 MPa [357]. Authors showed that there is not significant difference in the structure and shape of the obtained CNTs obtained with similar processes in He and H_2_ atmospheres. Pulsed arch discharge in air may be also utilized to produce MWCNTs on a Ni substrate as shown in [358]. More information can be found in [352].

SWCNTs are produced with arc discharges using an H/Ar atmosphere and anodes made of graphite and a metal, such as Ni, Fe, Co, Pd, Ag, Pt, etc. or mixtures of Co, Fe, Ni with other elements like Co–Ni, Fe–Ni, Fe–No, Co–Cu, Ni–Cu, Ni–Ti [352]. The selection of the metal catalyst will influences the SWCNT production yield. The generation of SWCNTs is due to the co-evaporation of Co and graphite in an arc discharge leading to the formation of single atom wall nanotubes with a diameter of ~1.2 nm [359,360]. Besides Ni, one of the most utilized catalysts for SWNTs synthesis, the role of Ni, Pd, and Pt in the formation of carbon clusters was studied in [361]. More information can be found in [352,362]. The need to produce CNTs in a more controlled way led to the development of alternative synthesis methods as the laser ablation. Compared to other techniques, laser ablation is superior to produce high quality SWCNTs. Generally Nd:YAG and CO_2_ lasers are used to vaporize a graphite target containing Ni or Co as catalyzer [363]. In a typical process, the synthesis is performed in a quartz tube, the graphite target placed at the center and all the system is heated in a tubular furnace at ~1000 °C. The laser beam is focused onto the target which causes the evaporation of carbon and metallic catalyst in an inert gas flow. Vapors condensate on a cooled collector along the downstream producing CNTs with impurities [364]. This process is cost consumable since it uses high energy power to vaporize the target, but this process has a high yield and produces primarily SWCNTs. In [365] the production of CNTs by laser ablation is reviewed describing the growth mechanisms and options to make the synthesis more efficient. Other sources of information are [352,362].

Finally, another popular method for the synthesis of CNTs is the chemical vapor deposition (CVD). In a CVD process, hydrocarbon precursors as methane, acetylene, ethane, ethylene or alcohols… [352,366], are decomposed at high temperature in presence of a metallic catalyst, e.g., Ni, Co, Fe [366]. When the reaction gas passing through the flow furnace at high temperature about 1000 °C, hydrocarbon molecules decomposed into active carbon species on the catalyst surface and diffuse into the metal catalyst. Two different processes can then occur: in the tip-growth, the catalyst–substrate interaction is weak Figure 19A. The decomposed hydrocarbons diffuse down through the metal and the CNT grows between catalyst and substrate. In the opposite case, when the catalyst–substrate interaction is strong a base growth process take place. Again the decomposed hydrocarbons dissolve on the catalyst surface but now the precipitation is forced to emerge out from the metal’s apex with sp^2^ structure. Both the processes end when the catalyzer particles are completely covered with excess of carbon.

The characteristics of CNTs produced through CVD method depend on the hydrocarbon pressure and concentration, the nature of the catalyzer, the time of the reaction and temperature [362,366]. To lower the reaction temperature of the CVD process, plasma-enhanced CVD (PECVD) has been developed [367]. Besides lower temperature, advantages claimed include the ability to grow individual, free-standing, vertical SWCNTs, MWCNT and also carbon nanofibers on delicate substrates. 

Lower cost and mass production can be obtained using the flame synthesis of CNTs. This method is capable of a controlled production of CNTs on the selected substrates by using the appropriate catalyst, the correct temperature and the adequate source of carbon. In the flame process, the catalyst is introduced in the flame where it condenses in spherical nanoparticles. Ni, Co, stainless steal and Fe are generally utilized as catalyst in the flame process [368]. As for the temperature, variation of the flame parameters/patterns strongly influences the final product. Several flame patterns, including premixed, partially premixed, and inverse diffusion flames, have been used for the production of CNTs and nanofibers [369]. Finally concerning the source of carbon, different fuels as methane, ethanol, ethylene, methylene acetylene, propane were investigated to check the effect on the production of SWCNT, MWCNT and their morphology. Detailed information can be found in [368]. In Figure 20 are reported TEM images of CNTs synthesized by different techniques.

Most of the techniques used to synthesize CNTs, generate soot as a byproduct containing CNTs and other materials as amorphous carbon, fullerenes, nanocrystalline graphite, and metals introduced as catalysts for the synthesis. Freshly prepared CNTs need purification to separate them from the other form of carbon and from the metal catalysts. A variety of different purification methods have been developed for this purpose. However, common to all purification procedures are the following main steps: (i) removal of the large graphitic particles; filtration of CNTs and consequent resuspension in appropriate solvents; (ii) elimination of the metallic components via dissolution; (iii) and remove the amorphous carbon clusters and fullerenes, microfiltrations and chromatography for size selection [353]. To accomplish these tasks, chemical and physical purification processes can be used [373,374]. The chemical methods involve of concentrated acids or strong oxidants or treatments at high temperature generally in air or oxygen atmosphere. These treatments are used to accomplish a selective etching since they attack more easily amorphous carbon and carbon particles with respect to CNTs. However, both these treatments lead to consistent damage of the CNTs. Physical processes consist in filtration with membranes characterized by small pore size enabling separation of the CNTs from impurities and also to fractionating the CNTs by length. High-energy ultrasounds in the presence of the suitable solvents are used to get rid of the amorphous impurities [375]. Finally, centrifugation is applied to obtain the separation of particulate with different mass [376]. Low-speed centrifugation (2000× *g*) is applied to remove amorphous carbon and leave SWCNTs and CNPs. High-speed centrifugation (20,000× *g*) forces the CNPs to sediment while the SWCNTs remain suspended in aqueous media. Sedimentation and suspension depend not only on the mass but also on the different suspension stabilities e.g., surface charge of the particulate induced by functional groups [377]. 

### 6.2. Carbon Nanotube Functionalization

Since their discovery, several well consolidate routes to functionalize the CNTs were developed. Because of the graphitic nature, of the walls of pristine CNTs display a hydrophobic nature. As a consequence, van der Waals forces cause aggregation into bundles of freshly prepared CNTs. Therefore, it is difficult to separate CNTs and make a stable dispersion in water and in most solvents. This problem may be solved by functionalizing the CNT surface. Functionalization processes of CNTs were developed in the nineties and can be distinguished in covalent and non-covalent functionalization. Covalent functionalization consists in the formation of covalent bonds between a functional group and a carbon atom of the CNT. The covalent functionalization process involves several chemical reactions resulting in covalent chemical bonds between CNTs and the functional entities. They can be divided into two categories: (i) oxidation and end/defects functionalization, (ii) side wall covalent functionalization. An The defect-functionalization of CNTs proceeds by attacking the nanotubes wall defects by strong agents leaving holes functionalized with oxygenated functional groups [378]. Examples are depicted in Figure 21A,B. 

Generally, oxidation is performed using strong acids as HNO_3_, H_2_SO_4_, or their mixture [380] and increasing the reaction temperature [381], or strong oxidant as KMnO_4_ [382] or using ozone [383] or with reactive oxygen plasma [384]. These treatments induce the formation of acidic sites on the CNTs which are subsequently used for attaching other molecules. Oxidation of the CNTs generate oxygen containing functional groups such as carboxylic acid, ketone, alcohol and ester groups, that can be utilized to attach many other different types of chemical moieties [385]. These functional groups enable a rich chemistry allowing attachment of other molecules such as esterification [386], thiolation [387], silanation [388], polymer grafting [389], and some biomolecules [390] see Figure 21B. 

In the direct covalent sidewall functionalization the formation of the covalent bond induces a change of the carbon hybridization from sp^2^ to sp^3^ and a simultaneous loss of p-conjugation typical of the aromatic rings of graphene. This process can be made by reaction with some molecules of a high chemical reactivity of atoms at the ends and at the defects of the CNTs. One of the main advantages of the CNTs functionalization is their dispersibility in solvents thanks to the polar or non-polar groups grafted on the surface. Examples of sidewall functionalization are depicted in Figure 22A,B.

Fluorination of CNTs has become popular because providing substitution sites for additional functionalization [392] enabling replacements of the fluorine atoms by hydroxyl, amino, and alkyl groups [393]. Other possible chemical routes include the Diels-Alder cycloaddition [394], the carbene and nitrene addition [395], chlorination, bromination [396], azomethineylides [397], hydrogenation [398].

However, although covalent functionalization provides a rich list of possible routes to graft the desired molecules, there are some drawbacks. Functionalization accomplished though strong covalent bonds results in the rupture of the aromatic rings of the CNT graphitic lattice. This impacts on both the electrical and the mechanical properties of the CNTs. The disruption of the surface conjugated π network introduces defects which can behave as rupture sites while each covalent bond act as a scatter point limiting the electrical conductivity [399]. Therefore, many efforts have been made to find convenient solutions to limit the CNT damage introduced by the functionalization. Preservation of CNT properties is achieved using the non-covalent functionalization. One of the main advantages of non-covalent functionalization is that it does not modify the carbon hybridization thus maintaining the hexagonal lattice with aromatic graphitic rings.

This preserves the structural and electrical properties of the CNTs because the non covalent functionalization is based on supramolecular complexation made by hydrogen bonds, adsorption based on van der Waals forces, by electrostatic forces or π-stacking interactions [400]. An example of π-stacking interactionsis shown in Figure 23 to attach bio-molecules to the CNT sidewalls.

A side effect of the non-covalent functionalization is the lower stability caused by interactions between the grafted molecule and the CNTs which might be weak [403]. Adsorption of molecules including biomolecules is widely utilized to functionalize CNTs [400,402], some examples are depicted in Figure 23B. Non-covalent amino functionalized CNTs was obtained by adsorption of H_2_N-CH_2_CH_2_O-Na to MWCNTs walls [404]. The functionalization preserves the electrical, the thermal and mechanical properties thus making the functionalized MWCNTs-NH_2_ promising filler for developing electrical conductive MWCNT/epoxy composites. Van der Waals forces are utilized to decorate CNTs with polymeric molecules. CNTs coated with polypyrrole, poly(methylene blue), poly-(neutral red), poly(acrylic acid) and poly(3-methylthiophene) were used to fabricate sensors for the detection of dopamine [405,406,407,408,409]. Phenols were detected using poly(urea-formaldehyde) functionalized CNTs [410]. The interaction by π-π stacking is very versatile and the list of aromatic compounds that are employed is very rich [400,411,412]. Non-covalent functionalization of CNTs include also the adsorption of polymeric molecules [413], and biomolecules [414]. Some of the possible non-covalent functionalization processes of the CNTs are schematized in Figure 24. Functionalization of CNTs is covered in several review articles as [385,391,400,402,415].

### 6.3. Carbon Nanotube Sensing

The multiple and unique physical, mechanical and chemical properties of the CNTs make them suitable for a plethora of different applications. ranging from Their applications include but are not limited to electronics [416,417], energy, [418], environment [419], atomic force microscope [420] and nanomedicine [421], chemical and biochemical sensing [422,423,424]. Conducting and semiconducting properties of CNTs, the possibility to provide them with recognition elements and to organize them in nanowire networks make them the material of choice for sensing applications. In electrochemical CNT based sensors, sensing is based on charge transfer induced by the analyte that will change the conductivity of the CNTs modulating the concentration of the charge carriers. In normal conditions, CNTs are p-doped because of physisorption of oxygen molecules on their surfaces. As a consequence, Thus, further interaction with p-dopant molecules will increase hole concentration and increase the conductivity while while n-type dopants will increase the resistivity [425,426,427]. An excellent review regarding the use of the electronic properties of CNT for sensing application can be found in [428]. CNT conductivity may decrease also because of a reduction of the charge carrier mobility caused by charge carrier trapping or by scattering [429]. These mechanisms were utilized to detect NO_2_ via formation of nitro and nitrite groups [430]. 

A wide range of gases can be detected using CNT and some examples will be given hereafter. Ammonia vapors are toxic and can cause severe irritation of the respiratory system and pulmonary edema at high concentrations. Detection is based on the ability of NH_3_ molecules to adsorb on the CNTs walls without any need of particular functionalization changing their electric properties. The sensitivity obtained was as low as 20 ppb with a LOD of 3 ppb [431] and ascribable to the scrupulous preparation of the CNTs and of the sensing electrode. Other studies showed an enhancement of the sensor response to NH_3_ with the defect level [432]. Deposition of SWCNTs on different kinds of paper were used to fabricate NH_3_ sensors showing a large dynamic range from 0.5 to 5000 ppm [433]. All the sensors showed a linear resistive response in the range 0.5 to 10 ppm while the LOD varied upon the kind of paper used as a substrate form 0.36 to 2.7 ppm. In another work, poly(4-vinylpyridine)-CNTs functionalized with Pd nanoparticles showed an excellent sensitivity to thioethhers with a LOD of 0.1 ppm [434]. NO_2_ is another gas detected using a SWCNT dispersions onto interdigitated electrodes with a linear response in the range 6 and 100 ppm and a LOD of 44 ppb [435]. A higher sensitivity to NO_2_ was obtained in using a PECVD reactor to deposit a thin film of MWCNTs on Pt electrodes with a dynamic response from 10 to 100 ppb and a LOD of 10 ppb [436]. NH_3_ and NO_2_ sensors were also fabricated coupling CNTs with metal oxides [437,438]. As an exaple SnO_2_ was coupled to SWCNTs to detect NH_3_ reaching a LOD of 1 ppm for NH_3_ and of 20 ppb for O_3_ [439]. Other metals used to enhance the sensitivity of the CNT electrodes to CO, CO_2_, NH_3_, CH_4_ and NO_2_ are Co, Cu, Pb, Pd, Ni, Pt, Ru, Ag [440,441]. Metals are also used to detect H_2_ and CH_4_. Pt [442] and Pd [443] metals are used to decorate HGCNTs for the detection of H_2_. Pd decorated SWCNTs are also used to detect CH_4_ with a dynamic range of 6–100 ppm [444]. CH_4_ sensors were fabricated also using semiconducting metal oxides as SnO_2_ or ZnO functionalized CNTs [445,446]. In the first case a sensitivity of 10 ppm was obtained while in the second the LOD was 2 ppm. Electrochemical sensing was also applpied to detect CO although weak interaction and the limited charge transfer with CNTs [447]. Fe-porphyrin functionalized SWCNT were used to detect CO reaching a LOD of 80 ppm. Unfortunately, CO_2_ does not engage in strong binding or charge transfer. Boron and nitrogen doping have been observed theoretically to increase the interaction between CNTs and CO, CO_2_ thus increasing the sensitivity [448]. Oxygen plasma was another strategy to increase the CO detection sensitivity reaching a LOD of 5 ppm [449]. The sensor showed good selectivity towards other gaseous species as H_2_ and CH_4_. CO can be detected also using an organo-Co-CNT complex via chemoresistive measurements [450]. The CNTs were functionalized with an organo–(CP^CoI_2_) complex which is able to coordinate CO molecules as depicted in Figure 25.

The sensor showed a good gas selectivity towards other gasses as CO_2_, CH_4_, C_2_H_2_, H_2_, air, and a LOD of 90 ppm. Also for the detection of CO_2_ metals can improve the sensor performances. Among others, Au [451], Pd (theoretically) [452], SnO_2_ [453]. In the first case a plasmonic structure was performed depositing Au-docorated CNTs on a Bragg grating produced on a single mode optical fiber. The sensor showed a decent selectivity with respect to CH_4_, C_2_H_6_, C_3_H_8_, C_4_H_10_ air with a LOD of 150 ppm. Both sulphur based gases such as H_2_S or SO_2_ can also be detected using Au and Pt decorated CNTs. H_2_S is an electron donor while SO_2_ is an electron acceptor. Au decorated CNTs were able to detect H_2_S wit a sensitivity of 3 ppb [454]. In the case of SnO_2_ decorated CNT the chemo-resistive device had a sensitivity to CO of 1 ppm. There are just a few examples of SO_2_ which was detected using SWCNT at room temperature [455] with a sensitivity 200 times higher than that for O_2_. Finally also benzene, toluene, and xylene can be detected using CNT based sensors [456] with a LOD between 500 ppb–10 ppm. Detection of volatile organic compounds (VOCs) plays an increasing role in the environmental control and in biomedical applications. Benzene, toluene, methanol, ethanol and acetone were detected by impedence spectroscopy with a LOD of 1.62 ppm and 1.8 ppm for benzene and toluene respectively [457]. MWCNTs treated in an oxygen plasma to increase the sensitivity to benzene vapors to 800 ppb. VOCs detection is based on the adsorption of the organic molecules on the CNT walls which depending on the VOC chemistry, might be a rather slow process. To enhance the sensor performances in detecting VOCs as acetone CNTs were functionalized with poly-porphyrins rendering the sensor dynamic range very large from 50 to 230,000 ppm with a LOD of 9 ppm and a stable response over a period of 180 days [458]. Another possibility is the decoration of amine functionalized CNTs with Au nanoparticles which improves the detection of hexane, toluene, trichloroethylene and chloroform against polar compounds such as water, proponol and ethanol [459].

Also changes of pH influence the electrical properties of CNTs allowing detection. Poly(aniline) SWCNTs are sensitive to protonation rendering the complex useful for pH sensing [460]. High stability was obtained by coordinating SWCNTs with poly(1-amino anthracene) resulting in a sensor providing a stable response over a period of 120 hours in a 2–12 pH range [461]. Other information can be found in [462,463,464]. The possibility to integrate CNTs in smart low cost microcircuits enables their use as sensors in the agro-food sector. An example is the control of fruit ripeness through the control of the fruit-ripening hormone ethylene. Virgin CNTs are insensitive to ethylene molecules which then can be detected only by an appropriate functionalization. It has been demonstrated the decoration of CNTs with SnO_2_ provides the required sensitivity with a LOD of 3 ppm [465]. However, SnO_2_ is also sensitive to NO_2_ which may be a problem which was solved by a complexation of SWCNTs with fluorinated tris(pyrazolyl)-borate copper(I) [466]. CNTs are also utilized to detect food smells [467,468] using a OR 2AG1 protein/CNTs complex and reaching a sensitivity of 1 fM for Amylbutirate. Food integrity was controlled through detection of bio-amines and ammonia using cobalt meso-aryl-porphyrin complexes [469]. Preservation of food and product integrity takes advantage form the control of the presence of oxygen in the packaging. As already described CNTs can be utilized to detect oxygen and thus the degree of product quality [470].

An important application of CNT is the detection of biomolecules which can be important for biomedical applications. Diabete is one of the pathologies experiencing an increasing impact in modern societies. As a matter of fact, detection of glucose is one of the most frequently performed medical tests. The sensing is based on the catalytic effect of glucose oxidase (GOx)/CNT complex and a direct electron transfer to the sensor upon glucose recognition [471]. However this electrode suffers from poor stability, problem which was solved adsorbing GOx onto Pt nanoparticle decorated CNT electrodes. The new sensor displayed a good linear range (0.1−13.5 mM) and high sensitivity of 14 μA/mM [472]. Other authors utilized a forest of vertically aligned CNTs coordinated with the glucose oxidase enzyme and were able to demonstate that CNT act as conductive nanowires connecting the enzyme redox-active site to the transducer surface [473]. Other authors developed a Polyaniline/functionalized SWCNT/Prussian Blue complex able to selectively detect glucose among acetaminophen, uric acid, lactate, and ascorbic acid which are common interfering species [474] with a sensitivity of 18.6 μA/(m cm^2^). In another work, CNTs served as chemoresisitive glucose sensors [475]. To render highly hydrophilic, SWCNTs were functionalized with poly(4-vinylpyridine) chains and with GOx molecules. Exposure to glucose induce the formation of hydrogen peroxide which causes a decrease if the SWCNTs resistance. This last changed linearly in a range between 0.08 and 2.2 mM glucose. In addition the hydrogen peroxide attacks the SWCNT walls increasing the number of defects thus inducing changes in the I(D)/I(G) Raman components allowing an indirect measure of the glucose concentration. 

Besides glucose also the measurement of the cholesterol level in the blood stream is important. This were accomplished using screen printed electrodes functionalized with cholesterol esterase, peroxidase, oxidase and MWNTs [476]. The sensor showed a linear response over a range from 20 to 200 mg/dL and 100 to 400 mg/dL for uric acid and cholesterol and the respective sensitivities were of 0.0721 and 0.0059 μA per mg/dL. In another work authors used CVD grown vertically aligned CNTs which were subsequently functionalized with polyaniline and cholesterol esterase to detect cholesterol [477]. The sensor showed a quasi-linear response in the range 50–300 mg/dl with a sensitivity of 22 μA/(mg dL). The sensor showed good repeatability and selectivity toward interfering glucose, uric acid acetaminophen and ascorbic acid. In parallel to the detection of glucose, it has been developed a CNT based sensor for insulin enzyme which is important for the treatment of type I diabetes [478]. The sensor was fabricated using dimethylformamide modified CNTs casted on a glassy carbon electrode. The sensor at physiological pH showed a linear response up to 1000 nM insulin and a LOD of 14 nM. In [479] RuOx modified CNTs were utilized to sense insulin. The modified electrode showed a linear response in the range 10–800 nM and a LOD of 1 nM. Another important parameter is the concentration of NO in the blood. Nitric oxide (NO) has different functions acting as a neurotransmitter, is involved in regulation of feelings, of pain, appetite, the circadian cycle, thermoregulation, synaptic plasticity, and neural secretion [480]. SWCNT are known to emit in the NIR region but the fluorescence signal is quenched in presence of NO thus enabling an optical detection. In [481] CNTs were modified binding DNA sequence of d(AT)_15_ oligonucleotides thus enabling single molecule sensitivity. In addition fluorescence quenching was shown to be proportional to the NO concentration adsorbed on the electrode. In another work, DNA oligonucleotide ds(AAAT)_7_ were bonded to CNTs using polyethylene glycol spacers [482]. These modified CNTs were injected in mice and allowed in vivo measurements of NO with a LOD of 1 μM. CNT functionalized with N-hydroxyphenyl maleimide were also used to sense other biomolecules as epinephrine [483] with a linear response in the range 0.09–5.90 ng/mL and a LOD = 0.02 ng/mL. Au decorated CNTs were utilized to sense dopamine [484] via square wave voltammetry with a linear range between 0.48 μM to 5.7 μM, a LOD of 0.071 μM. More information may be found in [485]. SWCNTs were used to detect H_2_O_2_ in carcinoma cells [486] via fluorescence quenching. Sensitivity of the technique is down to the single H_2_O_2_ molecule secreted by A431 carcinoma cells in in vitro experiments. 

Besides biomolecule sensors, CNTs are also used to fabricate DNA sensors. Ability of detecting specific DNA sequences is important for the diagnosis and the treatment of diseases generated by DNA aberration and genetic disorder. A DNA sensor was fabricated with mercaptohexonal and thiolated DNA sequence were attached to gold electrodes [487]. Hybridization with the complementary DNA sequence occurring on the sidewalls of SWCNT-FET induced a pronounced electrical conductance change due to the modulation of energy level alignment between SWNT and gold contact [487]. The sensing mechanism is depicted in Figure 26A. 

The sensitivity of the sensor was 4 × 10^13^ molecules/cm^2^. Frequently, in a sensor an electrode is modified with CNTs functionalized with DNA sequences. The DNA hybridization with the complementary sequence leads to a change of the current flowing through the electrode [489]. To increase the detection sensitivity, authors used alkaline phosphatase enzyme to preconcentrate the analyte thus amplifying the electrical sensing of proteins and DNA resulting in a LOD of the target DNA of 1 fg mL^−1^ [488]. Schematic representation of the analytical protocol and detection is represented in Figure 26B. It is also possible to fabricate CNT based devices for sensing any kind of DNA sequence [490]. To achieve this result, authors functionalized CNTs with a specific peptide (with the sequence Fmoc-RRMEHRMEW) as natural molecules having the unique ability to selectively bind to universal DNA sequences. The sensor showed a broad sensing range from nearly 1.6 × 10^−4^ to 5 μmol L^−1^ and a much lower detection limit of approximately 0.88 μg/L. An easy detection of DNA sequences was performed by mixing oxygen-functionalized MWCNTs in a buffer solution containing the DNA sequences [491]. The MWCNTs were able to modulate the sensor electrochemical current depending on the sequence of the target DNA and exhibited a LOD of 141.2 pM and a good DNA sequence differentiation. There are other sectors in which CNT based sensors are applied: detection of dangerous metal ions [492,493]. Pb, Cd, Hg, As, Zn, Cu, Ag, Cr, Ni, and Ti, ecc may be detected using CNT based sensors [494]. For Pb, Cd, Cu, Zn the LOD varies between 0.017 to 4.4 ppb [494]. CNT based sensors are utilized to detect pesticides and an overview is presented in [495]. Also pathogens may be sensed as presented in [496] or explosives [497]. 

Table 5 summarizes characteristics of CNT based sensors.

As a conclusion of this brief review on the use of CNTs for sensing, it clearly appears the versatility of these carbon nanostructures whose prominent properties enable a variety of different sensing applications. CNTs appear as the material of choice for biomedical sensing, environmental monitoring, food and agricultural applications, and application for security. The high flexibility offered by CNTs in the surface chemistry/functionalization, the possibility to integrate them in electronic device, the miniaturization of the sensor architecture and the high sensitivity offered are very important factors which are certain to play a role in a continuous evolution and commercialization of this technology. The rapid development of sensing methods based on CNTs indicate that superior responses are often obtained competing and likely surpassing most other sensor devices allowing application of this technology to protect and improve our environment, safety, and health. Some of the detection modalities in CNT based sensors are depicted in Figure 27.

## 7. Graphene

Graphene is a single layer of carbon atoms organized in a two-dimensional structure with a trigonal planar lattice. In graphene all the carbon atoms are sp^2^ hybrids characterized by sp_xy_-mixed orbitals oriented along directions at 120° while the remaining p_z_ orbital is orthogonal to the xy plane. The three sp_xy_ hybrids lead to the formation of hexagonal rings where each carbon atom is connected to other three carbon atoms via strong covalent σ bonds. Differently, unhybridized p_z_-orbitals generate rather weak π bonds formed by two lobes placed above and below the xy plane [43]. The σ bonds of the hexagonal lattice have a length of ~1.42 Å (let us remind that the length of the strong σ bonds of diamond is ~1.54 Å). This mirrors the in-plane great mechanical strength of graphene, with breaking strength of 42 N m^−1^ and Young’s modulus of 1.0 TPa [500]. Different from the strongly localized σ bond electrons, the conjugated out-of-planar π bond electrons are quasi-free.

The particular structure of graphene leads to an electronic band structure with valence and conduction bands are cone-like structures touching each other near the Dirac points K and K’ as shown in Figure 28.

These features lead to two important consequences. First the free carriers in graphene follow a linear dispersion relation and behave as massless relativistic quasi-particles with unprecedented high mobility of 200,000 cm^2^ V^−1^ s^−1^ in suspended graphene and Fermi velocity of ~1.10 × 10^6^ m/s and an electrical conductivity of 6300 S cm^−1^ [501]. Second, the two-dimensional honeycomb lattice of graphene displays a sublattice (chiral) symmetry [502]. The structural properties of single layer graphene are also mirrored by the high thermal conductivity of about 5000 Wm K^−1^ [503]. Finally, thanks to its single-atom thickness and 2D structure, graphene possesses an extraordinary high theoretical specific surface area of 2630 m^2^ g^−1^ [504]. However, despite graphene is a single atomic layer structure, it displays an unusual opacity absorbing a significant fraction of the incident light equal to 2.3% [505].

### 7.1. Graphene Synthesis and Functionalization

Despite the graphene structure is referred to the single sheet of carbon atoms, the name graphene is commonly used to indicate a class of materials formed by more stacked planes. The synthesis of graphene includes different methods resulting in materials possessing different properties e.g., layer number, lateral dimension, chemical residuals, surface charge and surface functional groups. Several synthesis methods have been developed to produce graphene and can be categorized in bottom-up and top-down approaches.

#### 7.1.1. Bottom-Up Synthesis

In bottom-up approaches graphene is built from atom levels through three main synthesis methods: epitaxial growth on silicon carbide (SiC), chemical vapor deposition method (CVD) and plasma-enhanced chemical vapor deposition (PECVD). Epitaxial graphene growth on SiC is based on the high temperature thermal decomposition of SiC under vacuum or in inert gas [506]. In SiC, Si and C are in 1:1 stoichiometry and the Si—C bonds along the tetrahedral directions lead to a hexagonal layered structure where carbon and silicon are in alternating positions. Strictly speaking, the growth mechanisms of a graphene layer on the SiC surface is related to the difference in the vapor pressures of silicon and carbon. At high temperature the Si atoms sublimate while the remaining carbon atoms self-reorganize in the graphene structure with strong in-plane sp^2^ bonds [507].

This method was firstly reported in 1965 by Badami who annealed SiC in vacuum at 2280 °C for an hour leading to the development of a graphite lattice crystal surface [508]. Successive studies suggested that a graphene monolayer already formed at 800 °C under ultra-high vacuum [509]. Authors also observed that graphite were formed on both Si- and C-faces of SiC but the Si-face led to a monocrystalline graphite layer while the C-face generated a polycrystalline graphite layer. However, the quality of the graphene obtained by this method is low because variable thickness, the small graphene grains caused by the changes in the surface morphology of SiC during sublimation. Figure 29A displays a typical graphene film grown on SiC: the formation of graphene is not a self-limiting process and with temperature Si sublimation proceeds with formation of a multilayer. To improve the graphene quality, SiC annealing was performed in an argon atmosphere [510] or in an external Si flux [511], or depositing a nickel [512] or cobalt [513] layer on SiC. Epitaxial graphene derived from SiC is useful to produce ready-to-use graphene for electronic applications. However, for other kinds of applications graphene transfer is required limiting the utility of this synthesis process.

CVD is a good alternative to the epitaxial growth on SiC, to produce high quality, large area graphene although the properties of the produced graphene, e.g., number of layers and crystallinity, are affected by several factors such as precursors, catalyst layer, assistant gases, and temperature. In this CVD process, a hydrocarbon precursor gas introduced in the reactor chamber, is catalytically decomposed at high temperature [514]. The carbon radicals arrange into a graphene structure on the catalytic layer of the substrate. In CVD synthesis, methane is the most used precursor although other gasses as acetylene [515], ethylene [516], propene [517], or liquid precursors like methanol [518], ethanol [519], propanol [518], hexane [520], benzene [521], or even solid precursors like poly(methyl methacrylate) (PMMA) [522], amorphous carbon [523]. Deposition is made using different catalysts including nickel (Ni) [524], copper (Cu) [524], rhodium (Rh) [525], cobalt (Co) [526] or alloys [527,528,529]. The catalyst layer plays a role not only to lower the activation energy for the precursor decomposition, but also triggers different graphene growing mechanisms. For example, Ni induces carbon segregation and precipitation processes because at high temperature C has a high solubility in Ni. However, decreasing temperature decreases also the carbon solubility, carbon atoms diffuse out from the Ni-C solid solution and precipitate on the Ni surface producing a graphene films. Cu has much lower carbon solubility than Ni and only a small amount of carbon dissolves in this catalyzer. Thus, using Cu graphene grows directly from the precursor decomposition. In addition, the growth of graphene growth on Cu has the advantage to be a self-limiting process, which is important to form single layer graphene [524,530]. High resolution TEM structure of a CVD deposited graphene is shown in Figure 29B,C showing the perfect hexagonal planar structure of the carbon atoms observed at RT and 700 °C applied to desorb contaminants. The correspondent diffraction patterns are shown in the insets. A careful control of the CVD deposition parameters allows also the growth of large graphene single crystal shown in Figure 29D, also reaching macroscopic dimensions as in Figure 29E.

In PECVD processes the plasma decomposes the precursor allowing the synthesis of graphene at lower temperatures and shorter deposition times [531]. This method can overcome the evaporation problem of catalyst layer at high temperature. However, one drawback of the PECVD process is risk of the generation of plenty of defects in the graphene structure caused by the presence of the plasma [532].

#### 7.1.2. Top-Down Synthesis

In the top-down synthesis, graphene sheets are detached from high-quality graphite crystals using mechanical or chemical methods namely the mechanical exfoliation method, the chemical exfoliation method followed by reduction of oxidized graphene.

The mechanical exfoliation was the first documented method to successfully separate graphene from crystalline graphite. Exfoliation can be made using a simple adhesive tape to break the van de Waals forces between graphite layers [536]. High quality graphene is produced using this method which, however, is not scalable. Graphene sheets are also produced by applying ultrasonic force to graphite in solvents as N-methylpyrrolidone (NMP) which relies on the strong interactions between NMP and graphene sheets [537]. Another mechanical exfoliation method is the use of the shear force in the ball-milling process. Ball-milling can be performed in both wet [538] and dry form [539]. A common problem of ultrasonic and ball-milling methods is the presence of some un-exfoliated graphite mixed with graphene sheets difficult to eliminate. In addition, ball-milling also produces undesired amorphous carbon domains and defects in the graphene sheets.

Chemical exfoliation is a versatile method for the large-scale synthesis of graphene in a two steps process: (i) production of oxidized graphene (GO) sheets, and (ii) oxygen removal from GO via reduction processes. To generate GO, Hummer’s method is usually used exfoliate crystalline graphite. The exfoliation process utilizes sodium nitrate, concentrated sulfuric acid and potassium permanganate, leading to strongly oxidized graphene sheets [540]. Graphene oxide layers can be separated by ultrasonic treatment in polar solvent, especially in water, due to the enlarged layer spacing introduced by the oxygen functional groups. Finally, the graphene structure can be restored using electrochemical reduction [541], thermal reduction [542], chemical reduction [543] or hydro/solvothermal method [544]. Figure 30A,B displays the structure of graphene oxide obtained by exfoliation. TEM shows the presence of defects which appear more clearly in Figure 30C,D. Chemical reduction is widely for GO reduction and a variety of reductant were studied. There are several reductants which are effective for GO reduction to rGO, e.g., hydrazine [545], sodium borohydride [546], hydrohalic acid [547], ascorbic acid [548]. The selection of the appropriate reductant is linked to the kind of application envisaged. Chemical residuals from reductants may affect the biocompatibility of the final product since most of them are toxic. To solve this problem, recently green reductants as plant extracts, sugars, microorganisms and amino acids have been tested [549]. As an example, nicotinamide adenine dinucleotide phosphate (NADPH) contained in baker’s yeast was utilized to reduce and, at the same time, functionalize the rGO. Specifically, the amine groups of NADPH couple with the epoxy functionalities of GO resulting in a stable water suspension of yeast-rGO [550]. The structure of reduced GO is shown in Figure 30E while in Figure 30F are highlighted the different defects present on the graphene flakes after the reduction process.

### 7.2. Graphene Functionalization

As for the other carbon nanostructures, chemical functionalization of graphene is one of the crucial steps of material preparation determining the final properties and the kind of applications namely electronics, filler in nanocomposite synthesis, biomaterials/biomedicine, sensing, energy, environment. Functionalization transforms the zero-gap graphene in a semiconductor suitable for electronic applications [553]. In the case of biomaterials, functionalization renders graphene soluble in water, an appropriate surface chemistry renders graphene a suitable nanoplatform for drug delivery, such as tissue engineering and other applications [554].

Graphene is broadly utilized as the ultimate electrode material for various electrocatalytic applications. Modification of the surface chemistry with appropriate functional groups makes the resulting electrode more efficient than the individual components [555].

The chemistry of graphene resembles that of graphite and other sp^2^ hybridized carbon allotropes [556]. As in CNTs, also in graphene the presence of defects play an important role. Atoms at the edges of defects are generally more reactive than the surrounding π surface although functionalization of graphene may occur also on the π surface [557] as we will describe later on. As for CNTs, also in the case of graphene functionalization may be performed by covalent or non-covalent routes.

#### 7.2.1. Covalent Functionalization

Among the chemical modifications of the graphene surface, those regarding the graphene oxide and the reduction processes have been extensively studied. This enables grafting of a large number of different atoms/organic groups on pristine graphene, radical additions, electrophilic substitution, and cycloaddition reactions. Talking about graphene oxide (GO) studies regarding this material dates back to the 18th when Brodie demonstrated the possibility of obtaining graphene oxide from graphite using potassium chlorate and fuming nitric acid [558]. An alternative was proposed in the sixties by Hummers, and Hoffman who used permanganate and concentrated sulfuric acid or a mixture of sulfuric and nitric acids, achieving similar levels of oxidation of graphite. However it has been demonstrated that the oxidation level depends not only on the nature of the oxidants but also on the quality of the graphite utilized and on the reaction conditions [559]. These scalable processes lead to the formation of hydroxyl and epoxides, and smaller amounts carboxy, carbonyl, phenol, lactone, and quinone groups [560] (see Figure 31A at the sheet edges groups breaking the continuity of the aromatic lattice and inducing a change of the C atom’s hybridization [561]. Thanks to the high concentration of polar functional groups, GO is readily dispersible in water. GO is a functional group rich material, but it may be considered as a starting substance for the preparation specific compounds. Moreover, the hydrophilicity of GO allowed its uniform deposition on different substrates as thin films, necessary for applications in electronics [108]. The presence of oxygen based functional groups allows to further tune the surface chemistry adding desired functional molecules. In [562] a selective functionalization of the hydroxyl groups was carried out using an esterification reaction with aminocaproic acid (see reaction scheme in Figure 31A) and the Williamson reaction with tert-butyl (4-iodobutyl)carbamate. Selective derivatization of Ketones by the Wittig reaction was also studied. The high surface area GO may be used as a support for nanoparticles dispersion, to maintain their electrochemical activities maximizing the electron transfer and also providing optimal mass transport of the reactants to the electroactive site. Nanoparticles are made of metal or metal oxides and are attached to the GO substrate for metal nanoparticles (such as Pt, Au, Ru, Ag, Cu…) and oxide nanoparticles (TiO_2_, ZnO, SnO_2_, Cu_2_O, MnO_2_, Mn_3_O_4_, NiO, SiO_2_) have been described for the fabrication of electrodes based on GO [563,564,565,566]. Moreover, quantum dots are utilized to functionalize GO for producing platforms for electrochemical applications [567,568]. Other species utilized to functionalize GO are Prussian blue [569], metal hydroxides [570] used to modulate the electrochemical properties. There is a wide group of organic compounds used to functionalize GO for example porphyrins [571], aromatic dyes [572], alkylamines [573], ionic liquids [574], pyrene and perylenediimide [575], cyclodextrin [576], aryl diazonium compounds [577] and polymers [578].

Graphene sheets are also hydrogenated. Hydrogenation converts sp^2^ into sp^3^ hybrids. This induces significant changes in the electronic properties converting graphene from a highly conductive into an insulating material [579]. RF-CVD was used to functionalize graphene in a hydrogen atmosphere [580]. Other research groups studied the wet chemical routes as the Birch reduction for the preparation of hydrogenated graphene [581]. Change of the electronic properties of graphene is performed also with doping with halogenated elements [582]. As it happens for hydrogen, covalently bound halogens (graphene halides) strongly modify the graphene properties. For example, fluorographene is only stable stoichiometric graphene halide and the thinnest insulator. Halogenated graphenes exhibit a plethora of remarkable and interesting electronic, optical, thermal, electrocatalytic, magnetic, mechanical, biological, and chemical properties in comparison with their graphene counterparts [582].

Addition of organic functional groups is also possible and has been developed for example to render graphene sheets dispersible in common organic solvents, which is of pivotal importance for the production of nanocomposite materials (see Figure 31B). Radical addition to sp^2^ carbon hybrids is one of the routes utilized to overcome the low reactivity of pristine graphene. The radical addition of phenyl species obtained by heating of aryl diazonium salts is suitable option to form new covalent bonds to graphene [583]. The aryl diazonium salt covalent modification allows the preparation of functionalized graphene with refined dispersibility in polar organic solvents and water by modifying the substituents in the aryl-ring. This kind of functionalization was utilized for developing high performance graphene-based nanocomposites, with enhanced dispersibility of graphene nanosheets in polymer hosts leading to strong interlayer cohesive energy and surface inertia [584] (see Figure 31C). Covalent modification using 4-bromobenzene diazonium tetrafluoroborate (4-BBDT) was utilized to change the electronic properties of graphene [585]. Authors estimated a downward shift of the Fermi level dependent on the 4-BBDT molecule concentration thus offering the possibility to tune the electronic properties of graphene. The change of electronic properties was made using the diazonium based functionalization which results in a charge transfer which opens a bandgap transforming the graphene regions from quasimetallic to semiconducting and insulating [586]. Click chemistry is possible to perform highly regioselective, rapid and easy and efficient reaction to functionalize the graphene surface. Graphene was functionalized with 4-propargyloxybenzenediazonium tetrafluoroborate and a subsequent attachment by click chemistry of a short chain polyethylene glycol with terminal carboxylic end group [587]. The functionalization resulted in an increased ζ potential of the particulate thus increasing the wettability of the graphene. Another kind of covalent functionalization is the reaction of azomethine ylides with sp^2^ graphitic carbon atoms (see Figure 31D). This reaction has been successfully applied to carbon nanostructures as fullerenes, CNOs, CNTs, and also to graphene [588] despite the lack of surface curvature. This reaction, the Prato reaction, consists in a 1,3-dipolar cycloaddition reaction of azomethine ylides generated in situ by thermal condensation of aldehydes and α-amino acids [101]. Authors demonstrated that the reaction proceeds not only at the edges possessing higher reactivity but also at the internal carbon–carbon bond network. The Prato reaction is utilized to obtain a number of organic derivatives useful for different applications.

Graphene sheets were obtained from a dispersion of graphite in pyridine [589]. Dihydroxyphenyl groups perpendicular to the graphene surface were formed by the addition of azomethine ylide precursors. The presence of hydroxyl groups induced higher dispersibility of the functionalized graphene in polar solvents. The Prato reaction was utilized to add particular molecules while maintaining the graphene electronic transport. So, the basal plane of few-layer graphene was modified with 1,3-dipolar cycloaddition of azomethine ylides and the subsequently coupled covalently with an electron-donating phthalocyanine used for light-harvesting [590]. The system showed a very fast charge separation between phthalocyanine and the few-layer graphene and a slow recombination. It has been reported the graphene functionalization by nitrene chemistry based on different perfluorophenyl azides. Phenyl and alkyl azides can react with the C−C bonds of graphene by the formation of the reactive nitrene intermediate ([2+1] cycloaddition to the double bonds of graphene, forming an aziridino-ring) [591]. Depending on the functional group bonded to the perfluorophenyl azides the final product was soluble in water or organic compounds. The reaction is applicable regardless graphene size, shape, or configuration thus improving the processability, also considering the possibility to furthermore derivatize with additional molecules opening a number of opportunities in graphene-based materials. Alkyl azides, alkyl groups such as hexyl, dodecyl, hydroxyl-undecanyl, and carboxy-undecanyl have been inserted into surfactant-wrapped graphene sheets [592] allowing bonding gold NPs to the graphene surface used to visualize the reactive sites. Other possible reactions are the Bingel–Hirsch reaction [593], a nucleophilic [2+1] cyclopropanation of a double bond, usually used on fullerene but successfully applied also to graphene (see Figure 32A). The reaction proceeds reacting a bromomalonate in the presence of a base. In [594] graphene obtained by exfoliation was mixed to 8-diazabicyclo[5.4.0]undec-7-ene (DBU), carbon tetrabromide (CBr4), and the corresponding malonate. A Bingel reaction was activated by applying microwaves leading to cyclopropanate malonate units covalently grafted onto the honeycomb skeleton. The final product was a stable suspension for several days in a variety of organic solvents.

#### 7.2.2. Non-Covalent Functionalization

In noncovalent functionalization the linkage between graphene and the functional group is weak enough to preserve the intrinsic electronic structure of graphene. In non-covalent functionalization the interaction of the countermolecules to the hexagonal lattice preserves the intrinsic chemical and physical characteristics while adding interesting selective properties. This kind of functionalization is based on the so called π-π interaction established between aromatic rings. There are several molecules that can be utilized in non-covalent functionalization. Pyrene moiety has attracted great interest because the strong affinity toward the graphite basal planes. Graphene was functionalized with 1-pyrenebutyrate to increase the aqueous dispersibility of graphene in aqueous medium [595]. In another work π–π noncovalent functionalization was used to attach pyrenebutanoic acid succinimidyl ester to graphene obtaining an enhancement of the power conversion efficiency in graphene-based photovoltaic devices [596]. Unique optical and sensing properties were obtained with a 1-pyrenecarboxylic acid functionalization of graphene [597]. Other examples of non-covalent functionalization may be found in [598]. Some examples of π-π interaction are shown in Figure 32B.

### 7.3. Graphene Sensing

Advantageous utilization of graphene to develop sensing devices is due to the distinct properties of this material as the exceptional specific surface area, the unique optical properties, excellent electrical and thermal properties, the high strength and the flexibility. Notably, these properties persist also in the double and multi-layered graphene structures. A supplementary important feature of the graphene platform is the possibility to adapt its structure and the working conditions accordingly to the application. In electrochemical sensors, the functionalization and the surface engineering pay a crucial role in coupling the analyte molecule to the electrode surface. As for strain, among the two-dimensional materials graphene is the strongest being able to sustain reversible tensile elastic strain larger than 20%. This feature enables the possibility use graphene as a sensor being its properties changed by strain. This fact derives from the strain induced modification of the electronic structure as the shift of the Dirac cones [602] and the generation of a pseudo-magnetic field and a reduction of the Fermi velocity [603]. Electron phonon coupling is also sensitive to the application of strains. In particular under the application of uniaxial strains, the G band redshifts and splits into two single G+ and G− bands [604,605]. The G+ and G− band splitting as a function of the applied uniaxial strain is shown in Figure 33. The Raman scattering from the G+ and G− bands shows a distinctive polarization dependence on the angle between the strain direction and the underlying graphene crystal axes. The G+ and G− relative intensities vary with polarization, thus allowing to obtain the strain direction with respect to sample crystallographic orientation.

These graphene properties may be utilized to derive information about the applied strain. Another important parameter is the gauge factor reflecting the efficiency in generating a change in the electric properties of the composite material based on mechanical deformations. In [606], authors mixed graphene with natural rubber to create conducting composites. The material displayed excellent strain sensing capability with a 10^4^-fold increases in resistance and working at strains exceeding 800%. In another work sensors were fabricated by depositing graphene flakes on flexible plastic supports [607]. The change of the flake-flake overlap for different percolation networks upon application of a strain explains the changes in the gauge sensitivity of the material associated to the superior sensitivity of the sensor (see Figure 34). Flexible plastic or stretchable rubber substrates were also used to fabricate graphene based piezo-resistive strain sensor [608]. The sensor properties were studied upon applied strains up to 7.1%. The sensor performances were tested on a glove showing the ability to measure magnitudes and directions of the principal strains on the glove induced by the motion of fingers. A film of GO was deposited on a PET coated DVD and was reduced using a Light-Scribe DVD burner [609]. Under substrate deformation the device showed a good linear response to strain and good multi-cycle operation opening perspectives for wide applications in medical-sensing, bio-sensing, artificial skin and many other areas.

To overcome the fragility, reduced graphene oxide aerogels were mixed to water soluble polyimide leading to formation of superflexible 3D architectures [610]. The rGO/polyimide was produced by freeze casting and thermal annealing. The resulting monoliths exhibited as low density, excellent flexibility, super-elasticity. The high recovery rate, and extraordinary reversible compressibility were exploited for sensing strains. The sensor showed perfect linearity: a compression strain of 10%, resulted in an electronic resistance change of 9.96%. Interestingly, graphene strain sensors may be fabricated using textiles [611]. The sensing properties are obtained by using graphene as a filler in the textile fibers to obtain conduction properties or the using graphene served as conductive coatings. The response of the sensors shows a linear behavior only within a limited strain range due to the slip inside the fiber or fabric and the change of textile strain sensor structure [611]. Although the properties of the graphene-based textile sensors have to be improved, they have important potentialities in many sectors as flexible and wearable sensors for electronic skin, human-machine interfaces, human activities monitoring, intelligent robots, and human health detection.

Detection of substances based on graphene is an extremely broad area of research either for the number of molecules probed and for the different methods used. Hereafter we will review only the electrochemical graphene-based sensors.

A broad area of sensing is based on the electrochemical reactions developed at the electrode/analyte interface. A material is suitable as electrochemical sensor if it possesses a large electrochemical potential window, good electron transfer rate, appropriate redox potentials. Graphene has a wide electrochemical potential window of ca. 2.5 V in 0.1 M PBS at pH 7.0 [612], which is comparable to that of graphite, GCE [612,613], but possesses a charge-transfer resistance much lower than these electrodes [612]. Finally, the peculiar electronic structure of graphene, and the high density of the electronic states over a wide energy range are the reasons for the graphene fast electron transfer [613].

Among the possible sensing application of graphene those related to the detection of biomolecules is one of the more important. In this respect, much attention was paid to the glucose sensing. As for the CNTs, also in the case of graphene enzymatic detection of glucose is based on glucose oxidase (GO_x_). To increase the device sensitivity, GO sheets were decorated with gold nanoparticles through a benzene (Ph) bridge employing aryldiazonium salt chemistry [614]. The functionalized GO sheets were then attached to a 4-aminophenyl functionalized GCE. Finally, the GOx enzyme was attached to the gold NP through a 4-carboxyphenyl linker (see Figure 35). The sensor showed a linear behavior in the range of 0.3–20 mM with a sensitivity of 42 mA mM^−1^cm^−2^ for glucose detection and a fast electron transfer and enzyme turnover rates of 8.3 and 112/s, respectively. This sensor was characterized also by a high selectivity for glucose showing an almost null amperometric response when in contact with 100 mM of interfering analytes as dopamine, ascorbic acid, uric acid, acetaminophen, fructose, lactose, and galactose. In an attempt to increase the sensitivity, newer sensors try to implement a direct electron transfer allowing amperometric measurement of the glucose level. In some sensors the GOx enzyme to be directly attached to the sensing electrode [615] reaching a LOD of 1.7 μM with a linear response between 0.02 and 3.2 mM. However, an efficient direct electron transfer between the biocatalyst and the electrode surface remains a key challenge.

Increased efficiency was obtained using the flavin adenine dinucleotide (FAD) as mediators in the redox reactions [616]. Other researchers tried to increase the sensing efficiency by tailoring the electronic properties of the electrode. As an example a GCE was engineered with a direct electrochemical reduction of single-layer GO and subsequent functionalization with (3-aminopropyl)triethoxysilane (APTES) used to immobilize GOx by covalent bonds. The GCE-APTES-rGO-GOx showed a linear behavior in a range 0 to 24 mM glucose [617]. There are a series of different attempt to fabricate sensors with improved sensing capability. In [515] used good performances were obtained using a CVD-grown graphene, in other works the sensing electrode was coated with a Layer-by-layer deposition of graphene and GOx through π-π interaction [618] or immobilization of GOx on a porous-graphene structure [615]. Other sensors are based on Graphene/polymer composites as for example polyethylene glycol [619], polyvinylpyrrolidone [620], or chitosan [621]. A final note concerns the research made to overcome degradation of the GOx enzyme induced by pH, temperature, pressure or humidity. Non-enzymatic glucose sensors can be fabricated using nitrogen-doped graphene-encapsulated nickel cobalt nitride (NiCo2N) core–shell nanostructures [622]. The sensor displayed a linear behavior in a wide range from 2.008 μM to 7.15 mM, an excellent sensitivity of 1803 μA mM^−1^cm^−2^, and a low detection limit of 50 nM. In addition, the NiCo2N decorated graphene electrode showed a remarkable H_2_O_2_ sensitivity of 2848.73 μA mM^−1^cm^−2^, a wide detection range of 200 nM to 3.4985 mM, and a LOD of 200 nM coupled to a short response time of <3 s. Glucose was detected alsu using nitrogen-doped graphene decorated with copper nanoparticles [623], displaying a linear response in the range 0.004–4.5 mM and a LOD of 48.13 μA mM^−1^. Another possibility is the use of Pt nanoparticles supported on GO hydrogel [624]. In this case the electrode showed a linear behavior in the range 5–20 mM and a sensitivity of 137.4 μA mM^−1^cm^−2^. Finally, an electrode was fabricated using PtNi bimetallic nanoparticles supported by graphene. The electrode performances were a response current linear to glucose concentration up to 35 mM and a sensitivity of 20.42 μA mM^−1^cm^−2^ [625].

There are other molecules important for their role in biological processes. Among others, cholesterol and its fatty acid esters are important because one of the main constituents of mammalian cell membranes and are precursors of extra biological materials. In addition, cholesterol is one of the blood constituents and its imbalance is at the basis of various heart diseases thus making its detection very important. As made for glucose, also the determination of the cholesterol level may be done either by enzymatic and non-enzymatic routes. In the first case concentration of cholesterol is based on the detection of the H_2_O_2_ level which is the product reaction of cholesterol with cholesterol oxidase (ChOx). Graphene nanocomposites are generally used to fabricate cholesterol sensors. As an example, in [626] ChOx enzyme-functionalized GO was conjugated with ferrocene molecules to enhance the electron transfer. The sensor showed a linear response ranging from 0.5 to 46.5 mM with a sensitivity of 5.71 mA mM^−1^cm^−2^ and a LOD of 0.1 mM. A higher sensitivity of 1 μM was obtained using a graphene/poly(vinylpyrrolidone)/poly(aniline) nanocomposite to fabricate the cholesterol sensing electrode [627]. The sensor showed a linear behavior in the range from 50 mM to 10 mM. Metal nanoparticles are also used to increase the performances of the sensors. Among others, Pt and Pd nanoparticles are well known for their superior electrocatalytic activity. A hybrid graphene Pt decorated electrode surface was synthesized in [628] for cholesterol detection. Graphene was decorated with Pt nanoparticles by an in-situ reduction of H_2_PtCl_6_ in aqueous solution and functionalized with cholesterol oxidase (ChOx) and cholesterol esterase (ChEt). Cholesterol detection via H_2_O_2_ resulted in a linear response up to 12 mM with a LOD of 0.5 nM. The same graphene/Pt hybrid material was mixed with 3% ethanolic nafion solution and drop-casted on a CGE. Two types of sensors were developed: (i) a biosensor for free cholesterol using the enzyme ChOx and (ii) a biosensor for total cholesterol measurement using the enzymes ChOx and ChEt. The sensors showed a linear response up to a 35 μM cholesterol concentration and a LOD of cholesterol ester of 0.2 μM (see Figure 36).

A cholesterol biosensor was fabricated using a rGO decorated with Pd nanoparticles [629]. The high electrocatalytic activity of dendritic Pd nanostructures led to a remarkable sensitivity of 5.12 mA mM^−1^ cm^−2^ and a LOD of 0.05 mM. In another work a chitosan/graphene hybrid support was decorated using Pd/Pt bimetallic nanoparticles [630]. The synergistic activity of Pd and Pt catalyzers led to an accelerated direct electron transfer between ChOx and electrode surface. The cholesterol sensor showed a high dynamic range from 2.2 × 10^−6^ to 5.2 × 10^−4^ M, and a LOD was 0.75 μM. The same problems of stability affecting the GOx also influence the GhOx enzyme. To solve the problem a non-enzymatic cholesterol biosensor was fabricated using polyaniline nanofiber/graphene microflowers for the detection of cholesterol, with an impressive sensitivity of 0.101 mA mg^−1^ dL cm^−2^ and an LOD of 1.93 mg dL^−1^ [631]. A different solution was adopted in [632], where authors used a nickel oxide (NiO)/graphene composite. The sensor displayed a sensitivity of f 40.6 A μM^−1^ cm^−2^, and a LOD of 0.13 mM. The sensor showed a fast response time of 5 s and a long-term stability.

Hydrogen peroxide (H_2_O_2_) is another molecule important because is not only a byproduct of several enzymatic reactions but appears as a mediator in pharmaceutical, biomedical, environmental applications. This attracted great interest for developing H_2_O_2_ sensors although the presence of overpotentials at the working electrode likely generate a non-negligible interference [633]. H_2_O_2_ was detected using graphene produced via an organic salts assisted exfoliation method [634]. The sensor showed an improved electrocatalytic activity with a linear response over a range from 2.0 to 437 μM and a LOD of 0.19 μM. The use of graphene decorated with Ag, Au, Pt, Pd, and Cu metal nanoparticles was investigated by several groups due to their catalytic properties. Pt nanoparticles were deposited on graphene sheets through a microwave-assisted heating method and utilized to detect H_2_O_2_ [635]. The presence of Pt nanoparticles resulted in a linear response of the sensor in the range 1–500 mM and a sensor LOD of 80 nM. Both Ag and Au possess high catalytic activity with respect to H_2_O_2_. Ag nanoparticles decorated graphene was synthesized by a hydrothermal procedure [636]. The Ag led to improved catalytic and conductivity of the 3D porous graphene structure used as sensing electrode with a linear range between 0.03–16.21 mM and an LOD of 14.9 μM. In addition to nanoparticles also Ag nanorods [637] resulting in a linear sensor response in the range 0.1 to 70 mM and a LOD of 2.04 μM. An alternative solution is the use of graphene/nanowires [638] hybrids leading to a sensor linearity in the range 10.0 μM to 34.3 mM and a sensitivity of 12.37 μA mM^−1^cm^−2^In addition, other noble metal nanoparticles were utilized as Au [639] (linear range 0.5 μM to 4.9 mM, sensitivity of 75.9 μA mM^−1^cm^−2^), Pt/rGO nanoparticles [640] (linear range: 1 to 900 μM and a LOD of 0.1 μM), Pt/Ni nanowires/graphene [641] (linear range: 1 nM to 5.3 mM, LOD of 0.3 nM) or Cu_2_O/(3D, 2D) rGO structure [642] (linear range: 1 μM to 1.47 mM, were utilized for sensing H_2_O_2_. There are other molecules important for their role in biological processes. β-nicotinamide adenine dinucleotide (NAD^+^) and its reduced form (NADH) is a key factor in dehydrogenase processes which raised interest for the development of amperometric sensors. These latter are based on the redox reaction at the sensing electrode where NADH is oxidized in NAD^+^ which is important for the detection of lactate, alcohol or glucose [643]. The sensing was based on a GCE modified with graphene decorated with tungstate nanoparticles. The sensor was able to detect both NADH and NAD^+^ displaying a linear behavior for NADH and NAD^+^ from 10–270 μM and 100–500 μM and LOD equal to 2.188 μM and 49.8 μM for NADH and NAD^+^. Other biomolecules of interest are dopamine (DA) an important molecule involved in the transmission of neural signals ascorbic acid (AA) is a potent reducing and antioxidant agent and uric acid (UA) the final product of the purine metabolism and, as AA, is a potent antioxidant. Excess of UA is a symptom of altered health conditions including diabetes or kidney malfunctioning. Detection of DA, AA and UA is relevant for teranostics. A GCE modified with GO was utilized for the detection of DA in presence of interfering AA bioanalyte [644]. The sensor had a linear response in the range from 1.0 μM to 15.0 μM and a LOD of 0.27 μM for detecting DA. In addition, no interference was observed by the addition of AA thanks to the electrostatic repulsion towards GO. Simultaneous detection of DA, AA and UA was performed using Au/graphene [645]. The sensor performances were: linear responses to AA, DA and UA in the ranges 30–2000, 0.5–150 and 0.5–60 μM, respectively, while the LOD for AA, DA and UA are 10, 0.15 and 0.21 μM, respectively. DA, AA and UA can be simultaneously detected using Pt/graphene+CNTs [646] in respective concentration ranges of 200–900 mcM, 0.2–30 μM, 0.1–50 μM, showed good linearity and sensitivities of 0.186 μA M^−1^ cm^−2^ for AA, 9.199 μA M^−1^ cm^−2^ for DA and 9.386 μA M^−1^ cm^−2^ for UA. Other solutions to sense DA, AA and UA is the use of Ag nanowires/rGO [647] nanostructures. The sensor had a linear response of AA, DA and UA in the concentration range of 45–1550, 40–450 and 35–300 μM with a detection limit of 0.81, 0.26 and 0.30 μM respectively. Decorated graphene/polymer hybrid systems are also utilized to detect DA, AA and UA. For example sensors were fabricated using Pd/graphene+chistosan [648] with peak potential separations between AA and DA, DA and UA, AA and UA were 252 mV, 144 mV and 396 mV, respectively. In addition, the LOD (S/N = 3) for AA, DA and UA were 20 μM, 0.1 μM and 0.17 μM, respectively. Another possibility is MoS_2_ decorated PANI-rGO GCE [649]. Pulse voltammetry was utilized to characterize the sensor obtaining linear responses in a wide range from 5.0 to 500μM, 50 μM to 8.0 mM, and 1.0 to 500 μM, giving low detection limits of 0.70, 22.20, and 0.36 μM for DA, AA, and UA, respectively. Differential pulse voltammetry applied to ZnO nanowire arrays/Graphene foam modified electrodes was used to obtain higher sensitivity, leading to a LOD of 0.5, 5, and 0.5 μM for DA, AA and UA when detected simultaneously [650]. SnO_2_–rGO electrode was utilized to detect urea [651]. Amperometry show that the SnO_2_–rGO/GCE electrode is sensitive to urea in the concentration range of 0.016 to 3.9 pM, with a LOD as low as 11.7 fM.

Graphene is also used as material of choice for the detection of genetic substances important for the recognition of specific pathologies. DNA sensors can be based on different transduction techniques as electrochemistry electrochemiluminescence, fluorescence, surface plasmon resonance, colorimetry [652]. The electrochemical sensors possess a high sensitivity, are fast and low cost. The electrochemical detection mechanism is based on the change of the sensor response upon DNA sequence hybridization. Owing to their unique molecular structure and electronic properties graphene and its derivatives have captured much attention for the electrochemical sensing of genetic matter. In particular, it has been demonstrated that DNA or RNA interact with graphene with weak π-π interactions [653]. This allows tailoring the electronic properties of the graphene adsorbate thus optimizing the support effective signal amplification. Au NP decorated rGO was utilized to assemble an electrochemical sensor. This latter showed good selectivity towards single base DNA sequences, a linear behavior in the range 0.1 fM to 0.1 μM and a LOD of 35 aM [654]. Graphene/Au nanorods/polythionine was utilized to produce a biosensor for the detection of human papillomavirus [655]. The device showed unique sensing properties being able to detect the DNA papillomavirus in a range 1 × 10^−13^ to 1 × 10^−10^ mol/L with a LOD of 4.03 × 10^−14^ mol/L. In another work, graphene bifunctionalized with riboflavin 50-monophosphate sodium salt (FMNS) leading to high stability and sensitivity with a LOD of 8.3 × 10^−17^ M slightly higher than 7.4 × 10^−17^ M that of the conventional electrochemical indicator ([Fe(CN)_6_]^3−/4−^) [656]. Other researchers developed a GO/chitosan composite DNA sensor for the detection of the typhoid infection [657]. At this aim, a Salmonella single strand DNA was immobilized on the GO/Chitosan–ITO electrode using a glutaraldehyde crosslinker. The use of GO/Chitosan hybrid composite led to high sensing efficiency due to the fast electron transfer from the DNA and the ITO substrate. The biosensor was selective in the recognition of the complementary or non-complementary DNA sequences differing for only one base. The detection was linear in the range from 10 fM to 50 nM with a LOD of 10 fM within a response times of 60 s required for hybridization of the complementary sequence [657]. In addition, the detection may be performed in serum samples with a low LOD of 100 fM. The sensing activity may be reinforced by the integration of metallic NP in the device. The detection of a gene for the multidrug resistance was made using AuNPs/toluidine blue/GO electrode (see Figure 37) [658]. Under optimal conditions, the amperometric biosensor measured a current proportional to the logarithm of the target DNA concentration in the range from 1.0 × 10^−11^ to 1.0 × 10^−9^ M with a detection limit of 2.95 × 10^−12^ M. Moreover, the biosensor showed good selectivity and a reasonable stability and reproducibility.

In other sensors were utilized gold nanorods supported on GO as reported in [659]. The biosensor showed high selectivity being able to detect the complementary DNA sequence from a mixture with a 1000:1 proportion. In addition, the sensor had a linear behavior in a 1.0 nM to 10 fM with a LOD of 3.5 fM.

A list of different methods has been proposed to amplify the DNA detection signal. For example, in [660] is described the fabrication of a sensor for the detection of the human immunodeficiency virus (HIV) gene. Authors utilized an Au decorated graphene electrode using gold nanoclusters and applying an exonuclease III-assisted target recycling amplification method shown in Figure 38 obtaining a sensor with LOD of 30 aM.

Besides amplification, the biosensor showed high selectivity towards interfering sequences at concentrations 10 times higher than the HIV target. Recently an opposite route is gaining increasing interest: biosensensing activity is based on enhanced sensitivity obtained with the use of CRISPR-Cas9 to instead of amplification of the target DNA [661]. In conventional graphene FET based on complementary probe DNA, the sensitivity is limited by the concentration of the target sequences on the sensing electrode. An alternative method recently utilized, consists in a graphene-based field-effect transistor (FET) that uses clusters a matrix of short palindromic repeats (CRISPR) to enable the digital detection of a target DNA sequence. Without the need for amplification, the CRISP based technology in 15 min is able to generate a detection signal with a sensitivity of 1.7 fM. The graphene electrode is functionalized with a deactivated Cas9 CRISP (denoted with dRNP) complex which scans the whole genomic sample unzipping the double helix until finds the target sequence. Now Cas9 binds to the target sequence modulating the FET output. Two kinds of DNA were used, the first collected from HEK293T cell lines expressing blue fluorescent protein, the second derived from clinical samples and showing two distinct mutations at exons commonly deleted in individuals with Duchenne muscular dystrophy. The CRISP chip functionalized with the mutating Duchenne sequence produced an enhanced signal output only in presence of the complementary target.

In [662] the device sensitivity is amplified by a factor of 20,000 using a hairpin probe DNA allowing a recycling and hybridization of the target. The sensor was capable to sense a 21 mer target DNA at sub-fM concentration with superior selectivity towards mismatched oligomers. In this sensor the sensitivity is independent on the binding affinity of nucleic acids as in conventional sensors.

Besides genomic mutations, cancer is a disease with high socio-economical impact. Detection of cancer markers is then of high importance for the prevention, early detection, the diagnosis and the care. Pd decorated rGO electrochemical sensor were fabricated for a label free detection of the prostate specific antigen, a biomarker of the prostate cancer [663]. The biosensor showed a good sensitivity of 28.96 mA mL ng^−1^cm^−2^ with an LOD of 10 pg mL^−1^. Other metallic nanoparticles as Pd, Pt, Ag and Au were deposited on rGO sheets to fabricate a nanocomposite-based sensor for the detection of epidermal growth factor receptor ErbB2 useful for the breast cancer diagnosis [664] with a LOD varying in the range 1.0 fM to 0.5 μM upon the different metal nanoparticle used.

Reduced graphene oxide sheets were decorated with iron nitride nanoparticles (Fe_2_N) NPs produced by a solvothermal method and followed by a nitridation process [665]. The nanocomposite was screen printed to fabricate electrodes possessing high electrochemical sensing properties toward 4-nitroquinoline N-oxide (4-NQO) an important cancer marker. The sensor showed a linear response in the range 0.05–574.2 μM and nanomolar detection limit (9.24 nM). Finally, the biosensor was utilized to analyze human blood and urine samples showing recoveries close to 100%. In another work, an emulsion of GO and water-toluene were self-assembled in microspheres were used to fabricate an electrochemical sensor [666]. The GO microspheres were functionalized with folic acid and octadecylamine and then dried at 400 °C for 5 h. The microspheres of ~1.2 μm size are characterized by a porous structure densely functionalized with ordered alignment of folic acid groups allowing a high efficiency in capturing specific cancer cell. The sensor has a linear behavior in range of 5–10^5^ cell/mL with a LOD of 5 cells mL^−1^. The biosensor was able to selectively discriminate between HepG2, HeLa and A549 model cancer cell lines and L02 cell line used as a control. The efficiency in the selective detection of cancer cells was attributed to the overexpression of folic acid on the cancer cell membrane. Another recent example is a cancer biomarker detection based on sulfur-doped reduced graphene oxide (SrGO) product fabricated using an eco-friendly biomass precursor [667]. The functionalized graphene is used to assemble a highly sensitive electrochemical sensor for detecting of 8-hydroxy-2′-deoxyguanosine (8-OHdG) molecules, which are an important biomarker for oxidative stress, cardiovascular diseases, and cancers. The biomarker displayed an excellent electrochemical sensitivity attributed to the strong electron-donating ability of sulfur facilitating the electron transfer to the biomolecules in the redox reactions. The device possesses a sensitivity of ~1 nM, a 20–0.002 μM wide detection window, good selectivity, stability and reproducibility (see Figure 39). Finally, in optimized experimental conditions, the biosensor showed remarkable recoveries for the detection of 8-OHdG biomarker.

Decoration of graphene with metal NPs is another effective route to improve the detection of cancer biomarkers. Among others, several studies have been carried out to study the graphene/AuNPs electrochemical response for various cancer biomarkers. In such work a three-dimensional graphene/gold NP composite was fabricated for a selective detection of the cytokeratin 19 fragment antigen 21-1 (CYFRA 21-1) [668]. The label-free electrochemical immunosensor was capable to selectively detect the tumor marker CYFRA21-1 in non-small cell lung cancer. Briefly, the anti-CYFRA21-1 was immobilized on the Graphene/AuNP modified GCE through the cross-linking of chitosan (CS), glutaraldehyde (GA).

The electrochemical immunosensor was employed to detect CYFRA21-1 and exhibited a wide linear range of 0.25–800 ng/mL and low detection limit of 100 pg/mL. Other nanoparticles as cobalt sulfide [669] (detection of CA15-3 antigen in wide linear range of 0.1–150 U mL^−1^ and a low detection limit of 0.03 U mL^−1^), Pd [670] (prostate-specific antigen with sensitivity 28.96 μA mL ng^−1^ cm^−2^ corresponding to 10 pg mL^−1^), oxides as ZrO_2_ [671] (CYFRA-21-1 biomarker with sensitivity 0.756 μA mL ng^−1^ and a remarkable lower detection limit of 0.122 ng mL^−1^), Fe_3_O_4_ [672] (detection of prostate specific antigen or prostate specific membrane antigen with excellent LOD of 15 fg/mL for PSA and 4.8 fg/mL for PSMA were achieved in serum).

A few words must be spent for describing the fabrication of sensors for heavy metal ions which are known to be dangerous for the human health. Electrochemical sensors based on graphene and graphene-based materials are characterized by high sensitivity and selectivity and high stability, rapid response time and low cost thus resulting useful for heavy metal ions detection. There are several review papers dedicated to the description of graphene-based nanomaterials for the detection of some important heavy metal ions [673,674,675]. In [676] are reviewed graphene-based sensors based on Cu^2+^, Pb^2+^, Hg^2+^, Cd^2+^; Ag^2+^. The use of graphene-based nanomaterials resulted in an improvement of the performances of heavy metal ion sensors. From a survey of the literature the heavy metal ions are detected using graphene based sensors and different electrochemical techniques as differential pulse voltammetry (DPV), differential pulse stripping voltammetry (DPSV), differential pulse anodic stripping voltammetry (DPASV), square wave anodic stripping voltammetry (SWASV) obtaining different degrees of detection limits [673]. In the case of Cd(II) the LOD ranged from 0.01 μg/L to 0.6 μg/L using DPV, DPASV or SWASV, while in other works and using different concentration units, LOD ranged from 0.19 nM to 3 nM. Pb(II) is detected using DPV, DPASV, SWASV obtaining LOD ranging from 0.02 μg/L to 0.41 μg/L, other authors found a LOD ranging from 0.6 nM to 1.41 nM using DPV, SWASV. As(III) was detected using DPV obtaining a LOD from 2.3 nM. Hg(II) was detected using DPASV resulting in a LOD of 0.277 nM. Cu(II) ions were detected using DPASV, SWASV, with LOD in the range 0.16 nM—0.52 nM or LOD = 0.02 μg/L. Zn(II) ions were detected using DPASV and SWASV obtaininf a LOD in the range 0.1–0.18 μg/L. Data regarding the sensitivities reached using the various techniques are summarized in the Table 6, while more information may be found in the given references.

Toxic substances as pesticides, herbicides and insecticides used to protect plants in agriculture as well as those produced in industrial processes and toxic gases developed by human activities need a systematic control for societal safety.

Graphene based sensors offer an option for the sensitive detection of agro-hazardous pesticide residues in food [677,678,679,680], and to control the quality of the water and air to avoid dangerous effects to humans. Electrochemical sensors play also a significant role in the industry and in everyday life to control toxic substances. For example, controlling the presence of 4-nitrochlorobenzene a toxic organic pollutant is highly desirable. Graphene based electrochemical sensors are the low cost, highly sensitive and offer ease of operation for the detection of toxix substances as: 4-nitrochlorobenzene [681] (LOD = 10 nM), for the detection of hydrazine [682], sulfides [683], 4-aminophenol [684], diphenolic compounds [685], hydroquinone [686]. In addition, toxic gaseous species can be detected as reported in [687] and example are H_2_ [688] (LOD: 20 ppm), CO [689] (LOD: 0.25 ppm), CO_2_ [690] (sensitivity 0.17%/ppm), NO [691] (sensitivity: 63.65 pm/ppm), NO_2_ [692] (LOD: 0.5 ppm) SO_2_ [693] (LOD: 0.5 ppm), ammonia [694,695]. More information may be found in the review on the toxic gas detection based on graphene metal-oxides [696]. 

Graphene is also utilized for fluorescence detection of substances. Among available techniques, Főrster resonance energy transfer (FRET) may be utilized to localize substances with a resolution of 10 nm. With respect to FRET, carbon nanostructures quench fluorescence over the entire visible region, making them efficient acceptors in FRET. Graphene-sensing is based on fluorescence quenching and several molecules as glucose, H_2_O_2_, dopamine, food toxins and metal ions, can be detected as well as nucleic acids, pathogens and toxins. More information may be found in [702,703,704].

## 8. Conclusions

Among the elements of the Periodic Table carbon appears as a very special element possessing unique properties and this is the reason why carbon is at the base of the organic chemistry and, in the end, of the life. Among these special properties is the carbon capability to generate rather different structures displaying peculiar properties enabling the development of new biosensors for the detection of a wide variety of biological, organic and inorganic analytes with desirable levels of sensitivity, low detection limits and good stability. The possibility to integrate the carbon nanostructures in other materials as organic polymers or hydrogels or the coupling to metal or oxide nanoparticles constitutes a possible route for increase the detecting possibilities of the sensors or enhance its performances. Moreover, the carbon-based nanostructures may be chemically assembled in 2D and 3Dstructures further providing attributes which improve the sensing capabilities. Finally, carbon nanostructures may be integrated in nano- or micro-scale devices such as microfluidic chips or microelectronic devices which appear particularly promising for the fabrication of devices for the multiple detection of analytes for the miniaturization of the sensors allowing a diffuse sensing to control for example the environment. These advances in biosensor research open interesting perspective for the development of promising platforms for the detection and control of biomarkers, food quality, toxic substances, pollution, air and environment monitoring.

## Figures and Tables

**Figure 1 nanomaterials-11-00967-f001:**
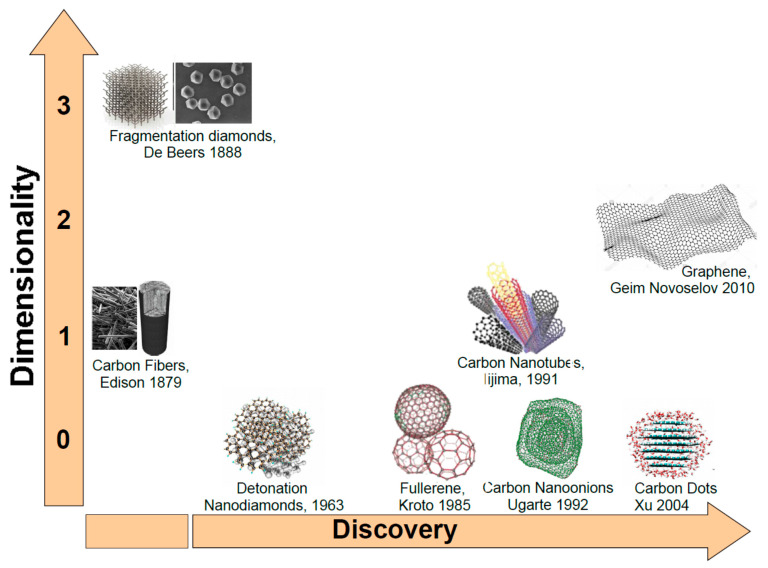
Carbon nanostructures ordered following dimensionality and discovery time.

**Figure 2 nanomaterials-11-00967-f002:**
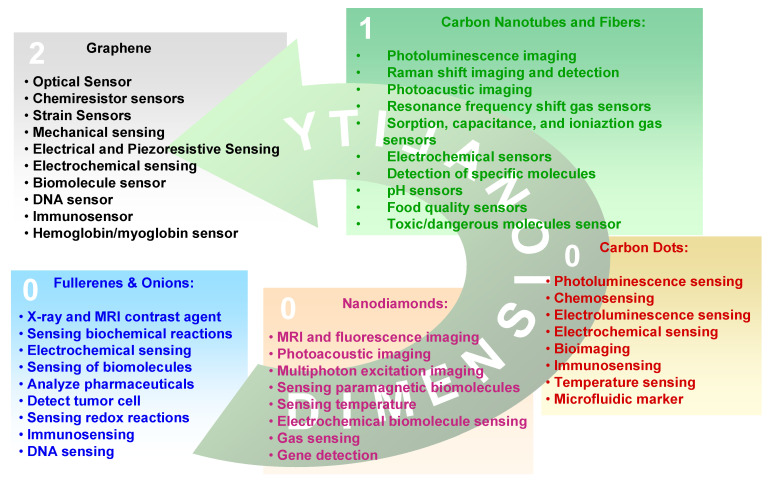
Carbon nanoallotropes and the relative sensing applications.

**Figure 3 nanomaterials-11-00967-f003:**
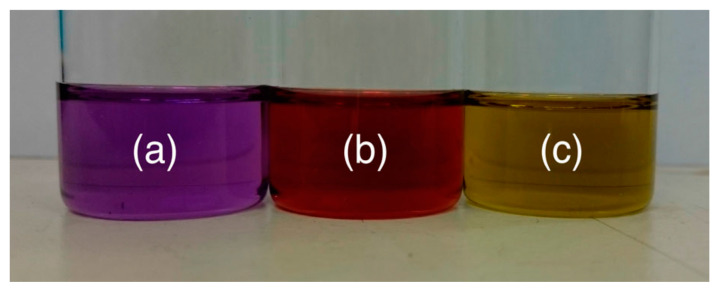
The absorption spectra of the fullerenes change as the size of the conjugated system increases: with slight variations depending on solvent, solutions of C_60_ are an intense purple color (**a**), C_70_ red like wine (**b**) and C_84_ a green-yellow (**c**) (all solutions are in toluene). Generally speaking, the gap between the highest occupied molecular orbital and the lowest unoccupied molecular orbital decreases with increasing cage size leading to optical absorptions of lower energy, i.e., longer wavelength. Reprinted with permission from [65].

**Figure 4 nanomaterials-11-00967-f004:**
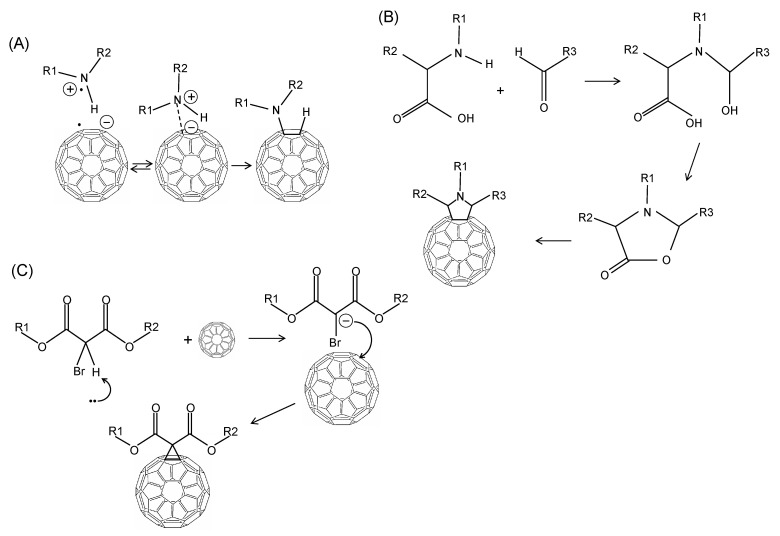
(**A**) Amination mechanism of C_60_. The first step is single-electron transfer to produce the C_60_ anion radical. Then radical recombination gives a zwitterion which can be stabilized by proton transfer to give the final product. (**B**) Prato reaction mechanism. The nitrogen lone pair of the amino acid attacks the aldehyde/ketone’s polar carbonyl group and leads to the expulsion of water, before decarboxylation gives the reactive ylide intermediate. (**C**) Bingel reaction mechanism. Deprotonation by the strong base gives the nucleophilic malonate anion which then attacks the [6,6] bond, Br is then expelled.

**Figure 5 nanomaterials-11-00967-f005:**
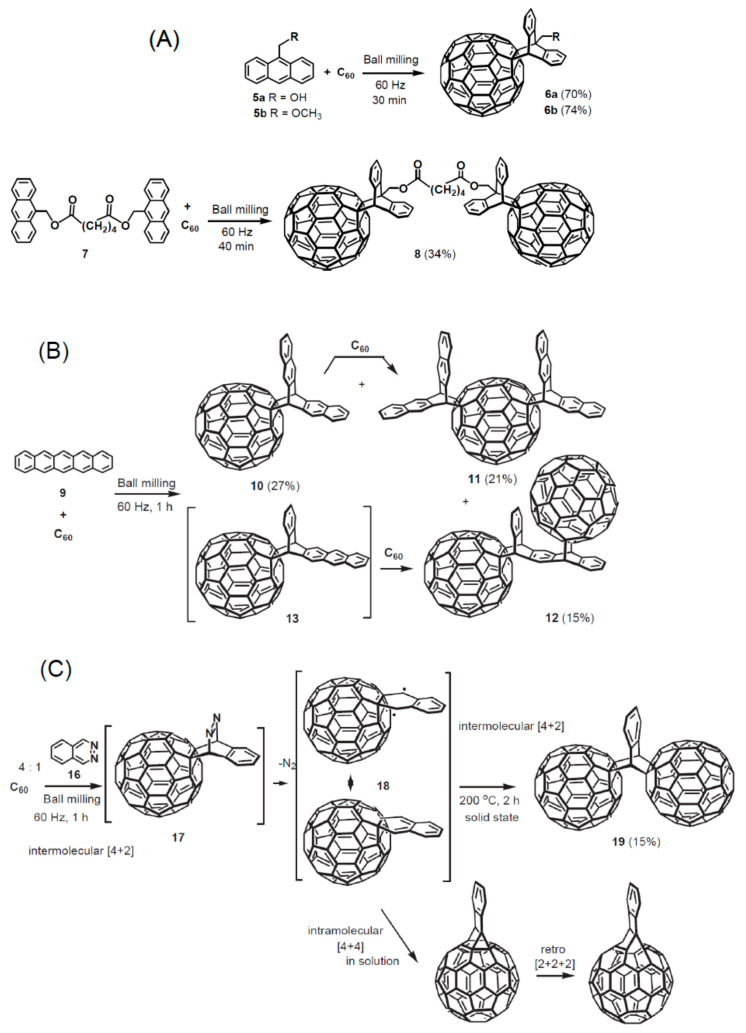
(**A**) Diels–Alder reactions of fullerene with 9-substituted anthracenes, (**B**) reaction of fullerene and pentacene, (**C**) reaction of fullerene and phthalazine. Reprinted with permission from [120]. Copyright 2016 Elsevier.

**Figure 6 nanomaterials-11-00967-f006:**
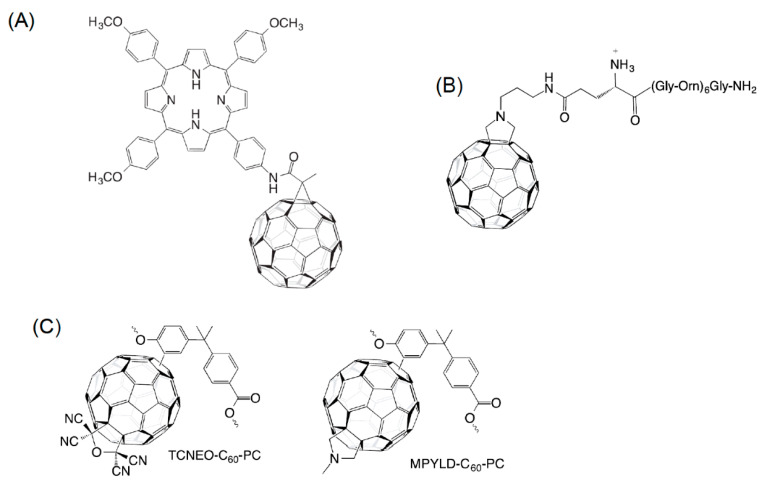
(**A**) Molecular structure of PS porphyrin-C60 dyad; (**B**) fullerene peptide fabricated via solid-phase peptide synthesis from amino functionalized fullerene and N-Fmoc-L-glutamic acid a-tert-butyl ester. (**C**) Polycarbonate containing fullerene derivatives. Reprinted with permission from [120]. Copyright 2017 Elsevier.

**Figure 7 nanomaterials-11-00967-f007:**
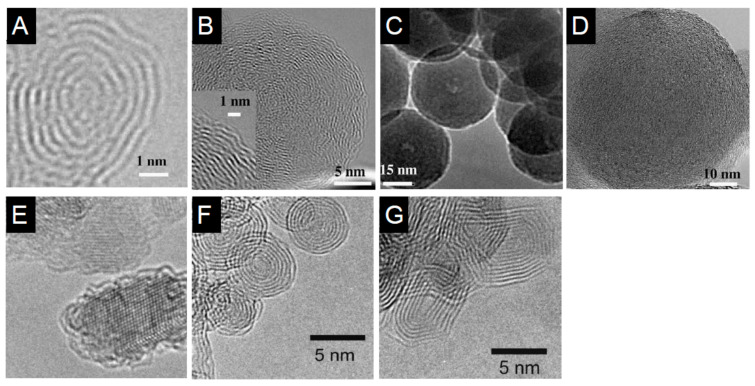
HRTEM images of CNOs produced by a counter-flow diffusion flame process as a function of the methane precursor concentration: (**A**) CH_4_ = 15%, (**B**) CH_4_ = 25%, (**C**) CH_4_ = 35%, and (**D**) CH_4_ = 45% (adapted with permission from [123]. Copyright 2009 Elsevier.). Nano-diamond to CNOs transformation as a function of the temperature. HRTEM images of (**E**) diamond nanoparticles starting material, (**F**) spherical carbon onions, and (**G**) polyhedral carbon onions: Diamond nanoparticles are transformed into spherical onions at about 1700 °C. Polyhedral onions are dominant in the sample annealed above 1900 °C (adapted with permission from [130]. Copyright 2001 AIP Publishing).

**Figure 8 nanomaterials-11-00967-f008:**
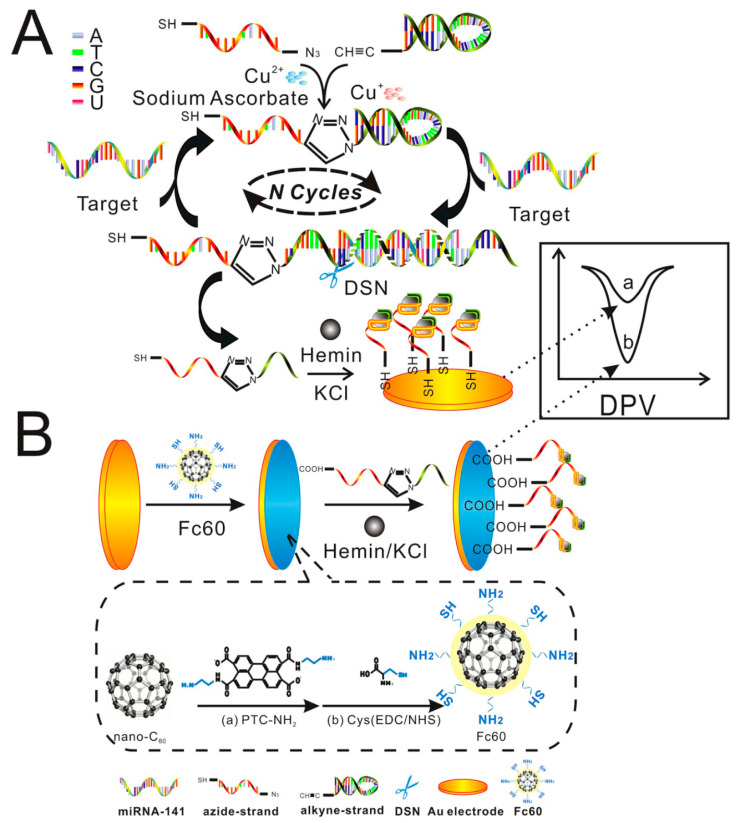
The schematic diagram of the EATR (**A**) and FC60 dual-amplified strategy electrochemical biosensor (**B**) (reproduced with permission from [164]. Copyright 2020 Elsevier.).

**Figure 9 nanomaterials-11-00967-f009:**
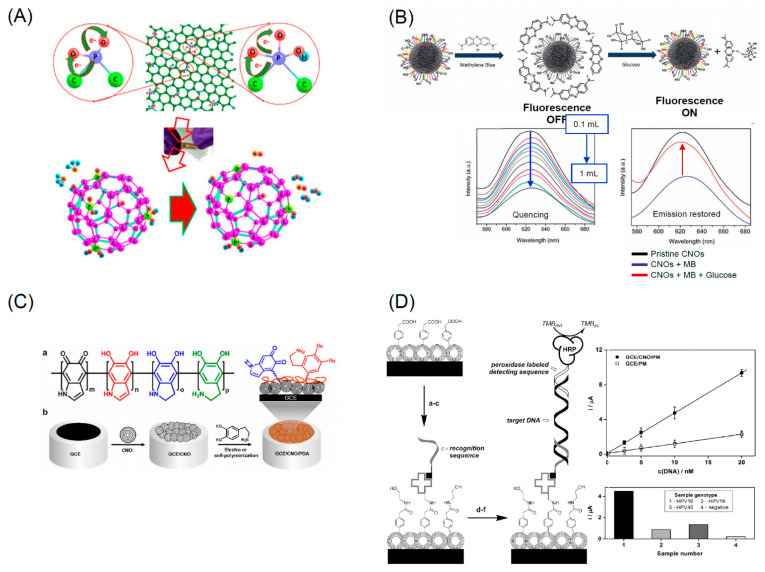
(**A**) electrochemical sensor based on CNOs: charge transfer between CNOs and NH_3_ molecules results in a change of the sensor resistance thus enabling detection of ammonia gas [183]. (**B**) Glucose sensor based on the optical properties of CNOs: addition of methylene blue quenches the natural emission from CNOs. Glucose molecules bind to MB thus reducing the interaction with CNOs and removing the quenching [185]. (**C**) Structure of polydopamine, and modification strategy of a glassy carbon electrode with polydopamine functionalized CNOs. Polydopamine voltammograms change as the pH changes thus enabling pH detection with potentiometric measurements [184]. (**D**) Fabrication of carcinoembryonic antigen-immunosensor based on a glassy carbon electrode modified with oxidized CNOs. Diazonoum salt was then used to functionalize the CNOs and sequentially immobilize streptavidin, biotinylated DNA, and target DNA. The hybridization of the target DNA with the DNA sequence of the papilloma virus can be detected by amperometric measurements [190]. Reprinted with permission from [183,184,185,190]. Copyright: 2019 American Chemical Society; 2018 Elsevier, 2016 RSC Publishing and 2009 Elsevier respectively.

**Figure 10 nanomaterials-11-00967-f010:**
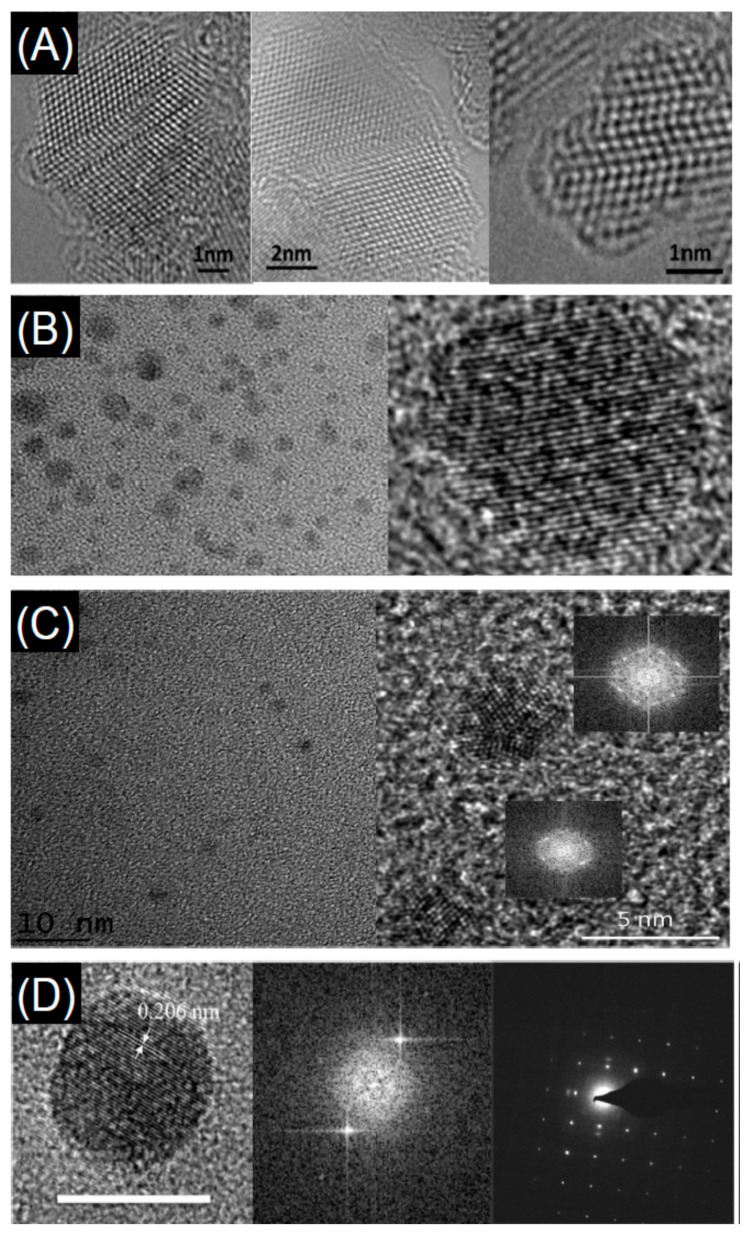
NDs synthesized (**A**) by detonation; (**B**) by pulsed laser deposition in ethanol; (**C**) by plasma in liquid ethanol; (**D**) by CVD plasma. Reprinted with permission from [201,202,209,210]. Copyright: 2017 American Chemical Society, 2019 Elsevier, 2018 Elsevier and 2016 Springer Nature respectively.

**Figure 11 nanomaterials-11-00967-f011:**
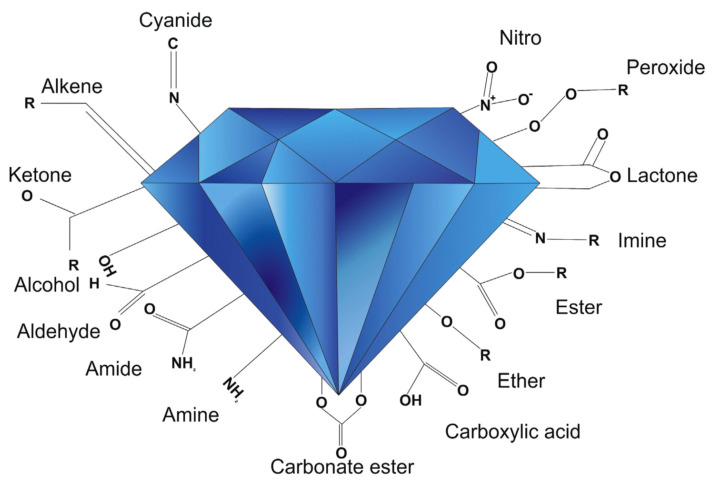
Modification of the surface chemistry of NDs with various functional groups. Reproduced with permission from [228]. Copyright 2019 Elsevier.

**Figure 12 nanomaterials-11-00967-f012:**
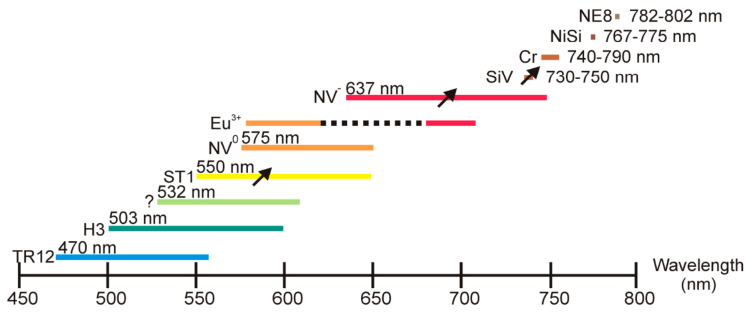
Spectral map of various single emitters in diamond. For centers with emission wavelengths shorter than 730 nm, the length of the colored line represents the approximate width of the emission spectrum of the color center including phonon sidebands. The wavelength given for each center denotes the zero-phonon-line (ZPL) wavelength. For centers with emission wavelengths above 730 nm, the labeled wavelength indicates the spread of the observed ZPL positions, while the colored line indicates the width of the ZPL for a single center. Note that especially in the near the infrared region, a multitude of unidentified color centers exists, which are not given here for clarity. Centers for which spin manipulation of single centers has been shown are labeled with a black arrow. Reprinted with permission from [238]. Free download.

**Figure 13 nanomaterials-11-00967-f013:**
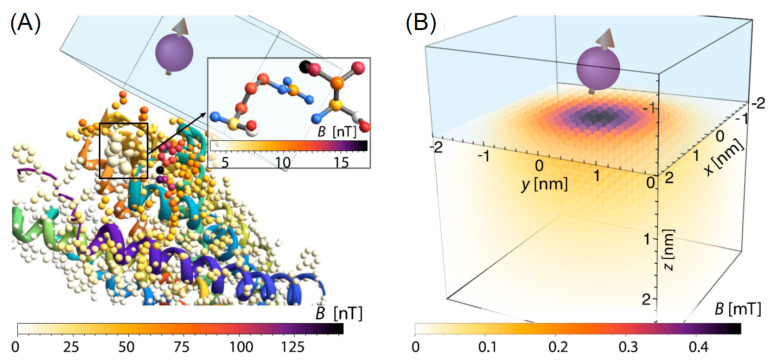
(**A**) Nuclear spin imaging with a shallow NV center in diamond. A single NV spin (purple) at 1–2 nm from the diamond surface can sense single nuclear spins in a molecule (the chemokine receptor CXCR4, ribbon diagram) anchored to the diamond. The magnetic field produced by individual ^13^C (spheres with color scale given by B^j^_⊥_/γe) is in the range of nT, within reach of NV sensitivity. In the inset, we show the binding site of interest [atoms other than 13C are blue (O) and red (N)]. (**B**) A shallow NV center (2 nm from the surface) creates a magnetic field gradient [A(r) = γ_n_] above the [111] surface of the diamond. Note the azimuthal symmetry of the field, which causes degeneracy of the frequency shift at many spatial locations. Reprinted with permission from [244].

**Figure 14 nanomaterials-11-00967-f014:**
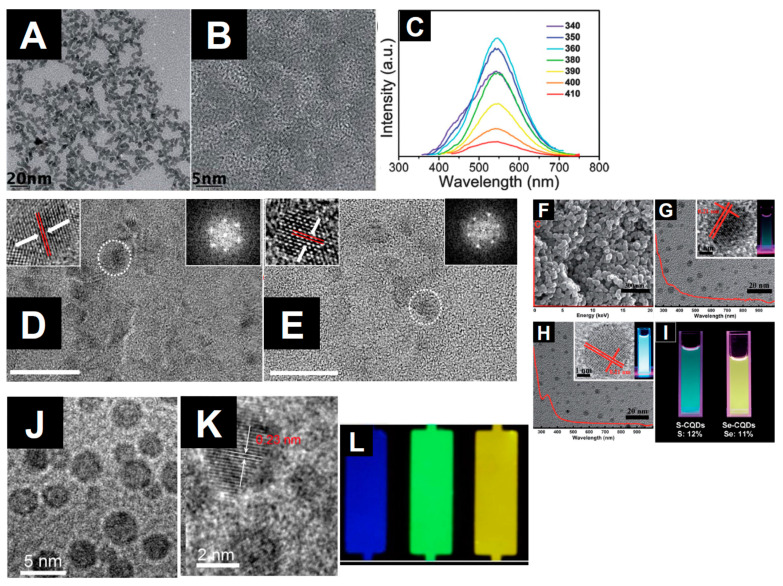
(**A**) TEM and (**B**) HRTEM images of the CQDs produced by exfoliation of a graphite rods in NaOH and deposited on freshly cleaved mica substrates. (**C**) PL spectra of the CQD aqueous solution at different excitation wavelengths. Reprinted with permission from [267]. Copyright 1991 Royal Society of Chemistry. (**D**) HR-TEM images of (**E**) pristine CQDs and N-doped CQDs produced by pulsed ND:YAG laser ablation of graphite flake suspension in a mixture of diethylenetriamine and ethanol. Insets show the high-quality crystalline hexagonal patterns and diffraction patterns of the CQDs and N-doped CQDs. Reproduced with permission from [270]. Copyright 2020 Elsevier. (**F**) FESEM image and overlapped in red EDS spectrum of CQDs from Chinese ink. (**G**,**H**) TEM images of O- (oxidized) and N- (nitrogen doped) CQDs respectively. Inset shows the high resolution of O-CQDs and N-CQDs with indicated the lattice spacing while overlapped in red are the UV-vis absorption spectra; (**I**) The digital images of S-CQDs and Se-CQDs emission under UV-light irradiation. Reproduced with permission from [266]. Copyright 2013 Royal Society of Chemistry. (**J**,**K**) TEM images of the Trp-CQDs of CQDs produced by reacting glucose with different kinds of amino acids and with microwave thermal treatment and (**L**) different emission obtained from Trp-CQDs, Leu-CQDs, and Asp-CQDs characterized by blue, green, and yellow emission. Reprinted with permission from [274]. Copyright 2014 Springer Nature.

**Figure 15 nanomaterials-11-00967-f015:**
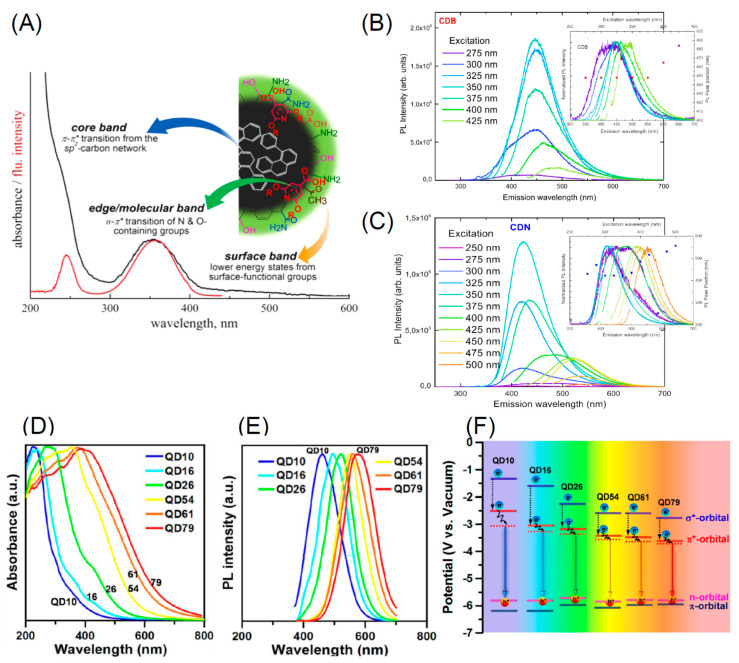
(**A**) depicts the collective explanation for the origin of PL in CDs, invoking core, edge, and surface bands. Reprinted with permission from [291]. Copyright 2017 American Chemical Society. (**B**) excitation PL spectra of pristine CQDs and (**C**) nitrogen doped CQDs. Insets show the normalized PL spectra to highlight the emission peak position on the excitation wavelength. Reprinted with permission from [292]. Copyright 2018 American Chemical Society. (**D**) normalized absorption and (**E**) normlalized PL as a function of the CQDs size. (**F**) dependence of the HOMO and LUMO energy levels with the CQD size. Reprinted with permission from [293]. Copyright 2016 American Chemical Society.

**Figure 16 nanomaterials-11-00967-f016:**
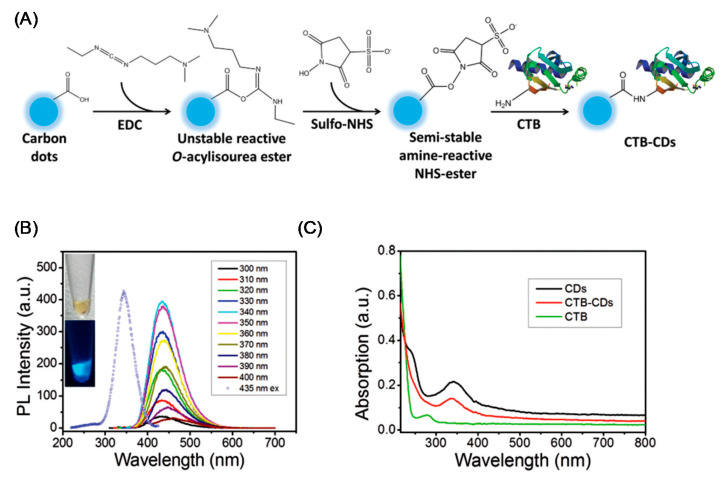
Preparation and characterization of CTB–CQDs conjugates. (**A**) Schematic illustration of the synthesis of CTB–CQDs conjugates using the EDC/NHS reaction: 1-ethyl-3-(3-dimethylaminopropyl) carbodiimide hydrochloride (EDC) was used to react with the carboxyl groups on the surface of CDs, forming an unstable reactive O-acylisourea intermediate. Subsequently, N-hydroxysulfosuccinimide (Sulfo-NHS) was added to convert it to a semi-stable amine-reactive NHS-ester. The amine groups of CTB then covalently conjugated with the NHS-ester modified CDs to produce CTB–CDs. (**B**) photoluminescence (PL) spectra of CTB–CDs (Ex: excitation; Em: emission) and insets show photographs of CTB–CDs in aqueous solution under visible (upper) and UV (lower) light. (**C**) UV-vis absorption spectra of CDs, CTB, and CTB–CDs Reprinted with permission from [328]. Copyright 2009 RSC Publications.

**Figure 17 nanomaterials-11-00967-f017:**
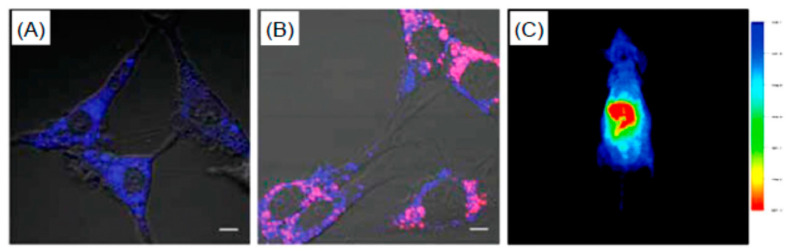
(**A**) UV-vis absorption spectra of CQDs, CTB, and CTB–CQDs; In vitro and in vivo bioimaging of CTB–CQDs: (D) NGF-treated PC12 cells were labeled with CTB–CQDs; (**B**) CTB–CQDs and Lysotracker; (**C**) CTB–CQDs were injected on the back of BALB/c nude mice. Reprinted with permission from [328]. Copyright 2009 Royal Society of Chemistry.

**Figure 18 nanomaterials-11-00967-f018:**
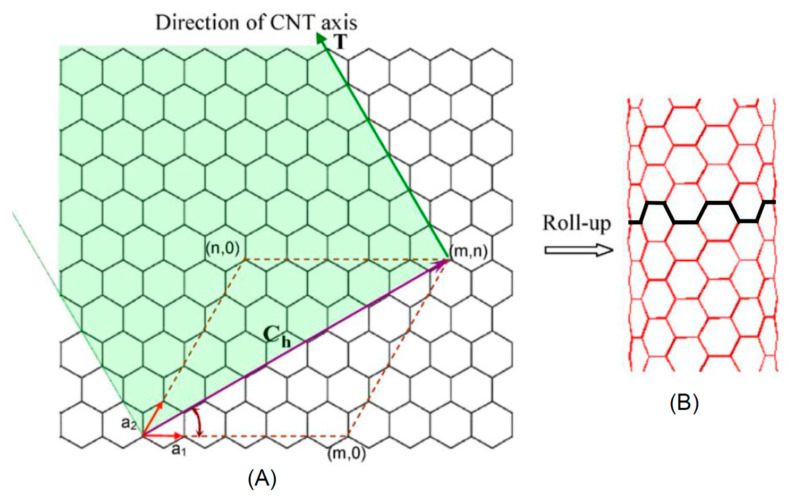
Roll-up vector defining the structure of carbon nanotubes. (**A**) Graphene lattice and (**B**) carbon nanotube along the indicated axis leads to an armchair CNT. Reprinted with permission from [342]. Copyright 2006 Elsevier.

**Figure 19 nanomaterials-11-00967-f019:**
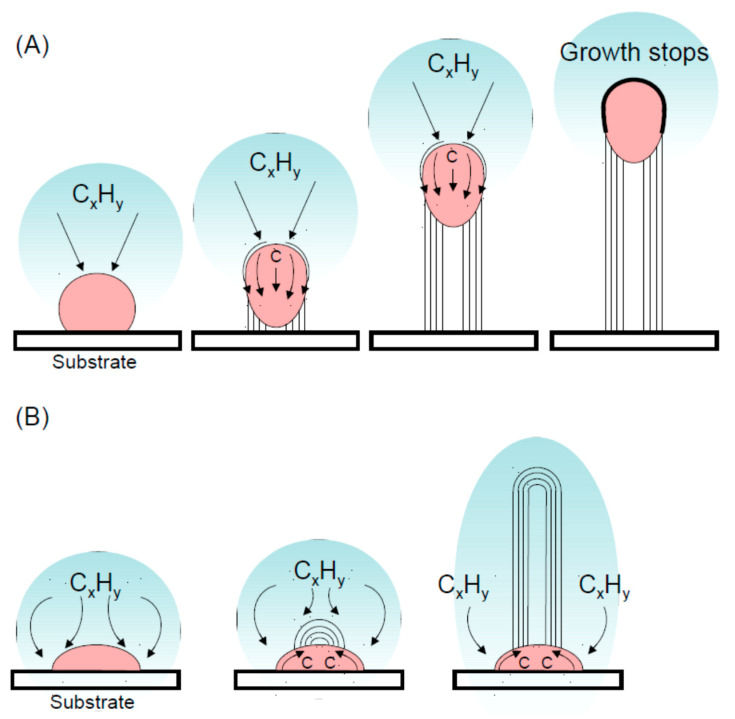
Widely-accepted growth mechanisms for CNTs: (**A**) tip-growth model, (**B**) base-growth model.

**Figure 20 nanomaterials-11-00967-f020:**
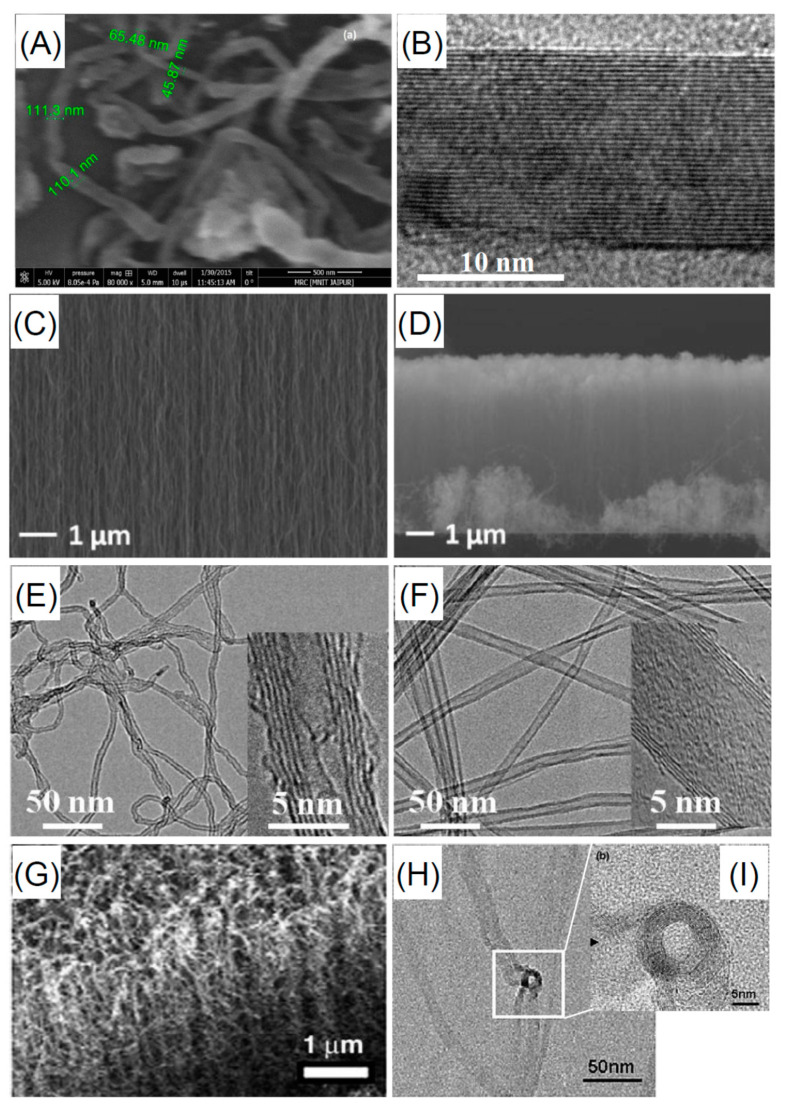
SEM (**A**) and HRTEM (**B**) images of MWCNTs synthesized by arc-discharge. Reproduced with permission from [370]. (**C**) Cross-sectional SEM images of vertically aligned CNTs grown on different buffer layers at 600 °C of ALD Al_2_O_3_ and (**D**) of SiO_2_ while (**E**,**F**) are the correspondent TEM images. Reproduced with permission from [371]. (**G**) A SEM image of CNT pillars on Si produced by pulsed lased deposition and (**H**) a TEM image of a junction connecting two CNTs and (**I**) its HRTEM magnification. Reprinted with permission from [372]. Copyright 2015 Taylor&Francis, Copyright 2019 Springer, Copyright, Copyright 2005 Elsevier.

**Figure 21 nanomaterials-11-00967-f021:**
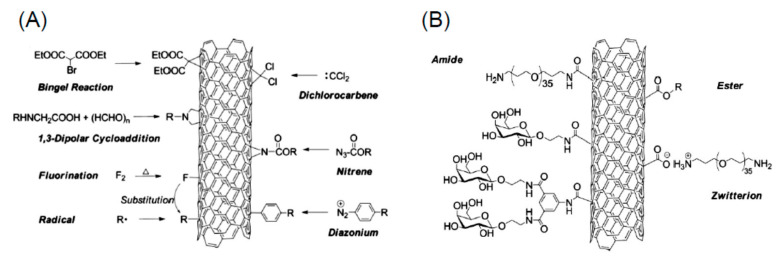
(**A**) Covalent addition reactions on the sidewall of carbon nanotubes. (**B**) Reactions targeting carboxylic acids (derived from nanotube surface defects). Adapted with permission from [379]. Copyright 2009 John Willey and Sons.

**Figure 22 nanomaterials-11-00967-f022:**
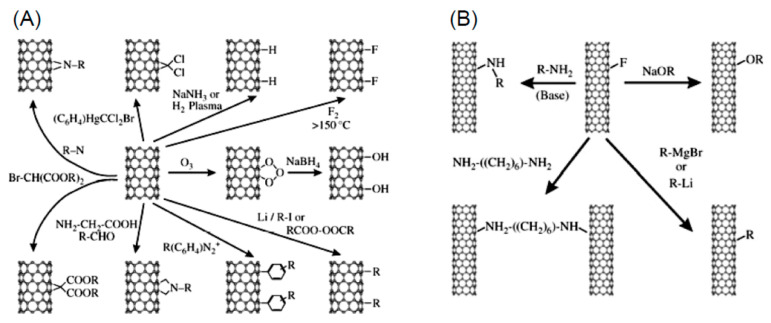
(**A**) Overview of possible addition reactions for the functionalization of the nanotube sidewall. (**B**) Functionalization of the sidewall through nucleophilic substitution reactions in fluorinated nanotubes. Reproduced with permission from [391]. Copyright 2005 John Wiley and Sons.

**Figure 23 nanomaterials-11-00967-f023:**
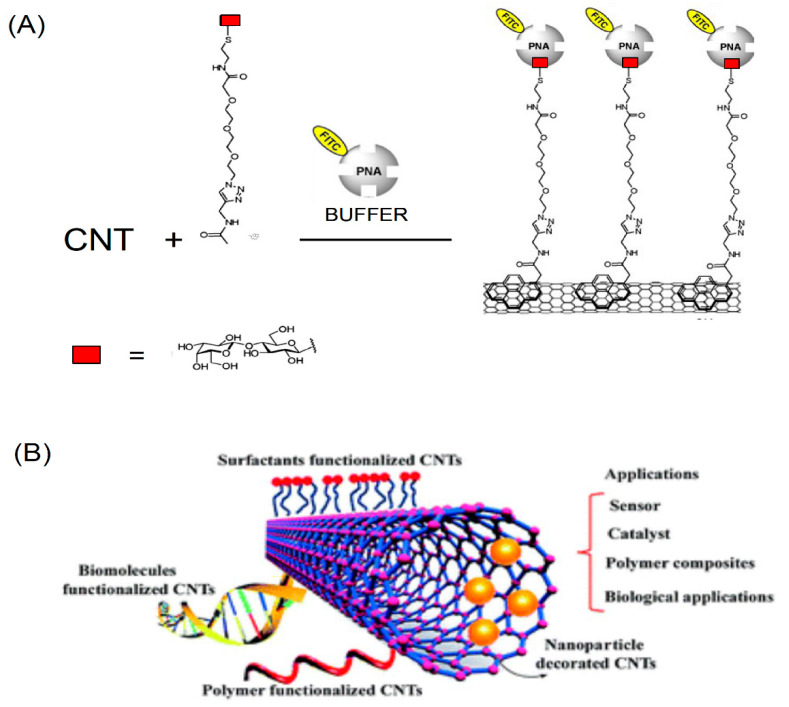
(**A**) π-π interaction used to bind a lactose-specific receptor such as the Arachis hypogaea Peanut agglutinin (PNA) for selective recognition of lactose. Reproduced with permission from [401]. Copyright 2005 Royal Society of Chemistry; (**B**) adsorption functionalization of CNTs and applications. Reprinted with permission from [402]. Copyright 2009, 2016 Royal Society of Chemistry.

**Figure 24 nanomaterials-11-00967-f024:**
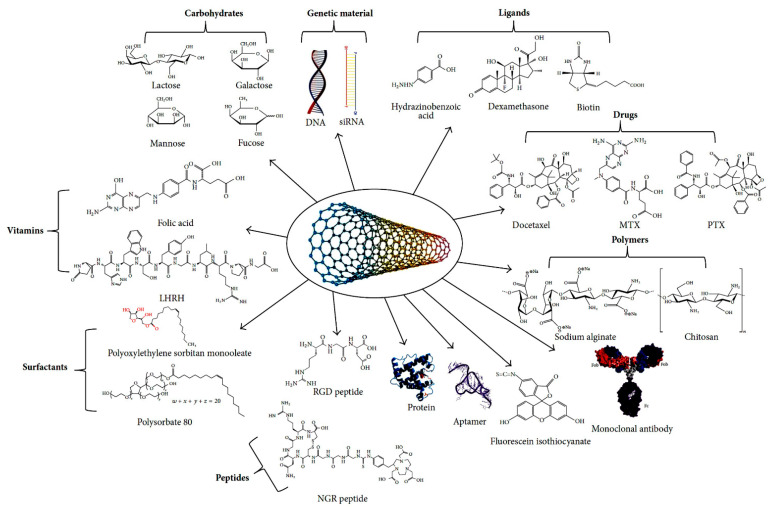
Non-covalent biofunctionalization of CNTs. Reprinted with permission from [414]. Copyright 2017 Hindawi.

**Figure 25 nanomaterials-11-00967-f025:**
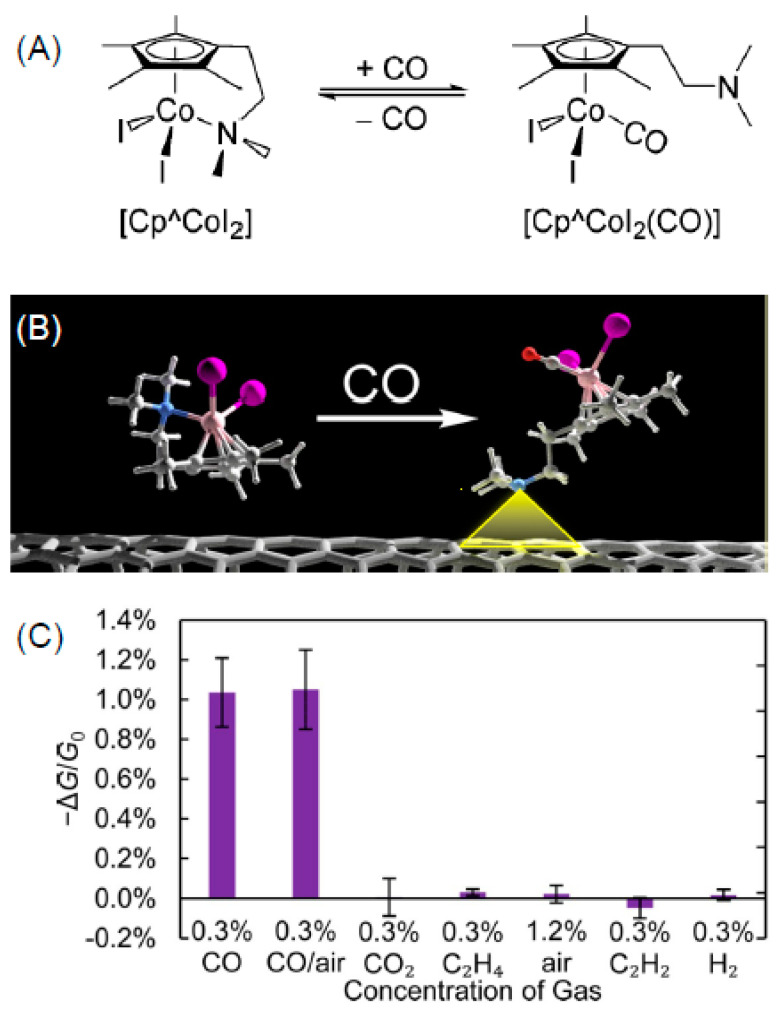
(**A**,**B**) reversible CO Binding to [Cp^CoI_2_]. (**C**) Average conductance changes of four chemiresistors in response to 60 s exposures to various gases diluted in N_2_ unless otherwise indicated. Error bars represent 1 standard deviation across the four devices. Reprinted with permission from [450]. Copyright 2016 American Chemical Society.

**Figure 26 nanomaterials-11-00967-f026:**
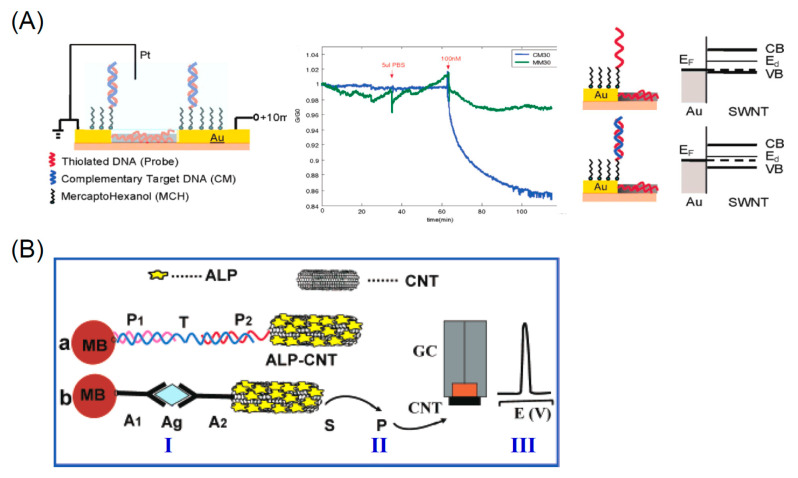
Detection modalities in CNT based sensors. (**A**) Left: schematic illustration of a single device during electrical measurement. Complementary ssDNA oligos hybridize to thiolated ssDNA coimmobilized with MCH on the gold electrodes. Center: the electrical conductance of a SWNT functionalized ssDNA capturing exhibits almost no change upon mismatch while shows a marked decrease when hybridized with the complementary target DNA. Right: schematic illustration of energy level alignment before and after DNA hybridization. Reprinted with permission from [487]. Copyright 2006 American Chemical Society (**B**) Schematic representation of the analytical protocol: (I) Capture of the ALP-loaded CNT tags to the streptavidin-modified magnetic beads (a) by a sandwich DNA hybridization or (b) Ab-Ag-Ab interaction. (II) Enzymatic reaction. (III) Electrochemical detection of the product of the enzymatic reaction at the CNT-modified glassy carbon electrode. Reprinted with permission from [488]. Copyright 2004 American Chemical Society.

**Figure 27 nanomaterials-11-00967-f027:**
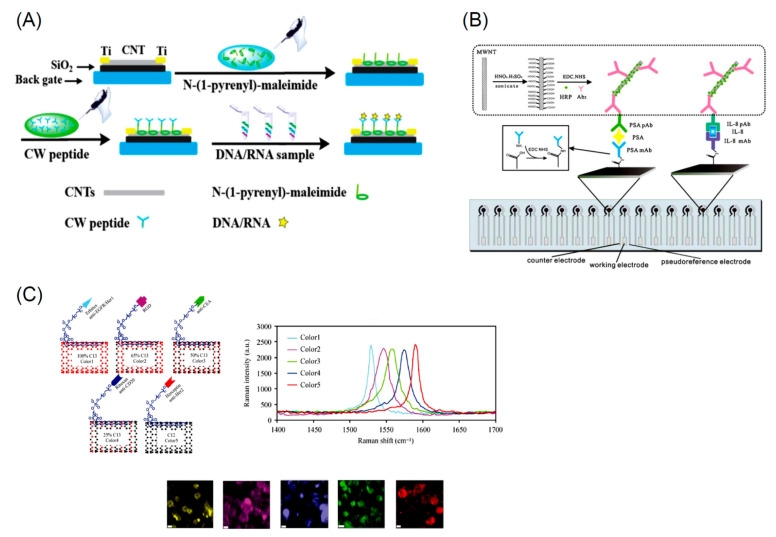
(**A**) Schematic representation of the novel peptide based CNT biosensor. Reprinted with permission from [490]. Copyright 2019 The Royal Society of Chemistry. (**B**) Schematic demonstration for the “sandwich” type strategy electrochemical immunosensor. A 16 channel screen-printed carbon electrode array was employed as the detection platform. MWCNT on the working electrode, were oxidized and functionalized with EDC/NHS + the capture antibodies PSA mAb or IL-8. The bonds between the target antigen (PSA or IL-8) to the antibody are recognized with an amperometric readout. Reprinted with permission from [498]. Copyright 2011 Elsevier. (**C**) Left: SWNTs with different Raman “colors”. Color 1, 2, 3, 4, and 5 represent SWNTs with C^13^ isotope percentages of 100%, 65%, 50%, 25%, and 0%, respectively. Right: Raman spectra of the five different SWNT samples where the shift of SWNT Raman G-band peak is clearly dependent on the C^13^/C^12^ ratio in the SWNTs. Bottom: Five-color Raman imaging of cancer cells with different surface protein expression profiles were successfully labeled by SWNT tags and detected using Raman imaging (scale bars = 10 μm). Reprinted with permission from [499]. Coptright 2010 Springer.

**Figure 28 nanomaterials-11-00967-f028:**
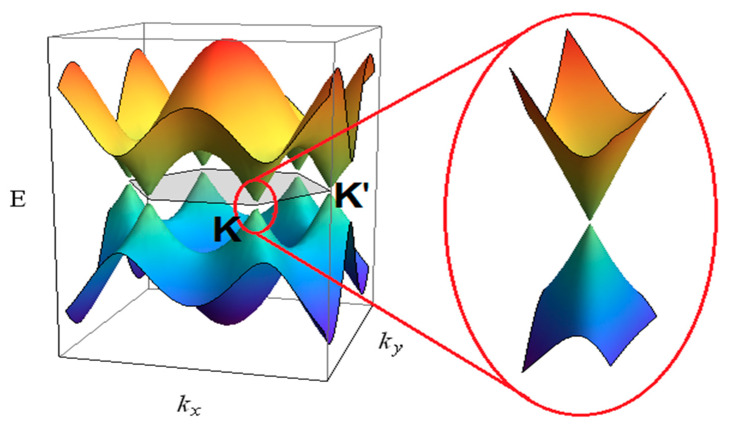
Band structure of graphene as a function of the momentum k_x_ and k_y_. One notices the valence band (at lower energy in blue) and the conduction band (at higher energy in orange). A zoom-in shows the bands close to the point where they touch each other in the Dirac point.

**Figure 29 nanomaterials-11-00967-f029:**
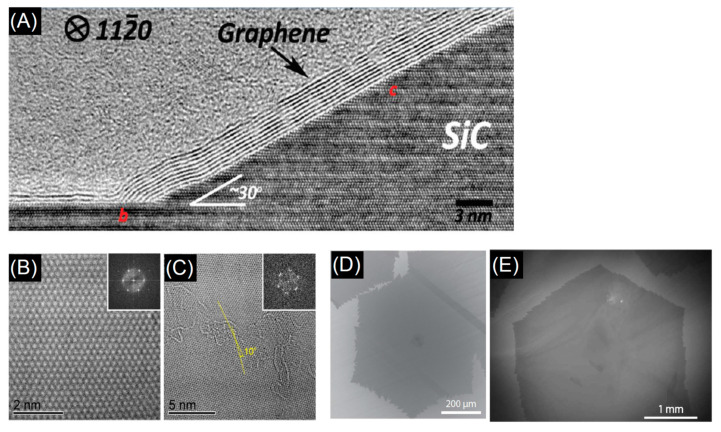
(**A**). TEM micrographs of epitaxially grown graphene on the (110n) plane of SiC at 1425 °C. Layer thickness appears not to be self-limiting, evidenced by the many-layer thickness (approximately 8) shown. Adapted with permission from [533]. Copyright 2010 American Chemical Society. (**B**) HR-TEM image of atomic structures of low temperature CVD grown graphene film and its corresponding diffraction patterns. (**C**) HR-TEM image near the graphene grain boundary. The misoriented angle between graphene domains is within 10 degrees and is seen as two adjacent diffraction patterns. (**B**,**C**) reproduced with permission from [534]. Large single crystal graphene produced inside a copper “pocket”. (**D**) SEM image of an isolated graphene crystal with a diagonal dimension of nearly 750 μm; (**E**) SEM image of a CVD graphene single grain with a diameter of over 3 mm. (**D**,**E**) reproduced from [535].

**Figure 30 nanomaterials-11-00967-f030:**
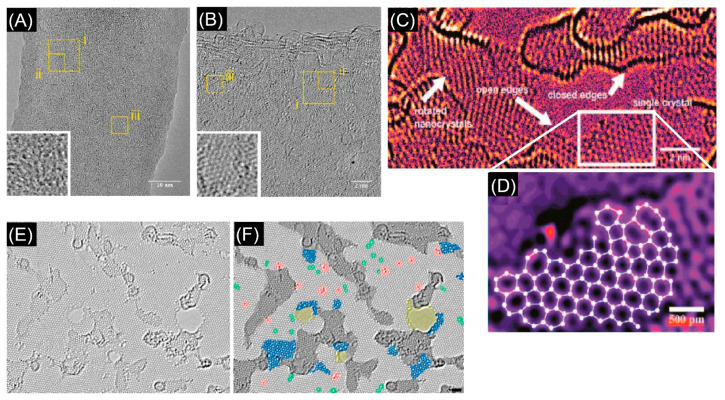
(**A**) Graphene oxide observed at room temperature and (**B**) at 700 °C which in AC-TEM appear notably different. (**C**) a region of graphene oxide in false color to enhance contrast. Both open (monolayer, low contrast) and closed (folded over bilayer, high contrast) edges are visible as are regions of disorder that separate the nanocrystallites from each other. (**D**) shows the atomic structures of the region of monolayer GO indicated in (**C**). (**A**–**D**) reprinted with permission from [551]. Copyright 2016 American Chemical Society. (**E**) TEM image of a single layer reduced-graphene oxide membrane (**F**) with color added to highlight the different features: defect free crystalline graphene in light gray; contaminated regions in dark gray; disordered single-layer carbon networks, or extended topological defects in blue; red areas highlight individual ad-atoms or substitutions; green areas indicate isolated topological defects. Reproduced with permission from [552]. Copyright 2010 American Chemical Society.

**Figure 31 nanomaterials-11-00967-f031:**
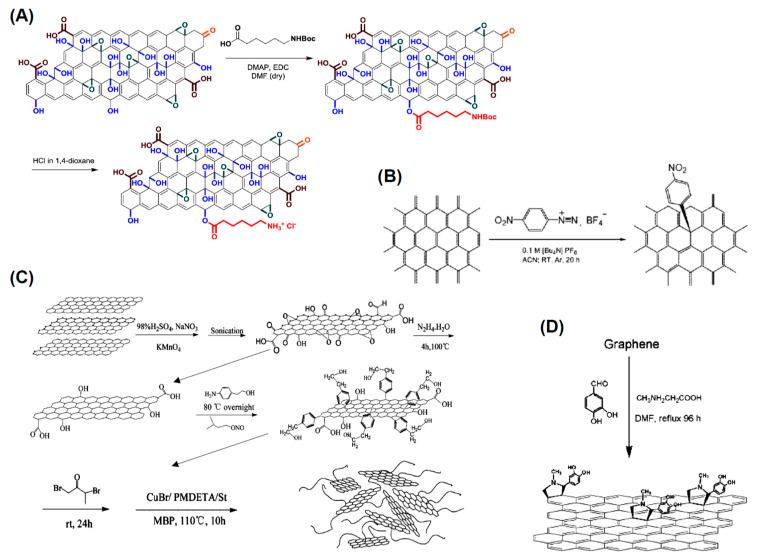
Some examples of graphene functionalization. (**A**) Possible oxide functional groups of graphene; esterification of the GO hydroxyl groups performed using Boc-protected aminocaproic acid in the presence of EDC and DMAP as coupling agents; Boc cleavage performed using HCl in 1,4 dioxane, leading to amine functionalized GO. Reprinted with permission from [562] Copyright 2018 Institute of Physics. (**B**) Schematic illustration of the spontaneous grafting of aryl groups to epitaxial graphene via reduction of 4-nitrophenyl diazonium (NPD) tetrafluoroborate. Adapted with permission from [586]. Copyright 2009 American Chemical Society. (**C**) Covalent polymer functionalization of graphene: synthesis route of polystyrene-functionalized graphene nanosheets. Reprinted with permission from [584]. Copyright 1991 Royal Society of Chemistry. (**D**) Schematic representation of the 1,3 dipolar cycloaddition of azomethine ylide to defect-free graphene. Adapted with permission from [589]. Copyright 1996 Royal Society of Chemistry.

**Figure 32 nanomaterials-11-00967-f032:**
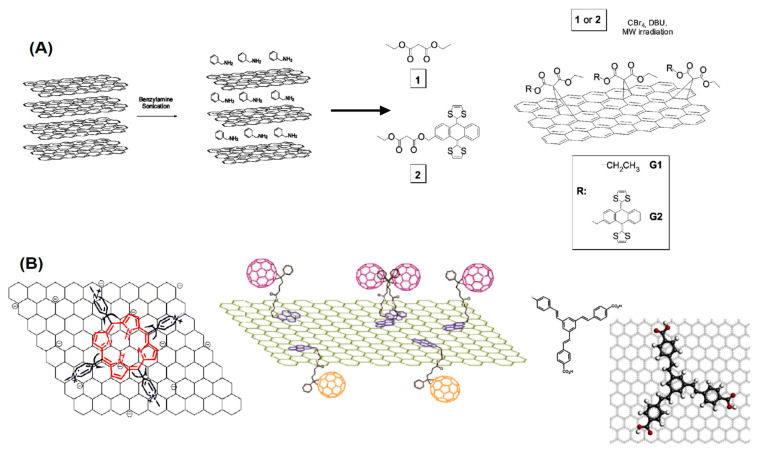
(**A**) Left: exfoliation of graphite flakes by sonicating in benzylamine; (right) functionalization of graphene by a Bingel reaction using 1,8-Diazabicyclo[5.4.0]undec-7-ene (DBU) and CBr4 and diethylmalonate. Adapted with permission from [594]. Copyright 2010 American Chemical Society. (**B**) Examples of non-covalent π-π interaction. Left, the four cationic methylpyridinium molecules (blue) are used to bind a porphirin moiety (red) to a graphene sheet. Reprinted with permission from [599]. Copyright 2009 American Chemical Society. Center Schematic π-π interaction used to bind rGO-pyrene-PCBM spacer to attach C_60_ far from the graphene sheet. Reprinted with permission from [600]. Copyright 2013 American Chemical Society. Right structure of Ramizol- (1,3,5-tris[(1E)-20-(40-benzoic acid) vinyl]benzene and its interaction with a graphene sheet. Reprinted with permission from [601]. Copyright 2015 Elsevier.

**Figure 33 nanomaterials-11-00967-f033:**
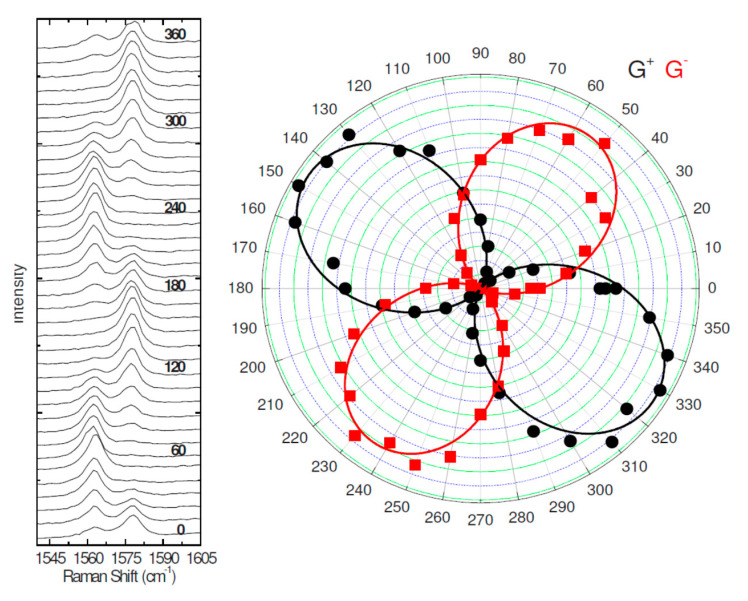
Raman spectra (**left**) and polar plot (**right**) of the fitted G+ and G^−^ peaks as a function of the angle between the incident light polarization and the strain axis θ_in_ measured with an analyzer selecting scattered polarization along the strain axis θ_out_ = 0. Reprinted with permission from [605]. Copyright 2009 American Physical Society.

**Figure 34 nanomaterials-11-00967-f034:**
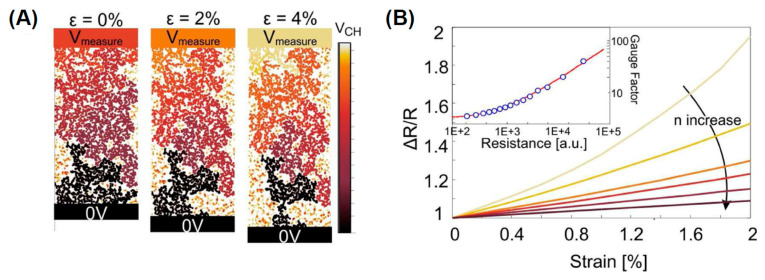
Modeling of percolation through graphene flake network under strain. (**A**) Representation of voltage drop at fixed current in a graphene film at different levels of strain. (**B**) Resistance−strain diagram for different graphene flake number density n. (inset) Gauge factor GF as a function of unstrained resistance R0. Reproduced with permission from [607]. Copyright 2012 American Chemical Society.

**Figure 35 nanomaterials-11-00967-f035:**
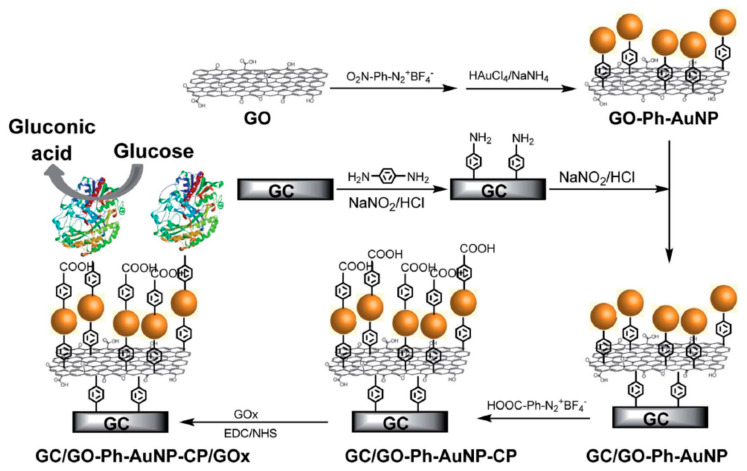
Scheme of a glucose biosensor based on AuNP loaded GO nanocomposites (GO-Ph-AuNP). Based on aryldiazonium salt chemistry, AuNPs were decorated to GO through a benzene bridging. GO-Ph-AuNP was then attached to the 4-aminophenyl modified GCE by C–C bonding. The formed GC/GO-Ph-AuNP interface was further modified with 4-carboxyphenyl before covalent attachment of glucose oxides (GOx) by amide bonds to achieve the GC/GO-Ph-AuNP-CP/GOx sensing interface. Reproduced with permission from [614]. Copyright 2016 Royal Society of Chemistry.

**Figure 36 nanomaterials-11-00967-f036:**
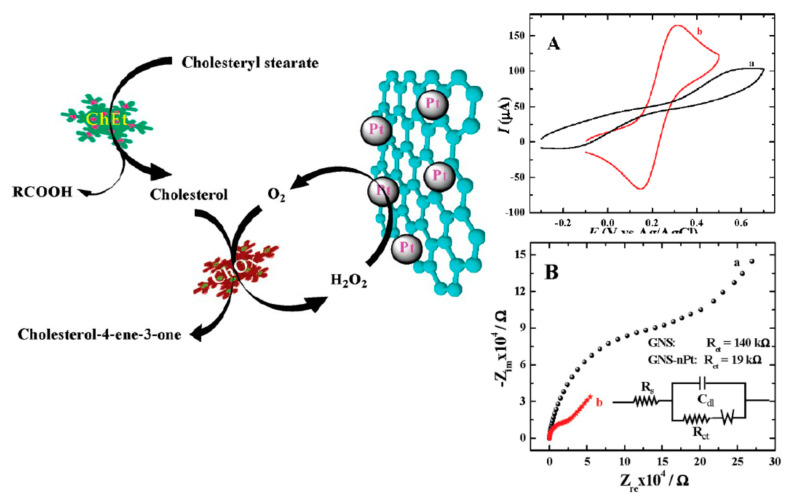
Left: Scheme Illustrating the Biosensing of Cholesterol Ester with the GNS-nPt-Based Biosensor. Right: Cyclic voltammetric (**A**) and impedance (**B**) response of graphene nanosheets (a) and graphene/Pt (b) electrodes toward the redox marker Fe(CN)_6_^3−^ (2 mM) in 0.1 M PBS: scan rate, 100 mV/s. The inset in the impedance plot is the equivalent circuit used to fit the data. Reproduced with permission from [628]. Copyright 2010 American Chemical Society.

**Figure 37 nanomaterials-11-00967-f037:**
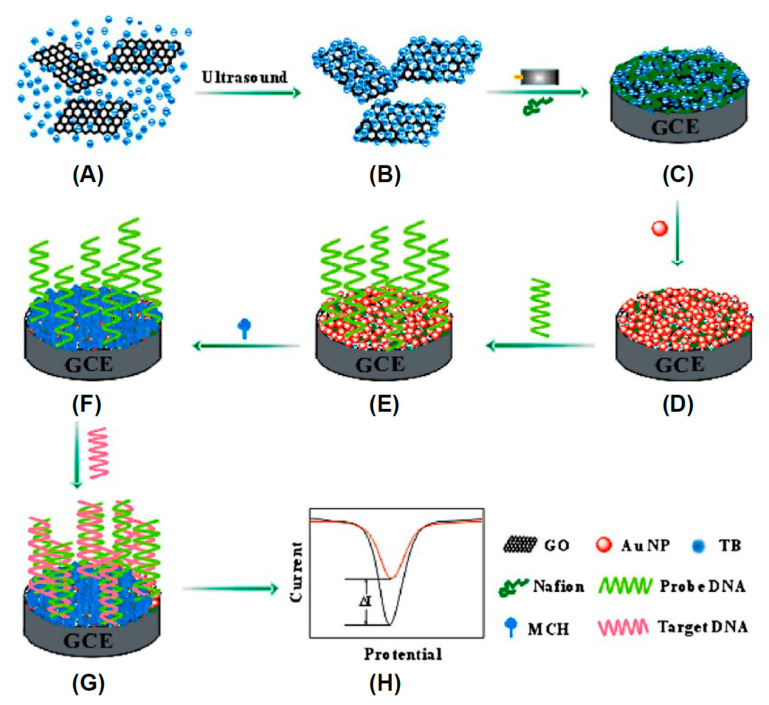
(**A**,**B**) coupling toluidine blue to GO; (**C**) deposition of the functionalized GO to a GCE; (**D**) linking Au NP to the electrode; (**E**) adsorption of the AuNP to the GO/toluidine/GCE; thiol terminated single strand DNA probes where covalently bonded to the Au NPs; (**F**) the electrode stabilized with mercapto-1-hexanol is ready for DNA detection; (**G**) hybridization of the probe and target DNA sequences; (**H**) amperometric detection of the target DNA. Reprinted with permission from [658]. Copyright 2015 Elsevier.

**Figure 38 nanomaterials-11-00967-f038:**
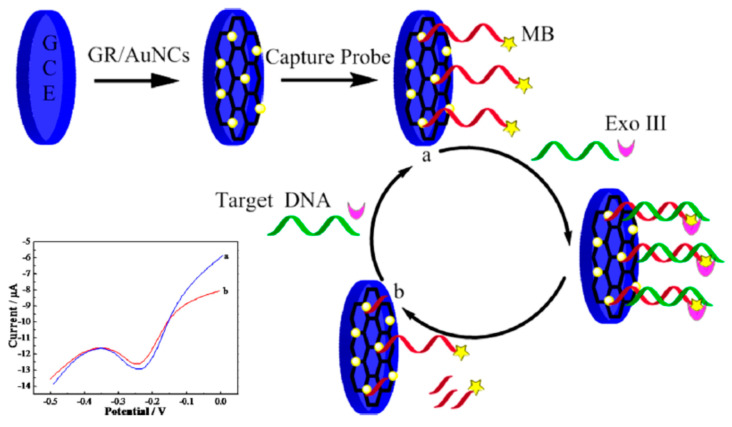
Au nanocluster decorated graphene is deposited on a GCE. Then a single strand capture probe is attached to the Au nanoclusters leading to an output current signal (a). In presence of the target DNA hybridization occurs and a variation of the current at the sensor electrode is obtained (b). The presence of Exo III leads to the digestion of the capture probe with release of the target DNA which is then ready for further hybridization reactions. In this way, an ultrasensitive signal-off electrical biosensor was realized. Reproduced with permission from [660]. Copyright 2015 American Chemical Society.

**Figure 39 nanomaterials-11-00967-f039:**
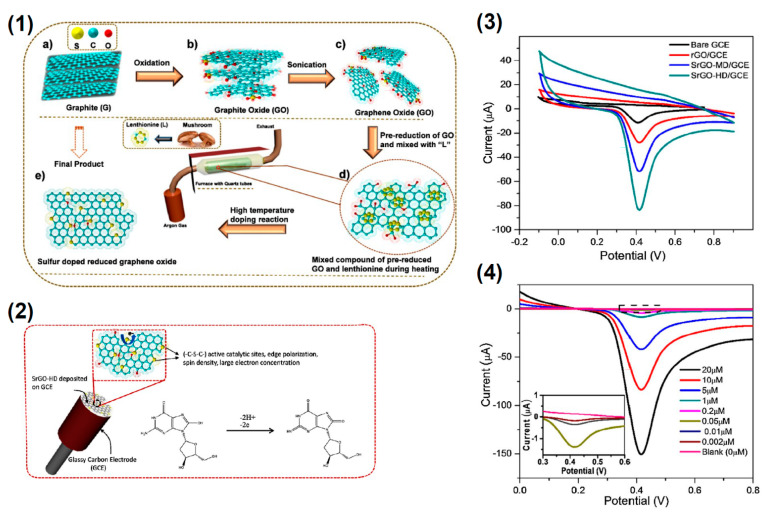
(**1**) Synthesis of S-doped reduced graphene oxide: (a) Graphite; (b) Oxidation of graphite to get graphite oxide (GO); (c) Sonication of GO to get exfoliated graphene oxide; (d) mixing of pre-reduced graphene oxide with lenthionine; (e) final product after high-temperature thermal treatment to obtain S-doped reduced graphene oxide; (**2**) The schematic representation of GCE modification and mechanism of electrochemical sensor for 8-OHdG by sulfur-doped reduced graphene oxide. (**3**) Cyclic voltammograms of bare GCE, rGO/GCE, SrGO-MD/GCE, and SrGO-HD/GCE in the presence of 10 μM 8-OHdG in0.1 M PBS (pH 7.2) at scan rate of 100 mV/s. (**4**) Biosensor responses at different concentrations for 8-OHdG with SrGO-HD/GCE sensor (inset shows the dotted rectangle zoom-in of lower concentrations) in 0.1 MPBS (pH 7.2) at scan rate of 100 mV/s;. Reproduced with permission from [667]. Copyright 2017 Elsevier.

**Table 1 nanomaterials-11-00967-t001:** Detection modality, analyte and detection limit of fullerene-based sensors.

Method	Functionalization	Analyte	LOD	Reference
Resistivity	C_60_ films	Temperature, Pressure	2%/°C	[145]
Chronoamperometry	Pd@Cys-C_60_	Glucose	1 μM	[172]
Amperometric	Carboxylated C_60_	Glucose	2 mg/mL	[149]
Amperometric	C_60_-GOD	Glucose	1.6 × 10^−6^ M	[169]
Piezoelectric	C_60_-GOD	Glucose	3.9 × 10^−5^ M	[170]
Potentiometric	C_60_-urease	Urea	8.28 × 10^−5^ M	[171]
Microbalance	C_60_ SAM (^1^)	VOCs (^2^) of methanol, hexane, benzene, toluene, aniline	High gas selectivity	[153]
Capacitive	C_60_ Alumina	Moisture	1 ppm	[146]
Voltammetry	C_60_ nanorods	Ethylparaben	3.8 nM	[160]
Amperometric	ZnPp-C_60_	H_2_O_2_	~0.81 μM	[161]
Cyclic voltammetry	NiNP-CuNP-C_60_	Vitamin D_3_	0.0024 1 μM	[173]
Amperometric	Cys-PTC-NH_2_-C_60_	miRNA-141	7.78 fM	[164]
Impedimetric	poly-hydroxylated fullerene	Fetuin-A	1.44 ng/mL	[166]
Amperometric	C_60_-PAn	*Mycobacterium tuberculosis*	20 fg/mL	[167]
Differential pulse Voltammetry	ACV-C_60_	Acylovir	1.48 nM	[174]
Electrochemical impedance spectroscopy	C_60_-Polyacrilamide	Cortisol	0.14 nM	[175]
Capacitive	C_60_-Amide C_60_-carboxyl C_60_-uracil	ATP	0.31 mM	[176]
Piezoelectric	Anti-IgG-PVA-C_60_	IgG	0.1 μg/mL	[177]

(^1^): SAM: self assembled monolayer. (^2^) VOCs: Volatile Organic Compounds.

**Table 2 nanomaterials-11-00967-t002:** Detection modality, analyte and detection limit of GCO based sensors.

Method	Functionalization	Analyte	LOD	Reference
Potentiometric	PolyDopamine-OLC	pH	-	[184]
Amperometric	Pd—OLC	Hydrazine, H_2_O_2_	14.9 nM, 79 nM	[182]
Resistivity	P—OLC	NH_3_	10 ppb	[183]
Potentiometric	PolyDopamine-OLC	pH	-	-
Resistivity	OLC/C_2_H_6_O	H_2_	<10 ppm	[180]
Fluorimetric	Oxidized OLC	Glucose	1.3 × 10^−2^ M	[185]
Amperometric	OLC-PDDA	Glucose	1 mM	[188]
Amperometric	CNO-biotin	DNA	LOD = 3.9 nM	[190]
Cyclic voltammetry	CNO-PDDA	dopamine, epinephrine, and norepinephrine	100 nM	[191]
Cyclic voltammetry	CNO	dopamine	1.23 μM	[192]

**Table 3 nanomaterials-11-00967-t003:** Detection modality, analyte and detection limit of ND based sensors.

Method	Functionalization	Analyte	Performances	Reference
NV^−^—quantum sensing	-	chemokine receptor CXCR4	Single molecule	[244]
NV^−^—quantum sensing	-	TEMPOL	fM	[245]
NV^−^—quantum sensing	-	E field	202 ± 6 V/cm√Hz	[246]
NV^−^—quantum sensing	-	E field	150 mV/cm√Hz	[247]
NV^−^—quantum sensing	-	Crystal stress	{0.023; 0.030; 0.027} GPa/Hz^1/2^	[248]
NV^−^—quantum sensing	-	Crystal stress	~0.1 MPa at 10 mK	[249]
SiV^−^	-	Cell temperature	521 mK/Hz^1/2^	[252]
SiV^−^	-	Cell high resolution microscopy	<150 nm	[255]
SiV^−^	-	Cell high resolution microscopy	<90 nm	[256]
Fluorescent ND	-	Lipoproteins tracking	Tracking duration = 12 h	[257]
Fluorescent ND	-	Cell exocytosis	Analysis duration = 8 days	[258]
Voltammetry	-	Pyrazinamide	2.2 × 10 ^−7^ mol/L	[261]
Suare wave voltammetry	-	Bisphenol	5 nM	[262]
Voltammetry	-	Cd and Pb ions	0.42 and 5.3 μA/(μmol cm^2^)	[263]

**Table 4 nanomaterials-11-00967-t004:** Detection modality, analyte and detection limit of CQDs based sensors.

Method	Functionalization	Analyte	Performances	Reference
FRET ^1^	-	Hg^2+^, Cu^2+^, Fe^3+^	Hg^2+^: 10 μM–0.2 nM Cu^2+^: 1 μM–0.58 pM Fe^3+^: 17.5 μM–2 nM	[295]
Fluorescence turn-off	-	Fe^3+^	1 ppm	[296]
Fluorescence turn-off	Oligodeoxyribonucleotide	Hg^2+^	5–200 nM	[297]
Fluorescence turn-off	Carboxyl, hydroxyl and amine	Cu^2+^	23 nM	[298]
Fluorescence turn-off	N doping	Cr^6+^	40 nM	[299]
Fluorescence turn-off	Carboxyl, hydroxyl	Pb^2+^	4.5 ppb	[300]
Fluorescence turn-off	N doping	Au^3+^	64 nM	[301]
Fluorescence turn-off	O, N functionalities	K^+^	0.0570 μM	[302]
Fluorescence enhancement	Carboxyl, hydroxyl	Ag^+^	320 nM	[303]
Fluorescence turn-on	Carboxyl	S^2−^	1.72 μM	[304]
Fluorescence turn-on	Carboxyl	H_2_S	0.7 μM	[305]
Fluorescence turn-on	Polyethylenimine	CN^−^	0.65 μM	[306]
Fluorescence turn-on	Carboxyl	PO_4_^3−^	15 μM/L	[307]
Fluorescence turn-on	Amine	SCN^−^	0.36 μM	[308]
Fluorescence lifetime	O, N functionalities	Cell imaging	-	[310]
Voltammetry	Carbonyl, hydroxyl/chitosan	Dopamine	11.2 nM	[311]
Pulse voltammetry	GO/CQDs	Dopamine	1.5 nM	[312]
Cyclic voltammetry	NiAl/CQDs	Acetylcholine	0.14 μM	[313]
Voltammetry	N doping	Epinephrine	3 × 10^−13^ M	[314]
Fluorescence turn-off	Boronic acid coordinated CQDS	Glucose	-	[315]
Fluorescence enhancement	N doping	Amoxicillin	-	[318]
FRET	O functionalities	Vitamin B_12_	0.1 μg/mL	[319]
FRET	Carboxyl and amine	Hemoglobin	30 pM	[320]
Fluorescence turn-on	Methylene blue/CQDs	DNA	1.0 μM/L	[321]
Fluorescence enhancement	S, N doping	Trichlorophenol	0.07 μg/mL	[323]
FLIM ^2^	N, Cl	pH	-	[325]
Fluorescence turn-off	Hydroxyl, amine	pH, cyt C	3.6 mg/L for cyt C	[326]
Fluorescence turn-off	Au-PAMAM/CQDs	CA125	0.5 fg/mL	[327]
Imaging	N doping	C12, A549, HepG 2, and MD-MBA-231 cells	-	[329]
Imaging	O functionalities	Reticuloendothelial system and kidneys	-	[332]
Imaging	N doping	Glioma tissues	-	[333]
Imaging	CQDs/Pt(IV)	Drug delivery	-	[335]
Imaging	Polyethylenimine/CQDs–hyaluronic/Doxorubicin	Hyaluronidase detection, self-targeted imaging and drug delivery	-	[336]
Photodynamic theraphy	PEG-Arg-Gly-Asp motif-protoporphyrin-carbon nitride/CQDs	Solid tumor tissues	-	[337]
Crossing the blood brain barrier	-	Β-amyloid targeting	-	[338]
Imaging	N-functionalities	GABA- and glutamatergic neurotransmission	-	[339]

FRET = fluorescence resonance energy transfer. FLIM = Fluorescence Lifetime Imaging Microscopy.

**Table 5 nanomaterials-11-00967-t005:** Detection modality, analyte and detection limit of CNT based sensors.

Method	Functionalization	Analyte	Performances	Reference
Chemo-Resistivity	GOx-poly(4-vinylpyridine)/SWCNT	Glucose	Linear between 0.08 and 2.2 mM	[475]
Chemo-Resistivity	-	NH_3_	20 ppb	[431]
Chemo-Resistivity	Paper/CNTs	NH_3_	0.36 to 2.7 ppm	[433]
Chemo-Resistivity	Pd/poly(4-vinylpyridine)-CNTs	Thioethers	0.1 ppm	[434]
Conductivity	Pristine SWCNT	NO_2_, nitrotoluene	44 ppb, 262 ppb	[435]
Chemo-Resistivity	Pt/MWCNTs films	NO_2_	10 ppb	[436]
Chemo-Resistivity	SnO_2_/SWCNTs	NH_3_, O_3_	1 ppm, 20 ppb	[439]
Chemo-Resistivity	SnO_2_/MWCNTs	CH_4_	10 ppm	[445]
Chemo-Resistivity	ZnO/MWCNTs	CH_4_	2 ppm	[446]
Amperometric	Fe Porphyrin/SWCNTs	CO	80 ppm	[447]
Amperometric	Oxidized CNTs	CO	5 ppm	[449]
Chemo-Resistivity	CP^CoI_2/_CNTs	CO	90 ppm	[450]
Plasmonics	Au/CNTs	CO_2_	150 ppm	[451]
Chemo-Resistivity	SnO_2/_CNTs	CO	1 ppm	[453]
Amperometric	Fe-Porphyrine/CNTs	Benzene, xylene, toluene	500 ppb–10 ppm	[456]
Impedence spectroscopy	Oxygen functionalities	Benzene, toluene, methanol, ethanol and acetone	1.62 for benzene 1.8ppm for toluene	[457]
Chemo-Resistivity	Poly-porphyrins/CNTs	Acetone	9 ppm	[458]
Chemo-Resistivity	Poly(aniline)/SWCNTs	HCl, NH_3_, pH	100 ppb for HCl	[460]
Chemo-Resistivity	poly(1-amino anthracene)/SWCNTs	pH	-	[461]
Chemo-Resistivity	SnO_2_/CNTs	Ethylene, NO_2_	3 ppm for ethylene 50 ppb for NO_2_	[465]
Chemo-Resistivity	tris(pyrazolyl)-borate copper(I)/CNTs	Ethylene	<1ppm	[466]
Chemo-Resistivity	OR 2AG1 protein/CNTs	Amyl Butirate	1 fM	[467]
Chemo-Resistivity	cobalt meso-aryl-porphyrin/CNT	NH_3_, putresceine, cadaverine	0.5 ppm for NH_3_	[469]
Cyclic voltammetry	GOx/CNT	Glucose	0.5 mM/L	[471]
Amperometry	Pt/CNT	Glucose	14 μA/mM	[472]
Amperometry	GOx/CNT	Glucose	sensitivity of 18.6 μA/(m cm^2^)	[474]
Chemo-Resistivity	GOx/CNT	Glucose	2.2 mM	[475]
Amperometry	Uricase/CNTs, cholesterol-esterase/CNTs	Cholesterol, uric acid	0.0721 and 0.0059 μA per mg/dL	[476]
Amperometry	Polyaniline and cholesterol esterase/CNTs	Cholesterol, glucose, uric acid acetaminophen and ascorbic acid	22 μA/(mg dL)	[477]
Amperometry	DMF/CNTs	Insulin	14 nM	[478]
Amperometry	RuOx/CNTs	Insulin	1 Nm	[479]
Fluorescence turn-off	SWCNT	NO	Single molecule	[481]
Fluorescence turn-off	ds(AAAT)_7_ oligonucleotide/CNTs	NO	1 μM in vivo	[482]
Amperometry	N-hydroxyphenyl maleimide/CNTs	Epinephrine	0.02 ng/mL	[483]
Square wave voltammetry	Au/CNTs	Dopamine	0.071 μM	[484]
Fluorescence turn-off	SWCNTs	H_2_O_2_	Single molecule	[486]
Resistivity	SWCNTs	DNA	4 × 10^13^ molecules/cm^2^	[487]
Chronopoterntiometric	alkaline phosphatase/CNTs	DNA	1 fM	[488]
Amperometry	Fmoc-RRMEHRMEW/CNTs	DNA	0.88 μg/L	[490]
Amperometry	Oxygen functionalities	DNA	141.2 pM	[491]

**Table 6 nanomaterials-11-00967-t006:** Detection modality, analyte and detection limit of electrochemical graphene-based sensors.

Method	Functionalization	Analyte/Entity	Performances	Reference
Optical	Pristine Graphene	Young Modulus	1.9 ± 0.4 cm^−1/^GPa	[604]
Optical	Pristine Graphene	Young Modulus	∂ω*_G_*_+_/∂ε~−18.6 cm^−1^/% ∂ω*_G_*_−_/∂ε~−36.4 cm^−1^/%	[605]
Piezo-Resistivity	Pristine Graphene	Strain	GF^1^ = from 4 to 14 for strains > 1.8%	[608]
Piezo-Resistivity	Reduced Graphene	Strain	GF = 9.49	[609]
Piezo-Resistivity	Reduced Graphene	Strain	Sensitivity of 0.18 kPa^−1^	[610]
Amperometric	GOx-Au/GO	Glucose	Sensitivity of 42 mA mM^−1^cm^−2^	[614]
Amperometric	NiCo2N/GO	Glucose H_2_O_2_	LOD: 50 nM for glucose 200 nM for H_2_O_2_	[622]
Amperometric	Cu/GO	Glucose	48.13 μA mM^−1^	[623]
Amperometric	Pt/GO	Glucose	Sensitivity ofSensitivity 137.4 μA mM^−1^cm^−2^	[624]
Amperometric	PtNi/GO	Glucose	Sensitivity 20.42 μA mM^−1^cm^−2^	[625]
Amperometric	ChOx-ferrocene/GO	Cholesterol	0.1 mM	[626]
Amperometric	poly(vinylpyrrolidone)/poly(aniline)/graphene	Cholesterol	1 μM	[627]
Amperometric	ChOx,ChEt/Pt/graphene	Cholesterol via H_2_O_2_ Cholesterol direct	0.5 nM (H_2_O_2_) 0.2 μM (Cholesterol)	[628]
Amperometric	Dendritic Pd/rGO structure	Cholesterol	0.05 mM	[629]
Amperometric	Pt-Pd chitosan/Graphene	Cholesterol	LOD: 0.75 μM	[630]
Amperometric	polyaniline nanofiber/graphene	Cholesterol	LOD: 1.93 mg dL^−1^	[631]
Amperometric	NiO/graphene	Cholesterol	LOD: 0.13 mM	[632]
Amperometric	Graphene	H_2_O_2_	LOD: 0.19 μM	[634]
Amperometric	Pt/Graphene	H_2_O_2_	LOD: 80 nM	[635]
Amperometric	Ag/Graphene	H_2_O_2_	LOD: 14.9 μM	[636]
Amperometric	Ag rods/rGO	H_2_O_2_	LOD: 2.04 μM	[637]
Amperometric	Ag wires/Graphene	H_2_O_2_	Sensitivity of 12.37 μA mM^−1^cm^−2^	[638]
Amperometric	Au/Graphene	H_2_O_2_	Sensitivity of 75.9 μA mM^−1^cm^−2^	[639]
Amperometric	P/Graphene	H_2_O_2_	LOD: 0.1 μM	[640]
Amperometric	PtNi nanowires/Graphene	H_2_O_2_	LOD: 0.31 nM	[641]
Amperometric	Cu_2_O/(3D, 2D) rGO structure	H_2_O_2_	LOD: 0.37 μM	[642]
Amperometric	WO_4_ ^−2^/G	NADH, NAD^+^	LOD: 2.188 μM for NADH and 49.8 μM for NAD^+^	[643]
Chronoamperometry	GO	Dopamine	LOD: 0.27 μM	[644]
Square voltammetry	AU/Graphene	Dopamine, Ascorbic acid, Uric acid	LOD: 10 μM for DA, 0.15 μM for AA, and 0.21 μM for UA	-
Amperometric	Pt/Graphene	Dopamine, Ascorbic acid, Uric acid	Sensitivity: 0.186 μA M^−1^ cm^−2^ for AA, 9.199 μA M^−1^ cm^−2^ for DA and 9.386 μA M^−1^ cm^−2^ for UA.	[646]
Voltammetry	Ag/rGO	Dopamine, Ascorbic acid, Uric acid	LOD: 0.81 μM for DA, 0.26 μM for AA and 0.30 μM for UA	[647]
Voltammetry	Pd-chitosan/Graphene	Dopamine, Ascorbic acid, Uric acid	LOD: 0.1 μM for DA, 20 μM for AA and 0.17 μM for UA	[648]
Pulse voltammetry	MoS_2_ decorated PANI/rGO	Dopamine, Ascorbic acid, Uric acid	LOD: 0.7 μM for DA, 22.2 μM for AA and 0.36 μM for UA	[649]
Differential pulse voltammetry	ZnO nanowire array/Graphene foam	Dopamine, Ascorbic acid, Uric acid	LOD: 0.5 μM for DA, 5 μM for AA and 0.5 μM for UA	[650]
Amperometry	SnO_2_/rGO	Urea	LOD: 11.7 fM	[651]
Differential pulse voltammetry	Au/rGO	DNA	LOD: 35 aM	[654]
Impedance spectroscopy	nanorods/polythionine-Au/Graphene	DNA	LOD: 4.03 × 10^−14^ mol/L	[655]
Differential pulse voltammetry	riboflavin 50-monophosphate Na/graphene	DNA	LOD: 8.3 × 10^−17^ M	[656]
Differential pulse voltammetry	Chitosan/GO	DNA	LOD: 10 fM	[657]
Amperometry	AuNPs-toluidine blue/GO	DNA	LOD: 2.95 × 10^−12^ M	[658]
Differential pulse voltammetry	Au nanorods/GO	DNA	LOD: 3.5 fM	[659]
Differential pulse voltammetry	Au/Graphene	DNA	LOD: 30 aM	[660]
Amperometry	Graphene FET ^1^	DNA	LOD: 1.7 fM	[661]
Amperometry	Graphene FET ^1^	DNA	LOD: <1 fM	[662]
Voltammetry	Pd/rGO	carcinoembryonic antigen	LOD: 10 pg mL^−1^	[663]
Resistivity	Pd, Pt, Ag and Au/rGO	ErbB2	LOD: 1.0 fM to 0.5 μM	[664]
Differential pulse voltammetry	Fe_2_N/rGO	4-nitroquinoline N-oxide	LOD: 9.24 nM	[665]
Differential pulse voltammetry	folic acid-octadecylamine/GO	HepG2 cells	LOD: 5 cells mL^−1^	[666]
Differential pulse voltammetry	S-doped rGO	8-hydroxy-2′-deoxyguanosine (8-OHdG) molecules	LOD: ~1 nM	[667]
Differential pulse voltammetry	Au/Graphene	cytokeratin 19 fragment antigen 21-1	LOD: 100 pg/mL	[668]
Differential pulse voltammetry	CoS_2_/Graphene	CA15-3 antigen	LOD: 0.03 U mL^−1^	[669]
Chronoamperometric	Pd/rGO	prostate-specific antigen	Sensitivity: 28.96 μA mL ng^−1^ cm^−2^	[670]
Amperometry	ZrO_2_/rGO	CYFRA-21-1 biomarker	LOD: 0.122 ng mL^−1^	[671]
Amperometry	Fe_3_O_4_/GO	detection of prostate specific antigen or prostate specific membrane antigen	LOD: 15 fg/mL for PSA, 4.8 fg/mL for PSMA	[672]
DPASV	CeO_2_/Graphene	Cd(II)	LOD: 0.195 nM	[697]
SWASV	Sn/rGO	Cd(II)	LOD: 0.63 nM	[698]
DPASV	Fe_2_O_3_/GO	Cd(II)	LOD: 0.08 μg/L	[699]
DPASV	CeO_2_/Graphene	Pb(II)	LOD: 0.10 nM	[697]
SWASV	Sn/rGO	Pb(II)	LOD: 0.6 nM	[698]
DPASV	Fe_2_O_3_/GO	Pb(II)	LOD: 0.07 μg/L	[699]
DPASV	CeO_2_/Graphene	Hg(II)	LOD: 0.277 nM	[697]
DPASV	CeO_2_/Graphene	Cu(II)	LOD: 0.164 nM	[697]
SWASV	Sn/rGO	Cu(II)	LOD: 0.52 nM	[698]
DPV	Ru/GO	As(III)	LOD: 2.30 nM	[700]
DPASV	Graphene	Zn(II)	LOD: 1.0 μg/L	[701]
DPASV	Fe_2_O_3_/GO	Zn(II)	LOD: 0.11 μg/L	[699]
DPV	C nanohorn/GO	4-nitrochlorobenzene	LOD: 10 nM	[681]
Amperometry	poly(Na styrenesulfonate)/Graphene	Hydrazine	LOD: 1 μM/L	[682]
Amperometry	Reduced Graphene	Sulfides	LOD: 4.2 μM	[683]
Voltammetry	Polyaniline/Graphene	4-aminophenol	LOD: 65 nM	[684]
DPV	Polyoxometalate/Graphene	Diphenols	LOD: 40 nM	[685]
Voltammetry	Graphene	Hydroquinone	LOD: 40 μM	[686]
Conductometric	Pt/Holey-graphene	H_2_	LOD: 20 ppm	[688]
Chemoresistive	Cu/Graphene	CO	LOD: 0.25 ppm	[689]
Conductometric	Graphene	CO_2_	Sensitivity 0.17%/ppm	[690]
Optical	Graphene coupled Long-Period Fiber Grating	NO	Sensitivity: 63.65 pm/ppm	[691]
Chemoresistive	PS/Graphene	NO_2_	LOD: 0.5 ppm	[692]
Chemoresistive	rGO	SO_2_	LOD: 5 ppm	[693]
Chemoresistive	Polyaniline/rGO	NH_3_	LOD: 0.2 ppm	[694]

^1^ FET: field effect transistor.

## Data Availability

All the data presented in this manuscript were derived from the indicated articles which are published in literature and listed in the Reference section.

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
