# Peer review of "Carbon Nanomaterials: Synthesis, Functionalization and Sensing Applications"

_nanomaterials, 2021, doi:10.3390/nano11040967_

Round 1

Reviewer 1 Report

This is a long review with a large number of references. The author already published a review on a closely related topic (Carbon nanomaterials, J Carbon Res 2019).

In my view, the title should be modified to «Carbon nanomaterials: Synthesis, functionalization and sensing applications», in order to better reflect the content.

I find the review too descriptive, with long enumerations of works but not much critical analysis. The literature is full of results that have no real applicability. We do not need a list of them all.

What are the advantages of the carbon nanomaterials with respect to sensing in specific cases? For instance, glucose sensing is mentioned (512, 1588). Are the described sensors superior to the existing (even commercial) ones?

Which properties of the nanomaterials are used in the sensors (luminescence, etc)? What is being sensed (physical properties, biological molecules, ions, gases, etc)? A table or some other systematization method would help in this respect. 

The English requires extensive revision. Sometimes, sentences make no sense, e.g. 529-530. There are also many typos, even in figures, e.g. Fig 2. 

Two specific points:

87 It is stated that Kroto discovered the fullerenes - however it was in fact  Kroto and Smalley (at least)

141 In Fig 3 the author shows the colours of solutions of fullerenes but omits C76.

Sensing using the fluorescence of fullerenes (mainly C70) namely for the measurement of temperature and oxygen is not mentioned.

I believe the author should point out what is new in his review, and should briefly discuss previous reviews on this and closely related topics.

Reviewer 2 Report

This work is well within the scope of Nanomaterials and it may be of interest to most of the readers of this journal. It is very well organized, covering all the aspects of carbon nanomaterials for sensing applications in an extended 90 pages review. Two minor issues can be depicted to improve the quality of this Review. The one is the various scattered grammatical errors that can be seen across text. For example, in the abstract (P1, L8: ‘resultsd’), (P9, L310: ‘in the years 2000’), (P67, L2363: ‘assembpled’) and many more, thus a final proofreading would be beneficial. The second is that some figures are of low quality or contain too much text that makes it impossible to read (For example please see: Fig.4, Fig.13, Fig.18, Fig.20, Fig. 29), thus they should be revised. 

Specific comments

P22, Fig.9: There is a ‘?’ within the text of the figure.

L35: ‘sensibility’ is not an appropriate term for characterizing a sensor; please consider using sensitivity

L60: ‘Tipically’

L108: ‘Carbon is a unique the element of the periodic table’ Please check wording

L166-168: ‘In addition reactions one of the important driving forces is the relief of strain affecting the cage structure, enabling the return to sp3 hybridization.’ Please check expression

L207: ‘fnctionalize’

L209-210: ‘Surface functionalization is a convenient route make fullerene soluble in both water and organic solvents’ Please check expression

L293: ‘Some of the functionalization reaction and product are sketched in Figure 4.’ Please check grammar

L353: ‘The different solvent behavior be explained’ Please check wording

L418: ‘In fullerene films and fullerene compounds with iodine are used to fabricate temperature and pressure sensors’ Please check expression

L419: ‘Fullerene films of 2-3 μm were deposited’ Please consider adding ‘thickness’

L645-646: ‘Type IIa diamonds are almost or entirely impurity free are colourless’ Please check expression

L696: ‘sourounded’

L853: ‘[‘

L854: ‘Because the’

Please consider adding ‘of’

L871-872: ‘FNDs are currently utilized follow fast and slow events in cells’ Please check expression

L873: ‘traking’

L883-885: ‘Electrochemical measurements involve the detection different electrostatic potential of two different electrodes immersed in a solution.’ Please check expressions

L932: ‘graphgite’

L1119-1120: ‘Glucose sensing is important whose unbalance’ Please check wording

L1187: ‘mices’ 

L1303: ‘for’ Consider changing to ‘form’

L1494: ‘ranging from’ Consider eliminating

L1555: ‘wit’

L1574: ‘integtrate’

L1594: ‘alignet’

L1674: ‘Graphene is represents’

L1764-1765: ‘In PECVD processes the plasma decomposes the precursor enabling synthesis of graphene at low temperature and short deposition time’ Please check expression

L2130: ‘cholesterolcconcentration’

L2340 ‘highgly’

L2363 ‘assembpled’

P67, The fields: ‘Author Contributions’, ‘Funding’, ‘Data Availability Statement’, ‘Acknowledgments’ and ‘Conflicts of Interest’ should be filled.

Round 2

Reviewer 1 Report

I am happy with the revision, all points raised were satisfactorily answered.